# Dynamic antagonism between key repressive pathways maintains the placental epigenome

Raha Weigert[1,2,13], Sara Hetzel [1,13], Nina Bailly[1,3], Chuck Haggerty[1,3], Ibrahim A. Ilik[4], Philip Yuk Kwong Yung[5,6], Carmen Navarro[5,6], Adriano Bolondi [1,3], Abhishek Sampath Kumar [1], Chiara Anania[7], Björn Brändl[1,8], David Meierhofer[9], Darío G. Lupiáñez[7], Franz-Josef Müller [1,8], Tugce Aktas [4], Simon J. Elsässer [5,6], Helene Kretzmer[1], Zachary D. Smith [10] ✉ & Alexander Meissner [1,3,11,12] ✉

DNA and Histone 3 Lysine 27 methylation typically function as repressive modifications and operate within distinct genomic compartments. In mammals, the majority of the genome is kept in a DNA methylated state, whereas the Polycomb repressive complexes regulate the unmethylated CpG-rich promoters of developmental genes. In contrast to this general framework, the extra-embryonic lineages display non-canonical, globally intermediate DNA methylation levels, including disruption of local Polycomb domains. Here, to better understand this unusual landscape's molecular properties, we genetically and chemically perturbed major epigenetic pathways in mouse trophoblast stem cells. We find that the extra-embryonic epigenome reflects ongoing and dynamic de novo methyltransferase recruitment, which is continuously antagonized by Polycomb to maintain intermediate, locally disordered methylation. Despite its disorganized molecular appearance, our data point to a highly controlled equilibrium between counteracting repressors within extra-embryonic cells, one that can seemingly persist indefinitely without bistable features typically seen for embryonic forms of epigenetic regulation.

DNA methylation is a covalent, reversible epigenetic modification that predominantly occurs at cytosines in the CpG dinucleotide context[1]. In healthy somatic cells, CpG methylation is bistable and largely determined by local CpG density: the majority of genomic CpGs are sparsely distributed and uniformly methylated, while CpG-dense regions—termed CpG islands (CGIs)—found at developmental and housekeeping gene promoters remain fully unmethylated[2,3]. As these genetic elements are generally protected from DNA methylation, transcriptional repression is instead carried out by Polycomb repressive complex (PRC) 1 and 2, chromatin modifiers that are responsible for

catalysing ubiquitylation of lysine 119 on histone H2A (H2AK119ub1) (refs. [4,5]) and mono-, di- and trimethylation of lysine 27 on histone H3 (H3K27me1/2/3) (refs. [6,7]), respectively.

Although PRCs and DNA methyltransferases (DNMTs) biochemically interact, show broad genomic co-occupancy and are both essential for proper cell fate control during early embryogenesis[8–14], these repressive pathways do not appear to simultaneously modify chromatin in healthy somatic cells[15,16]. Developmental gene promoters instead appear to preserve a constitutively unmethylated state for the majority of the mammalian lifecycle. DNA methylation and H3K27me3

can co-occur across CGIs when de novo DNMTs are ectopically over-expressed, but in this context H3K27me3 is depleted over time[15]. In contrast, naïve pluripotent stem cells have demonstrated the unique ability to transition into and out of a globally hypomethylated state without compromising their viability, and do so via genome-wide compensation by PRC2 (refs. [17–20]). Across these cases, the co-existence of PRC2 and DNMT-associated modifications is generally considered to be transient and unstable, such that repressive chromatin is ultimately dominated by one or the other.

The governing principles of epigenomic regulation are far more completely understood for embryonic stem cells (ESCs) and their derivatives than they are for the extra-embryonic lineages that emerge in parallel over early mammalian development. Notably, the major extra-embryonic lineages—the placenta-forming extra-embryonic ecto-derm (ExE) and yolk sac-forming extra-embryonic endoderm—both differentiate away from the embryo proper during pre-implantation development, a period of global DNA hypomethylation that follows fertilization[2,21,22]. As the embryo implants, the extra-embryonic lineages diverge to acquire an atypical epigenomic landscape characterized by globally intermediate methylation that encroaches into canonically protected Polycomb territories found at developmental genes[23,24]. Over the past several decades, ESCs[25], epiblast stem cells[26,27], trophoblast stem cells (TSCs)[28] and extra-embryonic endoderm stem cells[29] have been utilized as powerful cell culture models that preserve some degree of their native developmental potential and are believed to reflect the epigenetic status of transient progenitor states[30]. Many key discoveries about epigenetic regulation have been derived exclusively from mouse ESCs[14,31], while models for other lineages have overall received less attention[32]. As a result, the rather unusual epigenome of mouse and human extra-embryonic cell lines has not been investigated in comparative detail[28,30,33,34].

In this Article, we sought to investigate the fundamental molecular principles of the mouse TSC epigenome through a combination of chemical and genetic perturbation experiments. In particular, we highlight an active, ongoing recruitment of DNA methyltransferase 3B (DNMT3B) to direct global and CGI-specific methylation levels and show a counteracting role for Polycomb to prevent global hypermethylation. Moreover, we find that this intermediate methylation landscape is strikingly elastic and can be drawn to high or low global methylation values without losing the ability to return to intermediate steady-state levels. Together, our findings provide crucial insights into the complex interplay of positive and negative regulators of DNA methylation, including a non-canonical, but nonetheless stably propagating configuration of epigenetic repressors within extra-embryonic lineages.

## Results

### Mouse TSCs preserve intermediate methylation

Multiple regulatory and biochemical properties of the extra-embryonic epigenome remain unknown. We therefore evaluated the utility of mouse TSCs as a cell culture model for a systematic multi-layered investigation[28,30]. We first generated whole genome bisulfite sequencing (WGBS) data for four different TSC lines (male and female lines, derived in two different labs) and found that intermediate methylation levels seen in vivo are retained and most pronounced across megabase-scale partially methylated domains (PMDs)[35]. Similarly, we confirmed that hypermethylated CGIs (hyper CGIs, defined in mouse ExE) remain intermediately methylated and notably overlap with canonical targets of PRC2 in mouse ESCs (Fig. 1a,b and Extended Data Fig. 1a–e). In combination, these differentially methylated features represent a major departure from the somatic methylome, which acquires its bimodal status within the post-implantation epiblast and is then propagated throughout subsequent development[23,36].

The persistence of extra-embryonic methylation patterns in vitro allowed us to functionally evaluate whether intermediate methylation is still primarily maintained by the methyltransferase DNMT1 or

requires continuous de novo methyltransferase activity to counteract constant turnover. Interestingly, we found that TSCs maintain their global methylation levels in a state of high entropy[37–40], a disordered form of DNA methylation characterized by a broad distribution of unique epialleles across individually measured reads (Fig. 1c and Extended Data Fig. 1f). We reasoned that, if these distinct patterns are largely non-dynamic, single cell-derived subclones would differ substantially from the bulk population because they would largely propagate inherited methylation patterns from their parent cells. In contrast, if high entropy is better explained by dynamic exchange between methylated and unmethylated states, subclonal lines would quickly re-establish high entropy levels due to rapid turnover at single CpGs. To distinguish between these models, we sorted and expanded a total of 38 single TSCs from two parent lines and cultured them for four to five passages (Extended Data Fig. 2a). Genome-wide assessment of methylated CGIs using reduced representation bisulfite sequencing (RRBS, ref. [41]) showed that each clone re-acquired comparable high entropy levels (Fig. 1d and Extended Data Fig. 2b,c), confirming that CpG methylation patterns are continuously evolving within these TSC populations.

We then investigated DNA methylation heterogeneity across larger genomic spans through long-read nanopore sequencing. Our extended in-phase methylation measurements were consistent with our short-read WGBS data, but allowed us to examine the coordination and degree of epigenetic variation across multi-CGI territories, such as those typically found at Polycomb-regulated gene promoters (Fig. 1e and Extended Data Fig. 2d). With these data, we confirm that CGIs captured within the same read display a similar degree of disordered methylation. Moreover, CGIs captured in phase tended to show comparable methylation levels relative to the unphased average, indicating a degree of local coordination (ranging from 1 kb up to 50 kb; Fig. 1f and Extended Data Fig. 2e). Together, these observations support a model where population-wide intermediate methylation reflects the heterogeneous epigenetic status of individual alleles.

The bimodal DNA methylation landscape of somatic cells is established within the post-implantation epiblast and propagated throughout foetal gestation and life. To determine if the extra-embryonic epigenome was similarly stable over placental differentiation, we examined global and hyper CGI methylation in vivo from undifferentiated embryonic day (E)6.5 ExE as well as late gestational labyrinth and junctional zone tissue[34]. We found that intermediate methylation persists to term within both placental lineages (Extended Data Fig. 2f), highlighting the maintenance of this unusual landscape throughout the duration of foetal development.

### The TSC epigenome shows enhanced global H3K27me3

In TSCs, intermediate methylation is found across gene-poor PMDs as well as canonical Polycomb targets, which are generally retained within distinct nuclear compartments. PMDs are overall maintained as constitutive heterochromatin and found near the nuclear lamina, whereas Polycomb-regulated loci are enriched within the nuclear interior to support context-specific gene induction[42]. To determine if intermediate DNA methylation alters nuclear topology, we generated high-coverage Hi-C data to measure the 3D genome organization of TSCs. For comparison, we used mouse ESC data cultured in serum/leukaemia inhibitory factor (LIF) because these cells display embryonic DNA methylation patterns and stably maintain well-characterized epigenomic features[43–46]. Despite their divergent epigenomes, A (euchromatic) and B (heterochromatic) compartment organization is very similar between these two cell types, suggesting that the TSC epigenome may not reflect global changes to nuclear reorganization at these scales (Fig. 2a,b and Extended Data Fig. 3a–c)[47]. Similarly, Polycomb-regulated genes remained predominantly within A compartments, indicating that these regions retain topological features of euchromatin despite elevated DNA methylation levels (Fig. 2c–e).

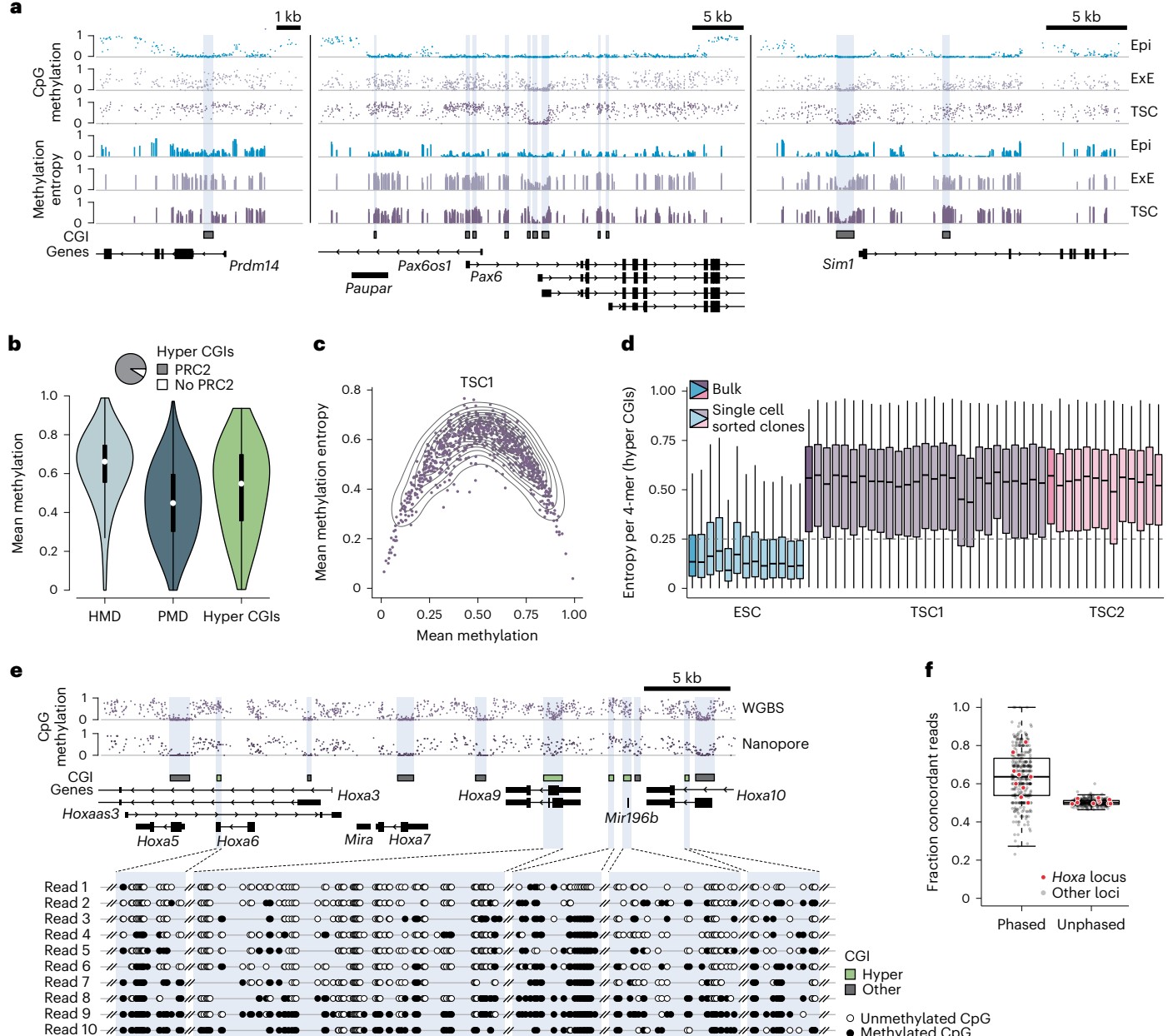

**Fig. 1 | Dynamic turnover of globally intermediate DNA methylation.**
**a**, Genome browser tracks of CpG methylation and methylation entropy for murine epiblast (Epi), ExE and TSCs. **b**, Methylation of HMDs (n = 959,249 1 kb tiles), PMDs (n = 954,783 1 kb tiles) and hyper CGIs (n = 1,102) in TSCs (single biological replicate). Pie chart shows the fraction of hyper CGIs targeted by PRC2 in ESCs. White dots denote the median, edges the interquartile range (IQR) and whiskers either 1.5× IQR or minima/maxima (if no point exceeded 1.5× IQR; minima/maxima are indicated by the violin plot range). **c**, Scatter plot comparing mean methylation entropy and mean CpG methylation at hyper CGIs in TSCs. **d**, Box plots of methylation entropy per 4-mer (n = 21,952) in hyper CGIs for individual subclones (RRBS data). Each subclone reaches similar entropy levels (low for ESCs, high for TSCs) to in silico generated bulk data. Lines denote the median, edges the IQR and whiskers either 1.5× IQR or minima/maxima (if no point exceeded 1.5× IQR; outliers were omitted). **e**, Top: genome browser track of the *Hoxa* locus comparing WGBS and long-read data. Bottom: single reads (98 kb average read length) all display intrinsically heterogeneous methylation. Missing CpGs within reads reflect low likelihood of the methylation call (Methods). **f**, Fraction of concordant reads that span two hyper CGIs (n = 383 CGI pairs, ≥10× coverage). A read is termed 'concordant' if CGI pairs are both above or below the median of their unphased values. Coordination between CGI pairs is apparent compared to randomly shuffled, unphased averages. *Hoxa* locus pairs are marked in red. Lines denote the median, edges denote the IQR, whiskers denote 1.5× IQR and minima/maxima are represented by dots.

We next examined the genomic distribution of chromatin modifications that have predictable relationships to DNA methylation in somatic contexts. We pursued a quantitative chromatin immunoprecipitation followed by sequencing (ChIP–seq) method (multiplexed indexed unique molecule T7 amplification end-to-end (MINUTE)-ChIP, ref. 48) that allowed us to directly compare the genomic distribution

and levels of different modifications in TSCs alongside ESC control samples. We prioritized the histone modifications H3K4me3, H3K27me3 and H2AK119ub1, which regulate unmethylated developmental gene promoters in embryonic cells[49–51]. Surprisingly, H3K27me3 remained enriched at methylated CGIs, with higher levels than observed for ESCs (Fig. 2f,g and Extended Data Fig. 3g). Moreover, TSCs show higher

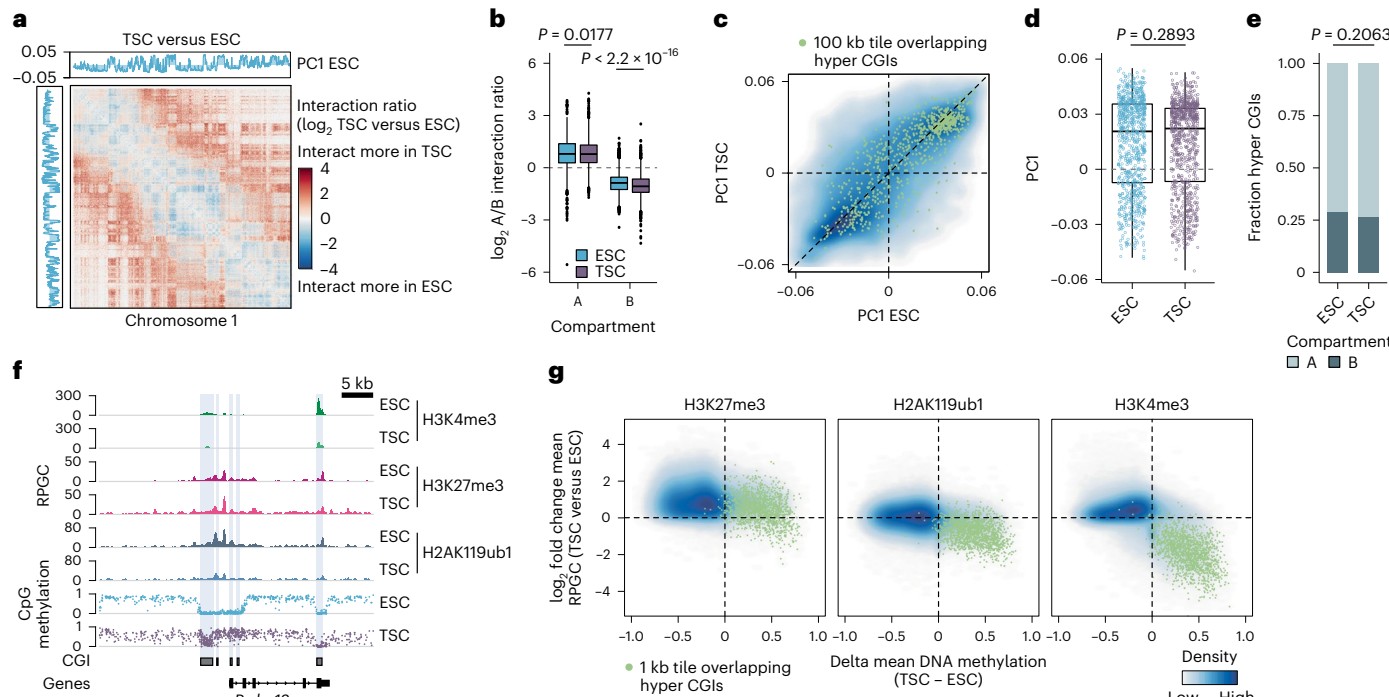

**Fig. 2 | Global increase of H3K27me3 in TSCs compared to ESCs. a**, Log₂ fold change of normalized Hi-C contact frequencies in TSCs compared with ESCs (two merged technical replicates per cell type; this also applies to **b–e**) on chromosome 1 (100 kb bins). Top and left: first principal component illustrating ESC compartments (A, positive values; B, negative values). **b**, Box plots of Hi-C A/B compartment interaction ratios per 100 kb bin ($n$ = 23,482 bins, see Methods). The A/B interaction ratio differs significantly between ESCs and TSCs for B compartments (two-sided Wilcoxon rank-sum test, $P$ = 0.0177 and $P$ < 2.2 × 10⁻¹⁶ for A and B compartments, respectively). However, the overall effect is minimal. Lines denote the median, edges denote the IQR, whiskers denote 1.5× IQR and minima/maxima are represented by dots. **c**, Density plot comparing PC1 across 100 kb tiles ($n$ = 24,026). Green dots mark tiles overlapping hyper CGIs ($n$ = 833). **d**, PC1 values for tiles overlapping hyper CGIs ($n$ = 833) do not significantly differ between ESCs and TSCs (two-sided Wilcoxon rank-sum test, $P$ = 0.2893). Lines denote the median, edges denote the IQR, whiskers

denote 1.5× IQR and minima/maxima are represented by dots. **e**, Fraction of hyper CGIs in A and B compartments do not significantly differ between ESCs and TSCs (two-sided chi-squared test, $P$ = 0.2063). **f**, Genome browser tracks of the *Prdm12* locus. In ESCs, unmethylated CGIs are enriched for H3K4me3 as well as for repressive H3K27me3 and H2AK119ub1. In TSCs, CGI methylation increases while H3K4me3 decreases. H3K27me3 spreads further into the flanking regions but remains enriched over CGIs. **g**, Density plots comparing DNA methylation (delta) and histone modifications (log₂ fold change) in TSCs compared with ESCs (1 kb tiles, $n$ = 3 merged biological replicates for each cell type). Globally, TSCs lose genome-wide methylation and gain H3K27me3. In contrast, tiles overlapping hyper CGIs show further H3K27me3 enrichment. Although TSCs tend to subtly increase global H3K4me3 signal, hyper CGIs demonstrate a clear loss. The global enrichment for H3K4me3 appears to correspond to differential retrotransposon regulation (see Extended Data Fig. 3g).

global enrichment for H3K27me3 across the genome as a whole, suggesting redeployment of PRC2 across the majority of intermediately methylated sequences. We also confirmed this global H3K27me3 elevation in TSCs via western blot as well as mass spectrometry (MS) for histone modifications (Extended Data Fig. 3e,f).

Combined with our MINUTE-ChIP data, our results point to a broad redistribution of PRC2 activity across the TSC epigenome (Fig. 2f,g and Extended Data Fig. 3d–g). In contrast, PRC1-mediated H2AK119ub1 levels remained largely stable between both cell types and H3K4me3 enrichment continued to be negatively correlated with DNA methylation, particularly at the CGI-enriched promoters of housekeeping genes (Fig. 2f,g and Extended Data Fig. 3d,e,g). In keeping with this rule, H3K4me3 was generally depleted from methylated CGIs, despite their frequent localization within developmental gene promoter regions (Fig. 2g and Extended Data Fig. 3g). We also found that TSCs exhibited enriched intergenic H3K4me3 signal, particularly within Intracisternal A-type particle (IAP)-family endogenous retroviruses that may have lineage-specific activity (Extended Data Fig. 3d,g)[52,53].

To evaluate if the simultaneous global enrichment of H3K27me3 and DNA methylation across the TSC epigenome represents the presence of dually modified chromatin, we performed a serial H3K27me3 ChIP followed by bisulfite sequencing (ChIP–BS-seq, refs. 13,54). In addition, we performed ChIP–seq for the essential PRC2 component

embryonic ectoderm development (EED) to confirm its continued genomic occupancy within TSCs, including at intermediately methylated developmental gene promoters. As for other assays, we included ESCs as the embryonic reference. Despite the non-quantitative nature of these ChIP–seq experiments, which probably led to diminished signal-to-noise ratios in TSCs, H3K27me3 ChIP-BS-seq and EED ChIP–seq data were highly concordant (Fig. 3a,b). Moreover, the underlying methylation status of H3K27me3 enriched DNA was almost identical to our WGBS data, indicating that these two modifications co-exist at a near equilibrium (Fig. 3 and Extended Data Fig. 4a–c). In contrast, ESCs maintain the canonical mutually exclusive relationship, particularly at CGIs, with high H3K27me3 enrichment corresponding to low DNA methylation levels (Fig. 3). Collectively, our investigation of the TSC epigenome finds that intermediate DNA methylation persists alongside a global redistribution of PRC2-deposited H3K27me3, a non-canonical relationship that includes a distinct form of regulation at developmental gene promoters.

**DNMT3B and Polycomb act as positive and negative regulators**
In embryonic and adult cells, local DNA methylation turnover is generally mediated by de novo DNMT and ten eleven translocation (TET) enzymes, which oxidize 5-methylcytosine (5-mC) to hydroxymethylcytosine (5-hmC) and other products[55]. In contrast, PRC2 occupies

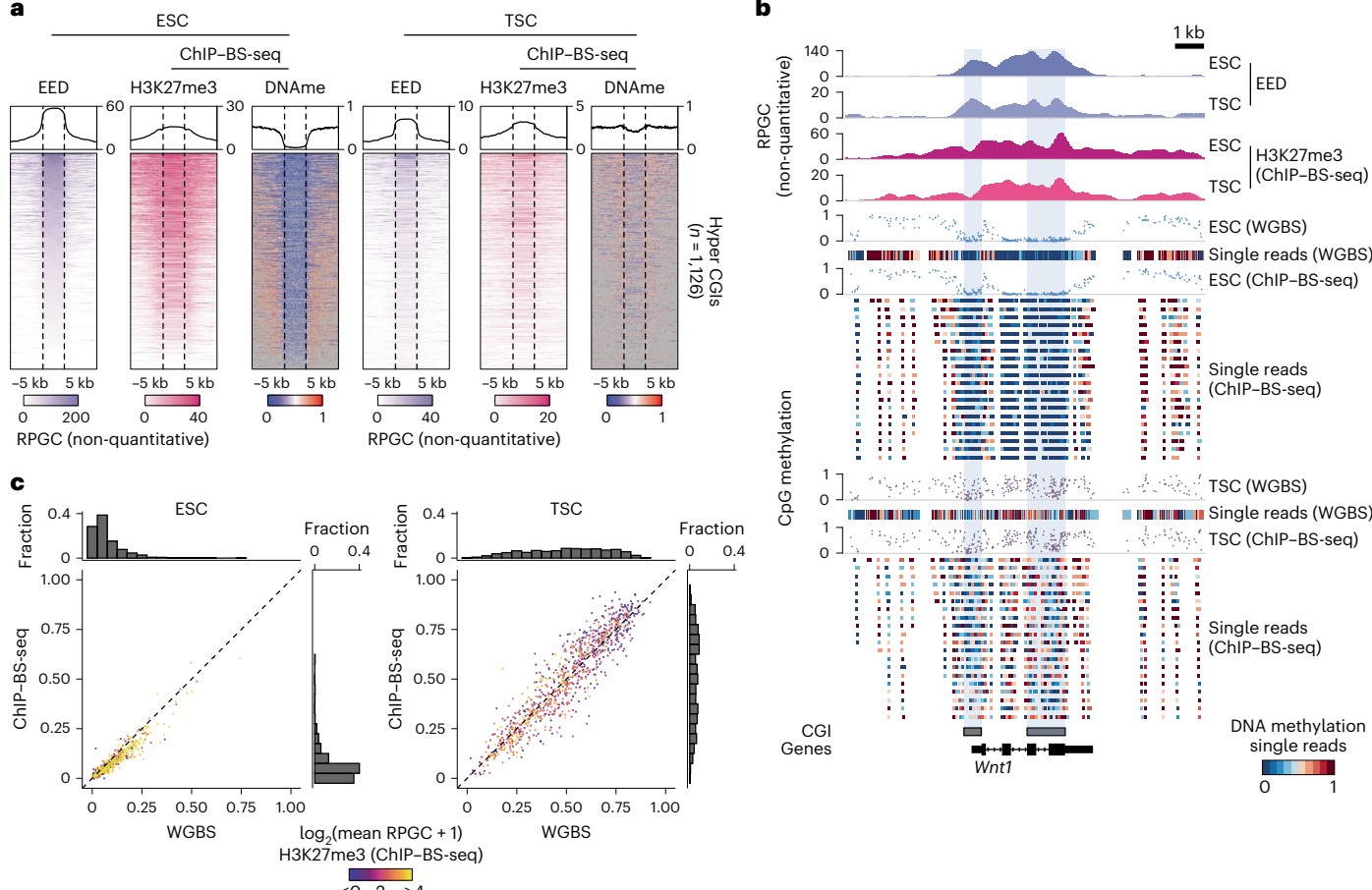

**Fig. 3 | TSC chromatin is dually modified by H3K27me3 and DNA methylation at equilibrium. a**, Metaplots and corresponding per-locus heat maps of EED (ChIP–seq), H3K27me3 (ChIP–BS-seq) and DNA methylation (ChIP–BS-seq) for hyper CGIs. ESCs display the expected inverse correlation between DNA methylation and H3K27me3, which is consistent with local enrichment of EED over these CGIs. TSCs also show local enrichment of EED over CGIs, but the canonical relationship between H3K27me3 and DNA methylation is lost and these modifications co-occupy the same loci. TSC ChIP signal is somewhat diminished in comparison with ESCs, probably due to increased global enrichment for this enzyme and its associated modification throughout the TSC genome. **b**, Genome browser track of the *Wnt1* locus in ESCs and TSCs for EED and H3K27me3 (as

measured by ChIP–BS-seq) enrichment alongside DNA methylation as measured by WGBS and ChIP–BS-seq. Average read-level methylation is expanded for ChIP–BS-seq data below the summary track (only the first 20 rows are shown, reads must have three or more CpGs to be included). Read-level analysis confirms that the diffuse, high entropy nature of DNA methylation in TSCs occurs within H3K27me3-modified nucleosomes. **c**, Scatter plot comparing the average methylation level of hyper CGIs as measured by WGBS and ChIP–BS-seq, coloured by the average H3K27me3 ChIP–BS-seq signal. WGBS includes no enrichment step and acts effectively as background; its high correlation with ChIP–BS-seq supports a model where intermediate DNA methylation in TSCs co-exists with H3K27me3 nucleosomes at equilibrium.

---

unmethylated CGIs within developmental gene promoters and cooperates with PRC1 (ref. 56). To better understand how these complexes might support intermediate methylation patterns in TSCs, we genetically ablated *Dnmt3b*, *Tet3*, *Eed*, *Rnf2* and *Kdm2b* (Fig. 4a and Extended Data Figs. 4d and 5a). We selected these targets for the following reasons: DNMT3B has the highest de novo activity in native ExE[23]; TET3 is the most highly expressed family member in TSCs, and prior descriptions of TET1 knockout (KO) TSCs did not report notable global DNA methylation changes[57]; EED is an essential core component of PRC2 and required for CGI hypermethylation during ExE differentiation;[23] RNF2 is a core subunit of PRC1 and involved in PRC2 recruitment as well as target regulation[49–51,58,59]; KDM2B is part of the PRC1.1 subcomplex and has been shown to block CGI methylation in embryonic cells[60].

We began exploring the effects of these knockouts on the steady-state maintenance of extra-embryonic methylation by generating WGBS data for each line. *Dnmt3b* ablation leads to a sharp genome-wide decrease in CpG methylation, a surprising shift given the ongoing presence of DNMT1 (Fig. 4b–d and Extended Data Fig. 5b). Loss of DNMT3B or even 3A/3B in other proliferating cells has a more

limited impact on global levels, including in ESCs[2,61–63]. In comparison, *Tet3* disrupted TSCs exhibited minimal changes that do not support a global role for enzymatic conversion of 5-methylcytosine in maintaining intermediate methylation (Fig. 4b,c and Extended Data Fig. 5b,c). We confirmed these results with quantitative MS for nucleotide modifications. Overall, 5-hmC levels are lower in TSCs compared with ESCs, even when accounting for their lower 5-mC levels (Extended Data Fig. 5d). Nonetheless, 5-hmC levels drop substantially in *Tet3* KO TSCs without dramatically changing 5-mC levels (Extended Data Fig. 5d). Together, these data highlight TSCs' unusual and ongoing reliance on de novo methylation to counteract a persistent, TET-independent drive towards global hypomethylation.

The apparent maintenance of globally intermediate CpG methylation would therefore require other complexes to act as negative regulators. To our surprise, core PRC component (EED or RNF2) KOs exhibited strong genome-wide DNA methylation gains. In particular, our PRC2 KO showed the most dramatic increase in DNA methylation, including thousands of CGIs that were previously unmethylated in wild-type (WT) TSCs (Fig. 4b–d and Extended Data Figs. 5 and 6). We also confirmed

that genome-wide hypermethylation within PRC2 KO cells corresponds to global loss of H3K27me3, but not to loss of H2AK119ub1, via western blot and MINUTE-ChIP (Fig. 4e,g and Extended Data Figs. 5f–h and 6a–c). In comparison, *Rnf2* KO (our core PRC1 KO) cells show a milder decrease in H3K27me3 as well as an expected reduction of H2AK119ub1 (Fig. 4e,g and Extended Data Figs. 5f–h and 6a–c), similar to prior studies on the interplay between these two modifications[42,56].

Finally, *Kdm2b* KO cells showed the same global trend as for other PRC KOs, but affected fewer CGIs (*n* = 3,276 of 3,967 CGIs that show increasing methylation across our PRC KOs are unaffected by *Kdm2b* disruption). H3K4me3 structurally antagonizes DNA methylation[64] and remains enriched within unmethylated CGIs in WT TSCs (Extended Data Fig. 3g). In *Eed* KO TSCs, de novo hyper CGIs lose WT H3K4me3 levels in rough proportion to DNA methylation gains (Extended Data Fig. 5g). In contrast, *Kdm2b* KO cells preserve H3K4me3 at these regions, explaining the diminished effect on DNA methylation (Fig. 4f,g and Extended Data Figs. 5g and 6b–e). KDM2B has been reported to have H3K4me3 in addition to H3K36 demethylase activity[65,66]. As such, its enzymatic function may be required to epigenetically reprogram these loci towards a hypermethylated state.

Notably, the effects of PRC disruption on CGI and global methylation levels differ from current in vivo observations, where zygotic mutants fail to accumulate DNA methylation at developmental gene promoters within the ExE[23,67]. Deeper investigation into this discrepancy indicates that PRC2-based maintenance of H3K27me3 within the early embryo may provide a necessary template to ensure initial de novo methylation, as the CGIs of zygotic *Eed*- or *Rnf2*-null embryos remain unmethylated, but the surrounding areas (CGI 'shores', ref. 68) become methylated (Extended Data Fig. 7a,b). In vivo, this epigenetic signature of PRC disruption is present within both the E6.5 extra-embryonic ExE as well as the embryonic epiblast, strongly indicating a requirement for PRC2 to consolidate divergent epigenetic landscapes during these early differentiation events (Extended Data Fig. 7a,b). As zygotic KO embryos are not viable beyond the earliest stages of embryonic and placental development[67,69], further work with lineage-specific perturbations will be necessary to establish the post-differentiation roles of PRCs in vivo.

## The TSC epigenome is highly elastic

Collectively, DNMT3B and Polycomb appear to be central epigenetic players that maintain globally intermediate DNA methylation levels for extended time in culture. To characterize the stability of this landscape, as well as the kinetics between methylation gain and loss, we treated TSCs with a DNMT1-specific inhibitor (GSK3484862, DNMT1i) (ref. 70) for 1 week, followed by a 2 week recovery period (Fig. 5a and Extended Data Fig. 7c). We evaluated multiple TSC lines and passage numbers, all of which displayed rapid and substantial genome-wide loss of DNA methylation, with equally rapid recovery after compound withdrawal (Fig. 5a and Extended Data Fig. 7c–g). TSC viability was not

compromised by global loss of DNA methylation, which is otherwise a unique feature of naïve or ICM-stage ESCs[71,72]. Furthermore, the ability to restore intermediate methylation levels after erasure strongly indicates the presence of additional regulatory encoding.

To test the equivalent dynamics of H3K27me3, we utilized the EZH2-specific inhibitor Tazemetostat (EPZ6438, EZH2i, ref. 73). These experiments confirm our genetic disruptions and independently show that PRC2 inhibition drives DNA methylation upwards. As with our DNMT1i treatments, the effects of PRC2 inhibition are reversible: hyper CGI methylation steadily increased from a median of 53% to 85% within 5 weeks of treatment and decreased at a similar rate upon withdrawal (Fig. 5b and Extended Data Fig. 8a). The comparatively slower recovery after EZH2i withdrawal suggested ongoing de novo methyltransferase activity even within these abnormally high methylation regimes. To address this hypothesis, we pulse-treated EZH2i cells with DNMT1i for 1 week and found that cells quickly restabilized intermediate DNA methylation levels upon inhibitor withdrawal (Fig. 5b and Extended Data Fig. 8a). We also confirmed that EZH2i-treated TSCs restore H3K27me3 enrichment, again highlighting the robust feedback between these two regulators despite their antagonistic relationship (Fig. 5c and Extended Data Fig. 8b). Combined, our different inhibitor treatments highlight an extraordinary degree of plasticity within the extra-embryonic epigenome, including the ability for these modifications to rise and fall without compromising their potential to return to steady-state levels.

Although the genome-wide effects on the TSC methylome are similar between our EZH2i treatments and EED KO, EPZ6438 is a competitive inhibitor for the universal methyl donor *S*-adenosylmethionine (SAM) and may have subtly different effects on the epigenetic status of TSC loci. We performed EED immunoprecipitation (IP) followed by western blotting as well as EED ChIP–seq on EZH2i-treated TSCs to examine PRC2 complex stability and genomic occupancy. Compared with untreated TSCs, PRC2 protein expression is subtly lower, reflecting some degree of destabilization and degradation (Extended Data Fig. 8c). Similarly, EED binding to CGIs does diminish with inhibitor treatment, but mainly for CGIs with lower initial enrichment: ~57% (3,423 of 5,998) of untreated peaks are not called after 5 weeks of EZH2i treatment (Extended Data Fig. 8d,e). When we compare EZH2i-insensitive and EZH2i-sensitive EED peaks, we find that both gain DNA methylation in EZH2i-treated and *Eed* KO cells, but that EZH2i-insensitive peaks are generally more resistant (Extended Data Fig. 8e,f). The comparable resilience of these CGIs to de novo methylation is similar in both inhibitor-treated and KO TSCs, indicating that they are intrinsically protected from DNA hypermethylation even without PRC2 present (Extended Data Fig. 8f).

Finally, we investigated the biochemical nature of this interaction by performing co-IP experiments for the PRC2 subunit EED, which has been extensively characterized in various stages of mouse pluripotency[74,75]. By both western blot and MS analysis, we find a clear enrichment for DNMT3B within our EED IPs (Fig. 5d,e and Supplementary

**Fig. 4 | Intermediate methylation in TSCs depends on opposing DNMT3B and Polycomb activity. a**, Epigenetic repression in embryonic cells. DNMT3A/B deposits DNA methylation whereas TET enzymes promote its removal. PRC1 and 2 shield developmental gene promoters and recruit each other through their respective modifications. **b**, Feature-level DNA methylation (1 kb HMDs/PMD tiles, CGIs and hyper CGIs, *n* = 904,532, 853,972, 14,790 and 1,030, respectively, single biological replicate per condition). Lines denote the median, edges the IQR and whiskers either 1.5× IQR or minima/maxima (if no point exceeds 1.5× IQR); minima/maxima are indicated by the violin plot range. **c**, Genome browser tracks of the *Tbx2* locus. *Dnmt3b* KO loses methylation, while PRC KOs gain methylation up to 100%. Regions marked by strong H3K4me3 signal are kept constitutively free while regions with low H3K4me3 remain unmethylated in *Kdm2b* KO. **d**, CpG-wise comparison of WT and KO TSCs (single biological replicate per condition). Barplots indicate the fraction of CpGs that change by >|0.2| compared with WT. **e**, Density plots comparing DNA methylation (delta)

and H3K27me3 (log₂ fold change) at 1 kb tile resolution between KO and WT TSCs (*n* = 3 merged biological replicates for MINUTE-ChIP data, single biological replicate for WGBS, also applies to **f** and **g**). *Eed* KO loses H3K27me3 accompanied by strong DNA methylation gains. **f**, Scatter plot comparing PRC hyper CGIs (*n* = 3,849) in *Eed* KO and *Kdm2b* KO with respect to WT. Points are coloured by H3K4me3 level in *Kdm2b* KO (log₂-transformed). PRC2 hyper CGIs with high H3K4me3 levels in *Kdm2b* KO remain unmethylated but gain methylation in *Eed* KO. **g**, Metaplots showing the average histone modification enrichment and DNA methylation for WT and KO TSCs at PRC hyper CGIs (respective heat maps are shown in Extended Data Fig. 6c). MINUTE-ChIP enrichment can be quantitatively compared within the same batch (*Dnmt3b* KO and PRC KOs have separate WT controls, see Methods). *Dnmt3b* KO exhibits mild H3K27me3 and H2AK119ub1 gain, while both H3K27me3 and H2AK119ub1 are reduced in *Rnf2* KO. *Eed* KO loses all H3K27me3 signal. Enrichment scales are distinct for H3K4me3 (green, left axis) and H3K27me3 or H2AK119ub1 (black, right axis).

Table 15). Notably, we do not enrich for PRC1 subunits, which supports our genetic finding that the DNMT–PRC2 axis dominates the antagonistic epigenetic relationship that regulates intermediate methylation in TSCs. Although IP–western for EED in WT ESCs also recovered DNMT3B, IP–MS against an IgG control did not confirm this enrichment with our statistical cut-off (Extended Data Fig. 8g and Supplementary Table

16). More generally, we find that the PRC2 interactome does appear to differ between TSCs and ESCs. IPs from both cell types recover the majority of direct subcomplex components (such as EZH2, JARID2 and MTF2), but the ESC interactome also includes proteins with functions in pre-mRNA binding and processing that have been previously shown to support early lineage priming (Extended Data Fig. 8h,i, refs. 76–79).

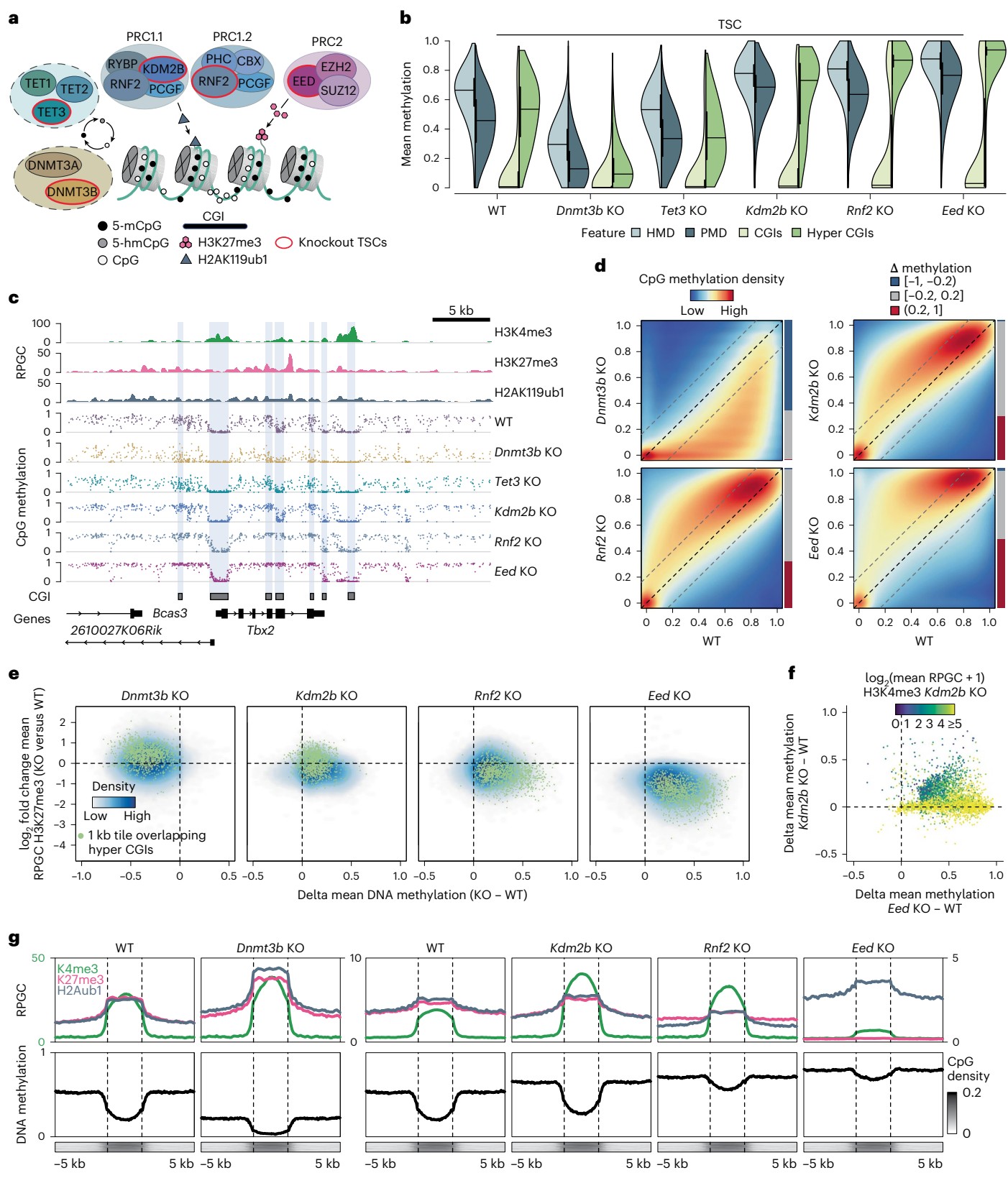

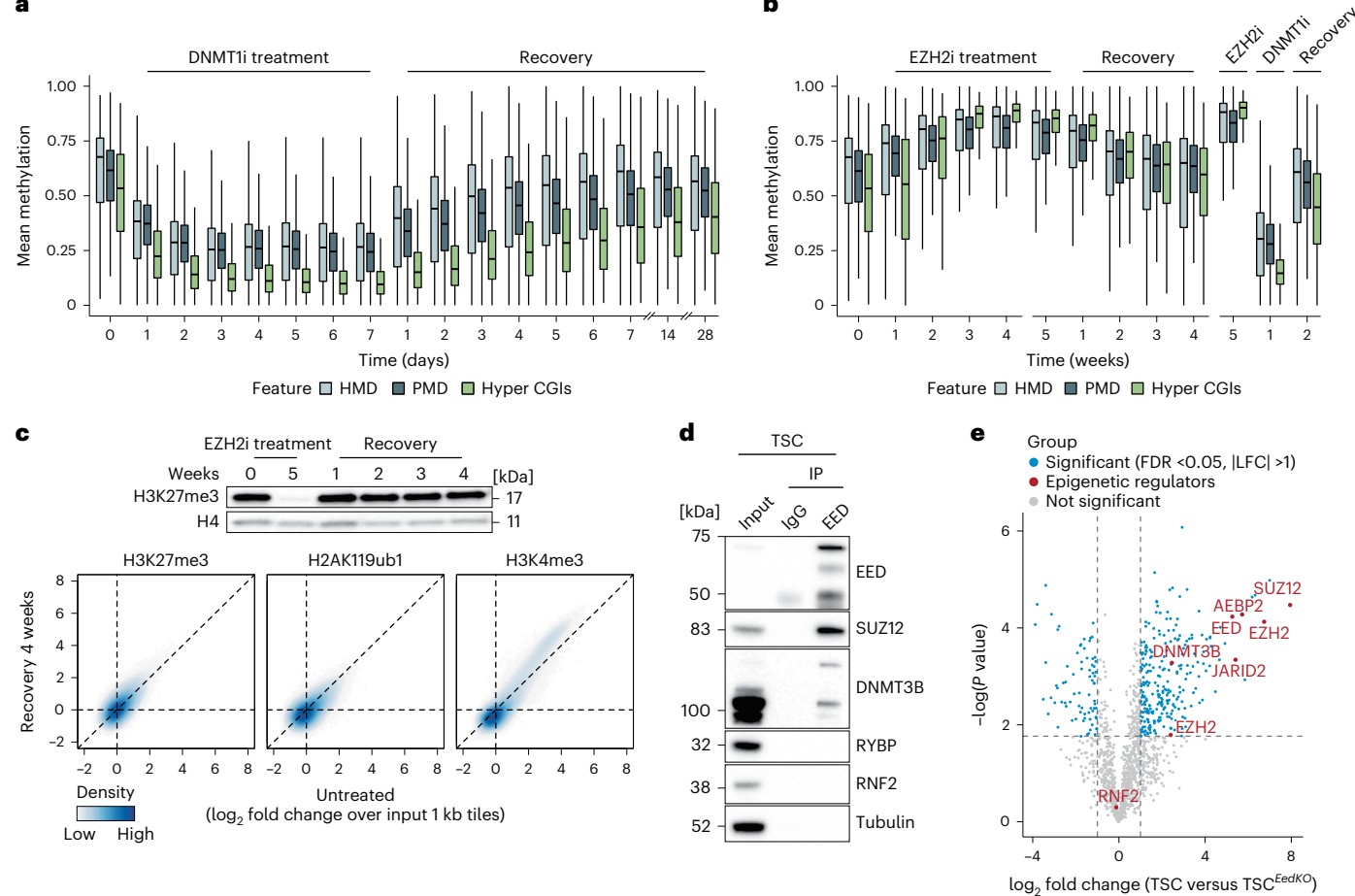

**Fig. 5 | The TSC epigenome can be reversibly driven to extreme DNA methylation levels. a**, DNMT1i treatment and recovery as measured by RRBS (untreated control measured by WGBS, single biological replicates). Genome-wide methylation drops drastically during the first 3 days and recovers most of the original methylation within one week of withdrawal ($n$ = 117,477 and 62,118 1-kb tiles in HMDs and PMDs, respectively). Hyper CGIs ($n$ = 970) show similar trends, although re-methylation efficiency is slightly lower. Lines denote the median, edges denote the IQR and whiskers denote either 1.5× IQR or minima/maxima (if no point exceeded 1.5× IQR; outliers were omitted). **b**, EZH2 inhibitor (EZH2i) treatment and recovery time series as measured by RRBS (untreated control measured by WGBS, single biological replicates). Left: genome-wide and CGI methylation rise to extremely high levels after 4 weeks ($n$ = 116,056 and 62,434 1-kb tiles in HMDs and PMDs, respectively, and $n$ = 960 hyper CGIs). The effect is progressively reversed following a 4 week washout period. Right: independent experiment that demonstrates accelerated recovery of steady-

state methylation levels by pulse DNMT1i treatment. Lines denote the median, edges denote the IQR and whiskers denote either 1.5× IQR or minima/maxima (if no point exceeded 1.5× IQR; outliers were omitted). **c**, Top: western blot for H3K27me3 in untreated TSCs, TSCs treated with EZH2i for 5 weeks and weekly recovery timepoints. Bottom: MINUTE-ChIP correlation between untreated and post-recovery TSCs, measured in 1 kb tiles (log₂ fold change over input) demonstrate the reversibility of the TSC epigenome. **d**, Co-IP of EED and core components of PRC1/2, DNMT3B and Tubulin (negative control) in WT TSCs. EED directly interacts with other components of PRC2 as well as DNMT3B, but not with components of PRC1. **e**, Enrichment and statistical significance of EED interactions within TSCs as measured by MS following IP (WT TSCs were compared with *Eed* KO to eliminate noise, two-sided Student's *t*-test, *P* values adjusted for multiple testing correction using FDR). EZH2 is plotted twice because of the recovery of two distinguishable isoforms, Q61188;D3Z774 and Q6AXH7, respectively.

In contrast, the TSC interactome is substantially more enriched for proteins with broader nuclear functions, including nuclear matrix proteins, components of the nuclear pore, and nucleolar RNA processing factors (Extended Data Fig. 8h,i and Supplementary Tables 15 and 16). Although the functional meaning of these interactions remains to be determined, our biochemical findings are consistent with a more global interaction between PRC2 and DNMT3B that operates across the TSC epigenome as a whole.

**Polycomb and DNA methylation support TSC viability**

We next sought to connect the non-canonical form of global genome repression found in TSCs to biological function by examining gross morphological, proliferative and transcriptional responses of our inhibitor treatments. Notably, cells treated with either inhibitor exhibited

minimal morphological effects and continued to proliferate, but senesced and flattened when exposed to both simultaneously (Fig. 6a). By RNA sequencing (RNA-seq), loss of either repressive mechanism results in distinct and reversible transcriptional responses, but neither affected the regulation of genes proximal to hyper CGIs (Fig. 6b–d and Extended Data Fig. 9). Instead, loss of DNMT1 leads to upregulation of germline-associated genes, particularly those with methylated promoters in TSCs (Fig. 6b and Extended Data Fig. 9a). Notably, TSCs (and the placenta in general) share aspects of their gene regulatory network with the male germline[80–82]. Although we saw no transcriptional effect on shared gametogenesis-placental genes overall (Extended Data Fig. 10a), de-repression of genes with similar functions by DNMT1i may reflect a role for DNA methylation as a buffering mechanism during placental development.

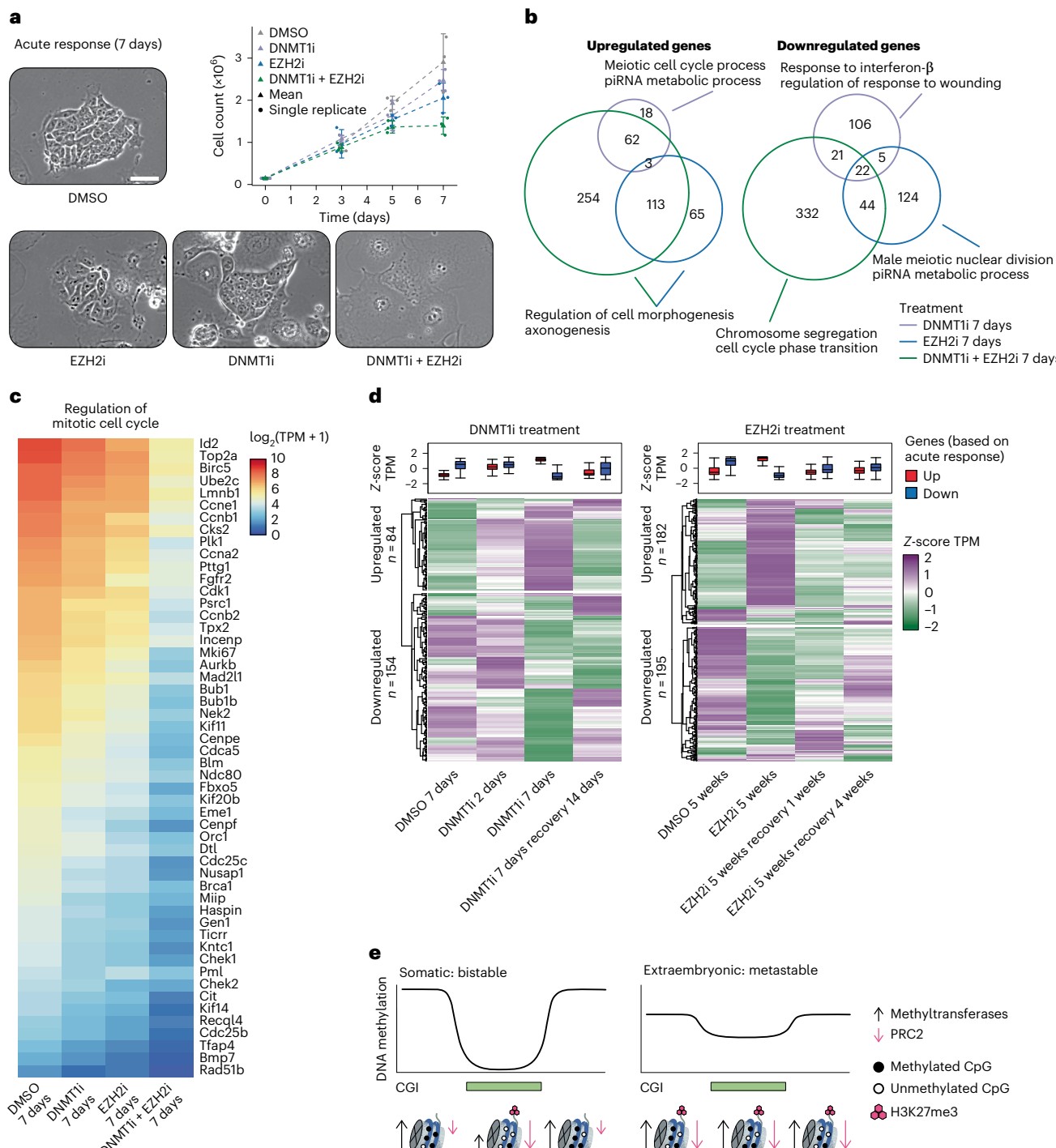

**Fig. 6 | Transcriptional response to epigenetic inhibitors. a**, Brightfield images of control and inhibitor treated TSCs. Top right: cell counts over 7 days of treatment (*n* = 3 biological replicates from independent experiments per condition, error bar reflects standard deviation). TSCs tolerate either DNMT1i or EZH2i, but dual inhibition has severe effects on morphology and proliferation (scale bar, 50 μm). **b**, Overlap of up- and downregulated genes between DNMT1i, EZH2i and combined treatment with select GO term enrichments for each gene set (piRNA, PIWI-interacting RNA). Notably, combined treatment significantly downregulates a large set of genes associated with cell cycle progression. A full list of top GO terms is presented in Extended Data Fig. 9a. **c**, Heat map visualizing gene expression (log₂-transformed TPM) associated with regulation of mitotic cell division in DMSO-, DNMT1i-, EZH2i- and double inhibitor-treated cells. Treatment with both inhibitors leads to significantly reduced expression of these genes (only differentially expressed genes are shown). The effect is milder in single inhibitor treatments. **d**, Heat map and box

plots of differentially expressed genes during DNMT1i (left) and long-term EZH2i (right) treatment including recovery timepoints (number of genes indicated in the figure, differentially expressed genes are identical to those in Fig. 5b). In both cases, the transcriptional response is largely reversible following inhibitor washout. Lines denote the median, edges denote the IQR and whiskers denote either 1.5× IQR or minima/maxima (if no point exceeded 1.5× IQR; outliers were omitted). **e**, Simplified model of DNA methylation and PRC2 dynamics in somatic cells compared with the dynamic epigenome found in TSCs. Somatic cells generally regulate genetic loci in a bistable fashion, preserving an overall highly methylated genome and unmethylated CGIs that are protected from DNMT3's by PRC2. In TSCs, the genome shifts to an overall intermediate, seemingly metastable methylation state, which co-occurs with PRC2-deposited H3K27me3. Although this state can be driven to high or low methylation levels by modulating these two inputs, this form of genome regulation is robust enough to return to the steady-state levels even after long spans of inhibition.

In contrast to the effects of DNMT1 inhibition, EZH2i affects genes associated with morphogenesis. The transcriptional responses of our EZH2i treatment are also observed in *Eed* KO TSCs, but neither is strikingly enriched for Polycomb targets compared with the genomic background and may be indirect (Extended Data Fig. 9b–d). Similarly, epigenetic disruption does not appear to spur substantial spontaneous differentiation, despite morphological changes that could otherwise be consistent with differentiation into trophoblast giant cells (TGCs). Curated lists of marker genes associated with multiple placental cell types and functions, including trophoblasts of the labyrinth and junctional zones as well as TGCs, showed minimal differentiation-associated changes[83–89]. We did find a subtle downregulation of progenitor-associated genes and low-level TGC marker gene expression, but this could be consistent with the accompanying stress of proliferation arrest and not a direct effect (Extended Data Fig. 10). More obviously, combined DNMT1i and EZH2i treatment has a drastic impact on core cellular functions, primarily concerted downregulation of genes associated with cell cycle maintenance, chromosome segregation and cell cycle progression (Fig. 6b,c and Extended Data Fig. 9a). This broad, considerable signal more clearly corresponds to the rapid morphological and proliferative changes induced by dual inhibition of PRC2 and DNA methylation, again supporting the convergence of these two pathways to support major genome-scale functions in TSCs.

## Discussion

We utilized TSCs as a model to investigate the placental epigenome, which is characterized by persistent intermediate methylation and differential regulation of canonical Polycomb targets. We find that this landscape is maintained through a dynamic, antagonistic relationship between two distinct epigenetic repressive pathways—DNA methylation and the PRCs—that typically regulate mutually exclusive genomic territories within the embryonic lineage. Within the TSC epigenome, these pathways appear to converge towards a stable equilibrium between positive and negative regulators (Fig. 6e). So far, dynamic DNA methylation turnover has been primarily described through opposing catalytic activity of DNMT and TET enzymes, but this largely operates locally, primarily resolves to favour either hypo- or hypermethylation, and conforms with core concepts of bistable genome regulation[63,90–92]. Although TSC methylation does not appear to rely on TET-based oxidation, there is emerging evidence that TETs have non-catalytic roles and interact with PRC2 in embryonic lineages[74,92,93]. Our results in extra-embryonic cells suggest a direct interaction of DNMT3B with PRC2 but less likely with PRC1. In line with this, we find that PRC1 subunit KOs display a more modest degree of hypermethylation, which may be explained by the incomplete depletion of H3K27me3. As such, the molecular and epigenetic relationships between PRC2, PRC1 and the TET enzymes within the context of the TSC epigenome warrants further investigation.

Compared with somatic rules of epigenetic regulation, the PRC2–DNMT relationship detailed here seems to direct loci towards a fundamentally disordered methylation state, operates across the majority of the genome, and maintains these features without distorting nuclear topology as measured by Hi-C. Notably, either regulatory input (DNA or H3K27 methylation) can be destabilized for long durations without compromising the genome's ability to return to a dually-modified molecular state. Our inhibitor experiments confirm robust restabilization upon inhibitor withdrawal and demonstrate this epigenome's highly elastic nature. Mechanistically, our results also point towards H3K27me3 as a potentially crucial signalling hub. It is also worth considering what distinguishes the long-term stability of intermediate methylation found within TSCs from common cell culture artefacts that emerge when immortalizing primary or cancer cells, including substantial loss of PMD methylation and extreme hypermethylation of CGIs[94–96]. Notably, similar extreme methylation changes only appear to happen in TSCs when either DNMT or PRC inputs are blocked. Taken together, our data demonstrate that the TSC epigenome operates according to a unique arrangement of ubiquitously utilized chromatin regulators. The structural basis of this continuous antagonism remains to be determined, as do the molecular boundaries after which feedback between H3K27 and DNA methylation break down. Similar experiments on other major pathways, such as those that interact through H3K36 methylation, may eventually allow for a complete molecular description of this landscape[97,98].

More generally, similar dramatic genome-wide shifts of methylation away from heterochromatic regions and towards CGIs are recognized as unifying features of diverse cancer types[99–101]. Despite extensive research over several decades, the underlying mechanisms for how CGI-containing promoters are first targeted for methylation and then maintained in an intermediate state are still unclear and challenging to model[102]. Similarly, the regulation and purpose of global hypomethylation in cancer is also unresolved[103,104]. The highly entropic methylation pattern that characterizes native cancers has similar features to what we have observed in TSCs, including exceptional stability that can propagate for decades or through extremely selective events such as chemotherapy with minimal change[105,106]. As a result, it is tempting to consider the possibility that some of the fundamental regulatory principles described here are shared with primary tumours. If indeed the case, it would shine new light on a major cancer hallmark and highlight relevant parallels between normal development and disease.

Finally, this extra-embryonic landscape still requires both developmental and evolutionary explanation. As described above, perturbation of either repressive pathway, alone or in combination, does not appear to cause notable fluctuation in the expression of embryonic genes with methylated promoters, and continued viability when either DNA or H3K27 methylation are depleted is rarely observed outside of naïve mouse ESCs[17–20,72]. Although extraordinarily valuable for biochemical and genetic characterization, TSCs are more limited for connecting placental genome regulation to physiological function. Future work that seeks to address these key points—what this form of genome regulation contributes to support foetal development and how it was evolutionarily innovated—will ultimately require more detailed investigations in vivo.

## Online content

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

¹Department of Genome Regulation, Max Planck Institute for Molecular Genetics, Berlin, Germany. ²Department of Medical Biotechnology, Technische Universität Berlin, Berlin, Germany. ³Department of Biology, Chemistry and Pharmacy, Freie Universität Berlin, Berlin, Germany. ⁴Otto Warburg Laboratories, Max Planck Institute for Molecular Genetics, Berlin, Germany. ⁵Science for Life Laboratory, Department of Medical Biochemistry and Biophysics, Karolinska Institutet, Stockholm, Sweden. ⁶Ming Wai Lau Centre for Reparative Medicine, Stockholm node, Karolinska Institutet, Stockholm, Sweden. ⁷Epigenetics and Sex Development Group, Berlin Institute for Medical Systems Biology, Max-Delbrück Center for Molecular Medicine, Berlin-Buch, Germany. ⁸Universitätsklinikum Schleswig-Holstein Campus Kiel, Zentrum für Integrative Psychiatrie gGmbH, Kiel, Germany. ⁹Mass Spectrometry Joint Facilities Scientific Service, Max Planck Institute for Molecular Genetics, Berlin, Germany. ¹⁰Department of Genetics, Yale Stem Cell Center, Yale School of Medicine, New Haven, CT, USA. ¹¹Broad Institute of MIT and Harvard, Cambridge, MA, USA. ¹²Department of Stem Cell and Regenerative Biology, Harvard University, Cambridge, MA, US. ¹³These authors contributed equally: Raha Weigert, Sara Hetzel. ✉e-mail: z.smith@yale.edu; meissner@molgen.mpg.de

## Methods

All research described here complies with the relevant ethical regulations at the respective institutions. Work at the Max Planck Institute was approved by the Landesamt für Gesundheit und Soziales.

### Derivation of TSCs

TSCs were derived from CD-1 strain blastocysts. TSCs were derived as previously described[28] with few modifications. Briefly, blastocysts were washed in five serial drops of TSC medium (RPMI + GlutaMAX (Thermo Fisher Scientific, #61870044), 20% foetal bovine serum (PAN, #P30-2602), 1 mM sodium pyruvate (Thermo Fisher Scientific, #11360070), 100 µM 2-mercaptoethanol (Thermo Fisher Scientific, #21985023) and 1× penicillin–streptomycin (Thermo Fisher Scientific, #15140122); 25 ng ml$^{-1}$ FGF4 (R&D systems, #235-F4-025) and 1 µg ml$^{-1}$ heparin (Sigma, #H3149) were added to the medium fresh before each use) and seeded onto single wells of a 24-well culture dish with irradiated CD-1 primary mouse embryonic fibroblasts (MEFs). Medium was changed after the blastocysts attached onto the wells (~3 days). The outgrowths were disaggregated with trypsin–EDTA 0.05% (Thermo Fisher Scientific, #25300054) and seeded onto the same wells with MEF-conditioned TSC medium (70% MEF-conditioned medium, 30% TSC medium, +FGF4 (37.5 ng ml$^{-1}$), heparin (1.5 µg ml$^{-1}$)) and 1× ROCK inhibitor Y-27632 (Tocris, #1254) (ROCKi). After 4–6 days, TSC colonies that appear are allowed to grow to a confluency of 70–80%. TSC colonies are disaggregated and expanded onto a six-well dish with TSC medium.

This manuscript contains data generated from four TSC WT lines. TSC1, TSC3 and TSC4 were derived for this study. TSC1 and TSC3 are male, and TSC4 is female. TSC2 is a female line and kind gift of the Zernicka-Goetz lab, Cambridge, UK. The TSC3 line was found to contain a homozygous 2,294 bp deletion reaching from intron 5 to the centre of exon 6 in the *Dnmt3b* gene, which does not appear to affect its methyltransferase function or global DNA methylation levels. Extended Data Fig. 1 confirms the similarity between all four TSC line methylation states.

The *Dnmt3b* KO was established in the WT line TSC2. The *Rnf2* KO, *Kdm2b* KO and *Tet3* KO were generated in the WT line TSC1. The *Eed* KO was generated in the WT line TSC3, all of which had highly similar global methylation levels (Extended Data Fig. 1). For determining the sex of the derived lines, MEF-depleted TSCs were expanded on plastic dishes with MEF-conditioned medium (+FGF4 37.5 ng ml$^{-1}$ and heparin 1.5 µg ml$^{-1}$) and genotyped by a simplex PCR for the *Rbm31* locus (Supplementary Table 21).

### TSC culture and genetic manipulation

TSCs were cultured in standard conditions as described before[28]. Briefly, cells were cultured on MEFs in TSC medium (see above). Splitting was carried out every 5–7 days by rinsing the cells once with Dulbecco's phosphate-buffered saline (DPBS; Thermo Fisher Scientific, #14190144) before detaching the cells using trypsin–EDTA (0.05%). TSCs were passaged in clumps.

Before sample collection, TSCs were passaged at least one passage without MEFs to dilute out feeder cells. During this time, cells were cultured in MEF-conditioned medium (see above). Cell pellets were washed twice with DPBS before snap freezing at −80 °C. The DNA was extracted using the PureLink Genomic DNA Mini Kit (Thermo Fisher Scientific, #K182002) according to the manufacturer's instructions. Genetic perturbations were performed using the sgRNA/Cas9 system. For KO experiments TSCs were transfected with two PX458 plasmids (Addgene, #48138) each containing one single guide RNA that together delete the locus of interest (*Dnmt3B, Eed, Rnf2, Kdm2b* and *Tet3*) by non-homologous end joining. TSCs were transfected using FuGENE HD Transfection Reagent (Promega, #E2311) or P3 Primary Cell 4D-Nucleofector X Kit (Lonza, V4XP-3024). FuGENE: 300,000 cells were plated the day before transfection (feeder free) in MEF-conditioned medium (+FGF4 37.5 ng ml$^{-1}$ and heparin 1.5 µg ml$^{-1}$). On the day of transfection,

8 µg of plasmid DNA was diluted in 125 µl Opti-MEM (Thermo Fisher Scientific, #31985062). Twenty-five microlitres of FuGENE reagent (room temperature) was diluted with 100 µl Opti-MEM. Diluted FuGENE was added to diluted DNA, incubated at room temperature for 15 min and added to the cells dropwise. Medium was changed on the next day. Nucleofection: 1 M cells were washed once with PBS and resuspended in a transfection volume of 100 µl (consisting of 82 µl P3 Primary Cell Nucleofector Solution and 18 µl Supplement 1) containing 5 µg of DNA (PX458, see above). Cells were transferred to a Nucleocuvette and transfected in a 4D-Nucleofector System using the pulse code DA113. Cells were seeded back in MEF-conditioned medium (+FGF4 37.5 ng ml$^{-1}$, heparin 1.5 µg ml$^{-1}$ and 1× ROCKi).

GFP-positive cells were sorted 48–72 h post transfection using the BD FACSAria Fusion instrument and plated on feeder cells in standard TSC medium containing ROCKi. KOs were verified by genotyping (Supplementary Table 21) and western blot.

### Single-cell-sorted clones

TSC WT cells (TSC1 and TSC2) were sorted as single cells onto MEF-coated 96-well plates containing TSC medium +FGF4 (37.5 ng ml$^{-1}$), Heparin (1.5 µg ml$^{-1}$) and 1× ROCKi using the BD FACSAria Fusion instrument.

ESC WT cells were sorted as single cells onto gelatin and MEF-coated 96-well plates containing ESC medium (Knockout DMEM (Thermo Fisher Scientific, #10829018, 15% foetal bovine serum (PAN, #P30-2602), 1× GlutaMAX supplement (Thermo Fisher Scientific, #35050-038), 1× non-essential amino acids (Thermo Fisher Scientific, #11140-035), 100 µM 2-mercaptoethanol (Thermo Fisher Scientific, #21985023), 1× penicillin–streptomycin (Thermo Fisher Scientific, #15140122) and lab-purified recombinat leukaemia inhibitory factor (LIF) using the BD FACSAria Fusion instrument. Cells were then expanded and MEF-depleted for methylation analysis by RRBS.

### DNMT1i treatment

TSCs were cultured on MEFs in TSC medium containing DNMT1i (GSK-3484862; dissolved in DMSO to 1 mM) at a final concentration of 1 µM or equal volume of DMSO (Sigma, D2650) only, for up to 7 days, with medium changed daily. Before sample collection, TSCs were passaged at least once without MEFs in MEF-conditioned medium (+DNMT1i/DMSO) to dilute out feeder cells. For collection after 2 days of DNMT1i, cells were MEF depleted just before starting the treatment to avoid MEF contamination at the time of collection. Subsequently, the inhibitor was removed by splitting the cells, and the cells were cultivated for up to 4 weeks with standard conditions for recovery.

### EZH2i treatment

TSCs were cultured on MEFs in TSC medium containing EZH2 inhibitor (Tazemetostat/EPZ6438, Biovision, #2383-5, dissolved in DMSO to 10 mM) at a final concentration of 10 µM or equal volume of DMSO only, for up to 5 weeks. Before sample collection, TSCs were passaged at least once without MEFs in MEF-conditioned medium (+EZH2i/DMSO) to dilute out feeder cells. Subsequently, the inhibitor was removed by splitting the cells, and the cells were cultivated for up to 4 weeks with standard conditions for recovery.

### Combined DNMT1i and EZH2i treatment

TSCs were cultured on MEFs in TSC medium containing EZH2 inhibitor and DNMT1 at a final concentration of 10 µM and 1 µM, respectively, or equal volume of DMSO only, for up to 7 days. Before sample collection, TSCs were passaged at least once without MEFs in MEF-conditioned medium (+EZH2i/DMSO) to dilute out feeder cells.

### Western blot

For histone and histone modification western blots, cells were resuspended in Triton Extraction Buffer (TEB: DPBS containing 0.5% Triton

X-100 (v/v) and 1× Protease inhibitor) and lysed for 10 min on ice with gentle stirring. The lysates were spun for 10 min at 6,500g and 4 °C to pellet the nuclei. Nuclei were washed once with TEB to remove cell debris and again spun for 10 min at 6,500g and 4 °C. Nuclei were then resuspended in 0.2 N HCl and incubated overnight at 4 °C. The next day, samples were spun for 10 min at 6,500g and 4 °C to pellet the debris. The supernatant was transferred to a new tube and neutralized with 2 M NaOH at 1/10 of the supernatant volume. Reducing agent (Invitrogen, #NP0004), 40 mM Tris/Cl (pH 7.5) and Novex Tricine SDS Sample Buffer (2×) (Thermo Fisher Scientific, #LC1676) were added to the lysates, and the mixture was denatured at 85 °C for 2 min. Lysates were run on Novex 10 bis 20%, Tricin gels (Thermo Fisher Scientific, #EC6625BOX).

Blots were transferred using the iBlot 2 Dry Blotting system with iBlot 2 transfer stacks (Thermo Fisher Scientific, #IB24001) and imaged by HRP chemiluminescence using SuperSignal West Dura Extended Duration Substrate (Thermo Fisher Scientific, #34075) and ChemiDoc XRS + System (Bio-Rad, #1708265). Western blots were performed with anti-H3K27me3 (Cell Signaling, #9733S, 0.102 µg µl$^{-1}$, used at 1:20,000), anti-H2AK119Ub1 (Cell Signaling, #8240S, 0.538 µg µl$^{-1}$, used at 1:20,000) and anti-Histone H4 (Cell Signaling, #2935S, 0.075 µg µl$^{-1}$, used at 1:500).

## Co-IP–western blot/MS

EED (anti-EED, Abcam, #ab4469) and control IgG (anti-rabbit IgG, Cell Signaling, #2729) IPs were carried out using whole-cell lysates prepared from WT ES and TS cells, as well as WT TS cells treated with the EZH2 inhibitor for ~5 weeks (Tazemetostat/EPZ6438, Biovision, #2383-5, dissolved in DMSO to 10 mM) at a final concentration of 10 µM.

Briefly, ~5–10 million cells were resuspended with 600 µl of 0.5× Nuclear Lysis Buffer, (NLB, composed of 218.5 mM NaCl, 1.35 mM KCl, 4 mM Na$_2$HPO$_4$, 1 mM KH$_2$PO$_4$, 0.5% Triton X-100 and 0.05% Tween-20, pH 7.4) + 1× cOmplete Protease Inhibitor Cocktail. We selected these conditions from the published literature as sufficient for stringent characterization of PRC2 subcomplex characterization in mammalian cells. Resuspended samples were sonicated with Bioruptor Sonicator (30 s on/off, five cycles) and centrifuged for 10 min at ~20,000g at 4 °C to remove cellular debris. Antibodies were then added to clarified lysates, and immune complexes are allowed to form overnight (~16 h) in the cold room with end-to-end rotation.

For western blot: The immune complexes were then collected with 20 µl of Protein G Dynabeads (Thermo Fisher Scientific, #10003D), for 1 h in the cold room. Beads were washed thrice with 0.5× NLB, and immunoprecipitated material was then released with 1× SDS sample buffer at 90 °C for 5 min. Eluates were run on 4–12% acrylamide gels and transferred to PVDF membranes for immunoblotting and ECL detection. The following antibodies were used: anti-EED (Abcam, #ab4469, 1 µg µl$^{-1}$, used at 1:1,000), anti-DNMT3B (Cell Signaling, #48488S, 0.1 µg µl$^{-1}$, used at 1:1,000), anti-RNF2 (Cell Signaling, #5694, 0.22 µg µl$^{-1}$, used at 1:1,000), anti-SUZ12 (Cell Signaling, #3737T, 0.08 µg µl$^{-1}$, used at 1:1,000), anti-EZH2 (Cell Signaling, #5246, 0.4 µg µl$^{-1}$, used at 1:1,000), anti-RYBP (Millipore, #AB3637, 1 µg µl$^{-1}$, used at 1:1,000), anti-TUBULIN (Santa Cruz, #sc-32293, 0.2 µg µl$^{-1}$, used at 1:2,000) and anti-H3K27me3 (Cell Signaling, #9733S, 0.102 µg µl$^{-1}$, used at 1:1,000).

For MS: Immune complexes were prepared for downstream MS analysis using the Pierce MS-Compatible Magnetic IP Kit (Thermo Fisher Scientific, #90409) following the manufacturer's instructions until the second wash with buffer B. Then, buffer B was exchanged with 100 µl of 100 mM HN$_4$CO$_3$. This was followed by a tryptic digest including reduction and alkylation of the cysteines. Therefore, the reduction was performed by adding tris(2-carboxyethyl)phosphine with a final concentration of 5.5 mM at 37 °C on a rocking platform (500 r.p.m.) for 30 min. For alkylation, chloroacetamide was added with a final concentration of 24 mM at room temperature on a rocking platform (500 r.p.m.) for 30 min. Then, proteins were digested with 200 ng trypsin (Roche) shaking at 600 r.p.m. at 37 °C for 17 h. Samples

were acidified by adding 2.5 µl 100% formic acid, centrifuged shortly and placed on the magnetic rack. The supernatants, containing the digested peptides, were transferred to a new low-protein binding tube. Peptide desalting was performed on self-packed C18 columns in a tip. Eluates were lyophilized and reconstituted in 19 µl of 5% acetonitrile and 2% formic acid in water, briefly vortexed, and sonicated in a water bath for 30 s before injection to nanoscale liquid chromatography coupled to tandem mass spectrometry (nano-LC–MS/MS).

## LC–MS/MS instrument settings for shotgun proteome profiling and data analysis

TSC co-IP: LC–MS/MS was carried out by nanoflow reverse-phase liquid chromatography (Dionex Ultimate 3000, Thermo Fisher Scientific) coupled online to a Q-Exactive HF Orbitrap mass spectrometer (Thermo Fisher Scientific), as reported previously[107]. Briefly, the LC separation was performed using a PicoFrit analytical column (75 µm inner diameter (ID) × 50 cm long, 15 µm Tip ID; New Objectives) in-house packed with 3 µm C18 resin (Reprosil-AQ Pur, Dr. Maisch). Peptides were eluted using a gradient from 3.8% to 38% solvent B in solvent A over 120 min at 266 nl min$^{-1}$ flow rate. Solvent A was 0.1% formic acid, and solvent B was 79.9% acetonitrile, 20% H$_2$O and 0.1% formic acid. Nanoelectrospray was generated by applying 3.5 kV. A cycle of one full Fourier transformation scan mass spectrum (300–1,750 $m/z$, resolution of 60,000 at $m/z$ 200, automatic gain control (AGC) target 1 × 10$^6$) was followed by 12 data-dependent MS/MS scans (resolution of 30,000, AGC target 5 × 10$^5$) with a normalized collision energy of 25 eV. To avoid repeated sequencing of the same peptides, a dynamic exclusion window of 30 s was used.

Raw MS data were processed with MaxQuant software (v2.2.0.0) and searched against the Mus musculus proteome database UniProtKB with 22,001 entries, released in March 2021. Parameters of MaxQuant database searching were a false discovery rate (FDR) of 0.01 for proteins and peptides, a minimum peptide length of seven amino acids, a first search mass tolerance for peptides of 20 ppm and a main search tolerance of 4.5 ppm. A maximum of two missed cleavages was allowed for the tryptic digest. Cysteine carbamidomethylation was set as a fixed modification, while N-terminal acetylation and methionine oxidation were set as variable modifications. The MaxQuant processed output files can be found in Supplementary Tables 15 and 16, showing peptide and protein identification, accession numbers, sequence coverage of the protein (%) and $q$ values.

ESC co-IP: IP samples were prepared as above, but only 10% of peptides per sample were loaded onto Evotips Pure (Evosep) tips according to the manufacturer's protocol. Peptide separation was carried out by nanoflow reverse-phase liquid chromatography (Evosep One, Evosep) using the Endurance column (15 cm × 150 µm ID, with Reprosil-Pur C18 1.9 µm beads #EV1106, Evosep) with the 30 samples per day (30SPD) method. The LC system was online coupled to a timsTOF SCP mass spectrometer (Bruker Daltonics) applying the data-independent acquisition with parallel accumulation serial fragmentation (PASEF) method. MS data were processed with Dia-NN (v1.8.1) and searched against an in silico predicted mouse spectra library. The 'match between run' feature was used. A $t$-test with Benjamini–Hochberg correction was performed by Perseus (v2.0.3.1) on normalized protein values to identify significant interactions between KO and controls.

## LC–MS/MS sample preparation for histone modifications

ES and TS cells (5 million per sample) were washed twice with DPBS before snap freezing at −80 °C. A small aliquot of extracts was used for the bicinchoninic acid assay to quantify the protein concentration. Ten micrograms of each core histone sample was used for derivatization by propionylation, as this has been shown to increase the chromatographic performance of peptides on reversed-phase columns[108]. In brief, LC–MS-grade water was used to reach 27 µl volume in each sample. Three microlitres of 1 M triethylammonium bicarbonate buffer

was added to reach pH 8.5. Propionic anhydride was mixed with LC–MS-grade water in a ratio of 1:100, and 3 µl of the anhydride mixture was added immediately to the histone samples, vortexed and incubated for 2 min at room temperature. The reaction was quenched with 3 µl of 80 mM hydroxylamine, vortexed and incubated for 20 min at room temperature. Tryptic digestion was performed with 2 µl of trypsin (Roche, 100 ng µl⁻¹ in water; enzyme:protein ratio 1:50) per sample on a rocking platform at 37 °C for 4 h. A second round of propionylation with fresh buffers was performed, as above. Core histones were acidified by adding 1.5 µl of 100% formic acid. Peptide desalting was performed according to the manufacturer's instructions (Pierce C18 Tips, Thermo Fisher Scientific), but sample loading and elution from the tips was performed by ten repetitive steps of pipetting up and down. Desalted and propionylated histone peptides were reconstituted in 47 µl of 2% formic acid and 5% acetonitrile. Three microlitres of a chicken lysozyme digest (70 pmol µl⁻¹) was added as an internal standard to each sample, vortexed, sonicated and transferred to micro-volume inserts.

## Core histone profiling by targeted MRM

Core histone peptide separation was performed on an LC instrument (1290 series UHPLC; Agilent) in technical triplicates (3 µg protein per injection), online coupled to a triple quadrupole hybrid ion trap mass spectrometer QTrap 6500 (Sciex). A Reprosil-PUR C18-AQ (1.9 µm, 120 Å, 150 × 2 mm ID; Dr. Maisch) column at a controlled temperature of 30 °C was used for separation of peptides. Peptides were eluted using a gradient from 2% to 30% solvent B in solvent A over 39 min at 250 µl min⁻¹ flow rate. Solvent A was 10 mM ammonium acetate, pH 3.5 (adjusted with acetic acid), and solvent B was 0.1% formic acid in acetonitrile. Transition settings for H3 histone multiple reaction monitoring (MRM) were taken from the literature[109] and consisted of two transitions for all 42 histone peptides including the following lysine PTMs: acetylation, methylation, dimethylation and trimethylation as well as phosphorylations on serine, threonine and tyrosine. Transitions were monitored in a 300 s window of the expected elution time and acquired at unit resolution (peak width at 50% was 0.7 ± 0.1 Da tolerance) in quadrupole Q1 and Q3. Data acquisition was performed with an ion spray voltage of 5.5 kV in positive mode of the ESI source, N₂ as the collision gas was set to high, curtain gas was set to 30 psi, ion source gas 1 and 2 were set to 50 and 70 psi, respectively, and an interface heater temperature of 350 °C was used.

Relative quantification of the peaks was performed using MultiQuant software v.2.1.1 (Sciex). The integration setting was a peak-splitting factor of 2, a Gaussian smoothing width of 2 was applied, and all peaks were reviewed manually. Only the average peak area of the first transition was used for calculations. Normalization was done according to the sum of the intensities of the three different H3 peptides (Supplementary Table 11).

## LC–MS/MS sample preparation for simultaneous determination of cytidine modifications

Genomic DNA (gDNA) was extracted from 1 million cells using 100 µl genome lysis buffer (10 mM Tris, 10 mM NaCl, 10 mM EDTA and 0.5% SDS[110,111], which was supplemented with RNase (Roche) and Proteinase K (Invitrogen) to a final concentration of 1 mg ml⁻¹. This solution was incubated at 37 °C for 1 h followed by overnight incubation at 55 °C. The next day, 300 µl of water were added along with an equal volume of phenol–chloroform–isoamyl alcohol (Invitrogen), samples were vortexed briefly, and then centrifuged at 21,000g for 10 min at room temperature. The aqueous phase was collected, and this process was repeated. After the second extraction, the aqueous phase was combined with 20 µl 5 M NaCl, 1 µl 20 mg ml⁻¹ glycogen (Thermo Fisher Scientific) and 880 µl 100% ethanol, then placed at −20 °C overnight. This solution was stored at −20 °C overnight and then centrifuged at 21,000g 4 °C for 1 h the next day. This was followed by two washes with 70% ethanol, elution in 50 µl ultrapure distilled water (Invitrogen) and quantification with the Qubit dsDNA HS assay (Thermo Fisher Scientific).

## Simultaneous determination of cytidine modifications by targeted LC–MS/MS

Five-hundred nanograms of gDNA was used for profiling the following cytidine modifications: 2′-deoxycytidine (dC), 5-methyl-2′-deoxycytidine (5-mdC, called 5-mC in text), 5-hydroxymethyl-2′-deoxycytidine (5-hmdC, 5-hmC in text), 5-formyl-2′-deoxycytidine (5-fodC) and 5-carboxyl-2′-deoxycytidine (5-cadC), as well as the other three bases 2′-deoxyguanosine (dG), 2′-deoxyadenosine (dA) and thymidine (T). An MRM method with three transitions was established by using pure compounds. Furthermore, a dilution series of these standards was used for absolute quantification. Five-hundred nanograms of DNA was dissolved in a total of 27 µl water in Protein LoBind Tubes, and 3 µl DNA Degradase buffer and 1 µl DNA Degradase Plus enzyme (Zymo Research) were added. The DNA digestion efficiency was monitored by using 100 ng 5-methylcytosine and 5-hydroxymethylcytosine DNA standard sets (Zymo Research) in parallel under identical conditions. The digest was carried out at 37 °C on a rocking platform (600 r.p.m.) for 2 h. The digest was stopped by adding 1 µl of 5% formic acid.

Ten microlitres of the DNA digests was used for LC–MS/MS analysis in each technical replicate. Cytidine separation was performed on an LC instrument (1290 series UHPLC; Agilent), online coupled to a triple quadrupole hybrid ion trap mass spectrometer QTrap 6500 (Sciex). Cytidines were eluted from a Reprosil-PUR C18-AQ (1.9 µm, 120 Å, 150 × 2 mm ID; Dr. Maisch) column at a controlled temperature of 30 °C, using a gradient from 2% to 98% solvent B in solvent A over 10 min at 250 µl min⁻¹ flow rate. Solvent A was 10 mM ammonium acetate, pH 3.5 (adjusted with acetic acid), and solvent B was 0.1% formic acid in acetonitrile. Transition settings are provided in Supplementary Table 12, consisting of three transitions for each base. Transitions were monitored in a 240 s window of the expected elution time and acquired at unit resolution (peak width at 50% was 0.7 ± 0.1 Da tolerance) in quadrupole Q1 and Q3. Data acquisition was performed with an ion spray voltage of 5.5 kV in positive mode of the ESI source, N₂ as the collision gas was set to high, curtain gas was set to 30 psi, ion source gas 1 and 2 were set to 50 and 70 psi, respectively, and an interface heater temperature of 350 °C was used.

Relative quantification of the peaks was performed using MultiQuant software v.2.1.1 (Sciex). The integration settings were a peak-splitting factor of 2 points and a Gaussian smoothing width of 2. All peaks were reviewed manually. Only the average peak area of the first transition was used for calculations. Data were normalized to thymidine levels to account for DNA input variations (Supplementary Table 12).

## RRBS

Concentration of gDNA was quantified using a Qubit 3.0 Fluorometer. RRBS was performed on 100 ng gDNA of each sample using the NuGen Ovation RRBS Methyl-Seq System (Tecan, #0353) following the manufacturer's recommendations with the following modifications: after the final repair step, the bisulfite conversion of DNA was conducted using the Qiagen EpiTect Fast Bisulfite Conversion kit (Qiagen, #59824) following the manufacturer's recommendations, eluting the bisulfite converted DNA in 23 µl EB. Libraries were amplified with 12 cycles of PCR. Amplified library purification with Agencourt RNAclean XP beads (Beckman Coulter, #A63987) was performed twice (1×). The purified libraries were quality-assessed on an Agilent 4150 TapeStation HS D1000 ScreenTape and sequenced for 100 bp single-end reads on a NovaSeq 6000 platform (Illumina).

## WGBS

gDNA was prepared as above and sheared in Covaris micro TUBE AFA Fiber Pre-Slit Snap-Cap tubes (SKU: 520045), followed by clean-up with the Zymo DNA Clean & Concentrator-5 Kit (#D4013) according to the manufacturer's guidelines. Sheared gDNA was bisulfite converted following the manufacturer's guidelines with the EZ DNA

Methylation-Gold Kit (Zymo #D5005), and libraries were prepared using the Accel-NGS Methyl-seq DNA library kit (Swift Biosciences, #30024-SWI). Libraries were cleaned using Agencourt AMPure XP beads (Beckman Coulter, #A63881), and the absence of adapters was confirmed on the Agilent TapeStation HS D5000. The final libraries were sequenced on a NovaSeq 6000 platform (Illumina) yielding 150 bp paired-end reads.

## RNA-seq

TSC cell lines (around 1 million cells per sample; see above for culture conditions, genetic background and treatments) were dissociated with trypsin–EDTA (0.05%) for 5 min at 37 °C, 5% $CO_2$ to obtain a single cell suspension. Cells were then collected, washed with ice cold DPBS and centrifuged at 4 °C, 300$g$ for 5 min. Two biological replicates for samples of the acute inhibitor response experiment were prepared. For all other samples, two technical replicates for each sample were prepared. Subsequently, cell pellets were resuspended in 350 µl RLT Plus buffer containing 1% 2-mercaptoethanol (Thermo Fisher Scientific, #21985023). After cell lysis by trituration and vortexing, RNA was extracted using RNeasy Plus Micro Kit (Qiagen, #74034) and RNA concentration and quality was measured using the Agilent RNA Screen-Tape (Agilent Technologies, #5067-5576) on an Agilent 4150 TapeStation system. All samples analysed had an RINe value higher than 8.0, and were subsequently used for library preparation. mRNA libraries were prepared using KAPA Stranded RNA-Seq Kit (KapaBiosystem, #KK8421/07962207001) according to the manufacturer's instructions. Five-hundred nanograms of total RNA was used for each sample to enter the library preparation protocol. For adapter ligation, dual indexes were used (NEXTFLEX Unique Dual Index Barcodes #NOVA-514150 and #NOVA-514151) at a working concentration of 71 nM (5 µl of 1 µM stock in each 70 µl ligation reaction). Twelve library PCR cycles were used. Quality and concentration of the obtained libraries were measured using Agilent High Sensitivity D5000 ScreenTape (Agilent Technologies, #5067- 5592) on an Agilent 4150 TapeStation. All libraries were sequenced using 100 bp paired-end sequencing (200 cycles kit) on a NovaSeq 6000 platform.

## Hi-C

Two million cells for both ESCs and TSCs were dissociated in a single-cell suspension using pre-warmed trypsin–EDTA (0.05%) and incubated for 5–10 min at 37 °C. Trypsin was blocked by adding 10% FCS/DPBS, and cells were centrifuged for 5 min at -300$g$. Cell pellet was resuspended in a 2% paraformaldehyde (PFA), 10% foetal calf serum (FCS) fixation solution and incubated at room temperature for 10 min while tumbling. The reaction was quenched on ice by adding glycine (final concentration 125 mM) and cells were collected by centrifugation at 400$g$ for 8 min at 4 °C. To extract the nuclei, cells were incubated on ice for 10 min with ice-cold Lysis Buffer (50 mM Tris–HCl pH 7.5, 150 mM NaCl, 5 mM 788 EDTA, 0.5% NP40, 1.15% Triton X-100 and 25× Protease Inhibitor in Milli-Q water). Extracted nuclei were centrifuged at 750$g$ for 5 min at 4 °C, washed twice with 1× DPBS, snap frozen and stored at −80 °C.

Hi-C libraries were prepared as described previously[112]. Briefly, nuclei were first permeabilized using 0.5% SDS at 62 °C for 10 min, and later, chromatin was digested with DpnII (NEB, #R0543) for 90 min at 37 °C with gentle rotation. The overhangs generated by digestion were filled and marked with biotin-14-dATP (Thermo Fisher Scientific, #19524016) by a 90 min incubation at 37 °C with gentle rotation. After, DNA fragments were ligated for 4 h at 20 °C with gentle rotation using T4 DNA Ligase (Thermo Fisher Scientific, #M0202M). Chromatin was then reverse-crosslinked, precipitated and sheared with Covaris S220 (two cycles, each 50 s long; 10% duty; 4 intensity; 200 cycles per burst). The biotin-marked-DNA shared fragments were pulled down using Dynabeads MyOne Streptavidin T1 beads (Thermo Fisher Scientific, #65601), purified and further processed for Illumina sequencing with NEBNext Ultra II Library Prep Kit for Illumina according to the kit

guidelines (NEBNext End Prep, Adaptor Ligation, PCR enrichment of Adaptor-Ligated DNA using NEBNext Multiplex Oligos for Illumina). Clean-up and size selection were performed with AMPure beads. Hi-C libraries were sequenced on a NovaSeq 6000 platform.

## Nanopore ultralong read sequencing

We extracted ultrahigh-molecular-weight DNA with the Nanobind CBB big DNA kit (Circulomics, #NB-900-001-01) following the manufacturer's protocol with minor modifications. Briefly, 6 million cells (TSC1) were collected and snap-frozen in liquid nitrogen for later use. Frozen cells were thawed on ice and thoroughly resuspended in 40 µl of room temperature equilibrated 1× PBS. The cell suspension was supplemented with 40 µl of Proteinase K solution and gently mixed by pipetting (10×). The elution of DNA bound to the Nanobind disk was performed using 760 µl modified elution buffer (EB+) for 18 h at room temperature. DNA-containing supernatant was transferred into an Eppendorf 1.5 ml DNA LoBind tube. The remaining DNA bound to the Nanobind disk was collected by a single centrifugation step at 10,000$g$ for 10 s at room temperature and transferred to the stock solution. The eluate was homogenized by gentle resuspension (5×) and subsequently incubated for 2 h at room temperature. The DNA solution was incubated on an Eppendorf Thermomixer for an additional 30 min at 37 °C with gentle mixing (5×) every 15 min. The homogenized sample was quantified using a Qubit Fluorometer (Thermo Fisher Scientific) in conjunction with the Qubit dsDNA BR assay kit following the manufacturer's instructions. DNA purity was assessed with a Nanodrop One Spectrophotometer.

We prepared an ultralong nanopore sequencing library with a total of 35 µg of high-molecular-weight DNA following the manufacturer's protocol with minor modifications. Briefly, utilizing the Oxford Nanopore Technologies (ONT) sequencing library kit SQK-ULK001, 6 µl of Transposase (FRA) was resuspended in 244 µl of fragmentation buffer (FDB) and subsequently added to 750 µl of DNA solution following gentle resuspension (10×). Fragmented DNA was supplemented with 5 µl of rapid sequencing adapter (RAP F) and gently resuspended (10×) following a 1 h incubation at room temperature. The sequencing library was precipitated using 500 µl of precipitation buffer (NAF) to a Nanobind disk for the removal of unbound sequencing adapter, and we removed small DNA fragments (<3 kb) by washing the disk with ONT's long fragment buffer. The library was then incubated with 225 µl standard elution buffer (EB) for 18 h at RT and supernatant was transferred into a fresh Eppendorf 1.5 ml DNA LoBind tube. The remaining DNA bound to the Nanobind disk was collected by a single centrifugation step at 10,000$g$ for 10 s at room temperature and transferred to the stock solution. Final eluate was homogenized by gentle resuspension (5×) and subsequently incubated for 2 h at room temperature. The library was incubated on an Eppendorf Thermomixer for an additional 30 min at 37 °C with gentle resuspension (5×) every 15 min. Prior flow cell loading 75 µl of the library was mixed with 75 µl of sequencing buffer (SQB) by gentle resuspension (10×) and incubated at room temperature for 30 min. A total of two PromethION flow cells were primed using ONT's flow cell priming kit (EXP-FLP002) following the manufacturer's recommendation, and the sequencing library was loaded onto the flow cell.

## MINUTE-ChIP

MINUTE-ChIP was performed essentially as described previously[48]. Briefly, native cell pellets containing 1–2 million cells of various treatment conditions or genetic background were lysed and digested with MNase to enrich for mononucleosome population. The digestion was quenched by EGTA-containing end-repair and ligation buffer, in which each sample was ligated to adaptor molecules carrying unique barcodes. Ligation was quenched by EDTA-containing lysis dilution buffer, before combining all samples in one tube. After centrifugation, pool supernatant was recovered and aliquoted for individual ChIP. A 2 million cell-equivalent of pool supernatant was used for ChIP against

each histone modification, with the anti-H3K4me3 (3 µl per ChIP, Millipore #04-745), anti-H3K27me3 (1 µg per ChIP, Cell Signaling #9733) and anti-H2AK119ub (0.6 µg per ChIP, Cell Signaling #8240) antibodies pre-coupled to Protein A magnetic beads. After thorough washes, ChIP DNA was recovered with Proteinase K treatment and purified for linear amplification by in vitro transcription. The RNA product was then ligated to a pre-adenylated RNA 3′ adaptor (RA3), which served as a primer binding site for reverse transcription. The resulting complementary DNA was purified and used as a template for library PCR with barcoded primers compatible with Illumina sequencing platform. Typically, 100,000 to 200,000 cell equivalents of pool supernatant was used as Input, which is subjected to the same experimental workflow for library construction as the ChIP DNA. All nucleic-acid purification were carried out with AMPure SPRI size selection method (Beckman Coulter). Library size distribution was assessed by Agilent BioAnalyzer and were quantified by Qubit DNA high sensitivity assay before dilution for sequencing on a NovaSeq 6000 platform.

Two sample pools were prepared in this study. Triplicates of TSC1 were included in each pool to serve as a reference for samples in each pool. Pool/Batch 1 includes triplicates of TSC1 WT, ESC WT and *Dnmt3b* KO. Pool/Batch 2 includes triplicates of TSC1 WT, *Kdm2b* KO, *Rnf2* KO, *Eed* KO and TSC1 WT recovered for 4 weeks from a 5 week EZH2i treatment (Supplementary Table 4).

## MNase-based ChIP–BS-seq and library construction
Five million cells were resuspended in 500 µl cell lysis buffer (20 mM Tris–HCl pH 8, 85 mM KCl and 0.5% NP40) and incubated for 5 min on ice followed by 2,500*g* centrifugation at 4 °C for 5 min (ref. [113]). Supernatant was removed and pelleted nuclei were resuspended in 100 µl PBS, after which an additional 100 µl of 2× lysis buffer supplemented with 40 U µl$^{-1}$ microccocal nuclease (NEB) was added (100 mM Tris–HCl pH 8.0, 300 mM NaCl, 2% Triton X-100, 0.2% sodium deoxycholate and 10 mM CaCl$_2$, ref. [114]). Nuclear lysis was carried out on ice for 20 min followed by a 25 min incubation at 37 °C, which was empirically determined to yield mononucleosome-sized fragments. The micrococcal nuclease reaction was terminated with the addition of 800 µl lysis dilution buffer (50 mM Tris–HCl pH 8.0, 150 mM NaCl, 1% Triton X-100, 50 mM EGTA, 50 mM EDTA and 0.1% sodium deoxycholate), and 2 µg of H3K27me3 antibody was added (Thermo Fisher Scientific Scientific, #MA5-11198). IP was carried out at 4 °C overnight with gentle rotation followed by incubation with protein A Dynabeads (Thermo Fisher Scientific) for 4 h the next day. Bead-bound immune complexes were washed at 4 °C two times with RIPA buffer (0.1% DOC, 0.1% SDS, 1% Triton X-100, 10 mM Tris–HCl pH 8.0, 1 mM EDTA and 140 mM NaCl), and then one time with each of the following: RIPA high salt (0.1% DOC, 0.1% SDS, 1% Triton X-100, 10 mM Tris–HCl pH 8.0, 1 mM EDTA and 360 mM NaCl), LiCl wash buffer (250 mM LiCl, 0.5% NP40, 0.5% deoxycholate, 1 mM EDTA and 10 mM Tris–HCl, pH 8.0) and TE pH 8.0. DNA was then eluted from the Protein A beads by dissolving them in 100 µl ChIP elution buffer (TE, 0.1% SDS and 300 mM NaCl) with 0.2 mg ml$^{-1}$ Proteinase K (Invitrogen) and incubating at 55 °C overnight. The next day 300 µl TE was added to the reaction along with 400 µl phenol–chloroform–isoamyl alcohol (Invitrogen), and the solution was briefly vortexed and then centrifuged at 21,000*g* for 10 min at room temperature in phase lock tubes (VWR). After centrifugation, the aqueous phase was collected and combined with 20 µl 5 M NaCl, 1 µl 20 mg/ml glycogen (Thermo Fisher Scientific), and 880 µl 100% ethanol. This solution was stored at −20 °C overnight and then centrifuged at 21,000*g* 4 °C for 1 h the next day. This was followed by two washes with 70% ethanol and eluted in 20 µl 1× TE. The resulting DNA was bisulfite converted with the EZ DNA Methylation Gold Kit (Zymo) and used as input for the Accel-NGS Methyl-Seq DNA Library Kit (Swift/Integrated DNA Technologies) following the manufacturer's protocol and described above. All libraries were sequenced using 100 bp paired-end sequencing (200 cycles kit) on a NovaSeq 6000 platform.

## EED ChIP–seq
Ten million cells were crosslinked in a 1% formaldehyde solution for 5 min at room temperature, after which glycine was added to a final concentration of 125 mM and incubated for 5 min to quench the reaction. These fixed cells were centrifuged at 2,500*g* for 5 min at 4 °C and the pellet was washed twice with 1 ml PBS. Nuclei were extracted by incubating the fixed cells with 500 µl of cell lysis buffer (20 mM Tris–HCl pH 8.0, 85 mM KCl and 0.5% NP40) for 10 min on ice then spun down for 3 min at 2,500*g*. The pellet was resuspended in nuclei lysis buffer (10 mM Tris pH 7.5, 1% IGEPAL, 0.5% sodium deoxycholate and 0.1% SDS), then sonicated on a Covaris E220 Evolution sonicator (peak incident power 140.0, duty factor 5.0, cycles per burst 200, 20 min). After sonication, chromatin was spun down at 21,000*g* for 10 min to pellet insoluble material. The supernatant was transferred to a fresh tube, the volume was increased to 1 ml with chip dilution buffer (0.01% SDS, 1.1% Triton X-100, 1.2 mM EDTA, 16.7 mM Tris–HCl pH 8.1 and 167 mM NaCl), and 5 µg of EED antibody was added (Abcam ab240650). IP was carried out at 4 °C overnight with gentle rotation followed by incubation with Protein A Dynabeads (Thermo Fisher Scientific) for 4 h. IP was followed by two washes of each of the following: low-salt wash buffer (0.1% SDS, 1% Triton X-100, 2 mM EDTA, 20 mM Tris–HCl pH 8.1, 150 mM NaCl); high salt wash buffer (0.1% SDS, 1% Triton X-100, 2 mM EDTA, 20 mM Tris, pH 8.1, 500 mM NaCl); LiCl wash buffer (0.25 M LiCl, 1% NP40, 1% deoxycholate, 1 mM EDTA and 10 mM Tris–HCl pH 8.1) and TE buffer pH 8.0 (10 mM Tris–HCl, pH 8.0 and 1 mM EDTA pH 8.0). DNA was eluted twice using 50 µl of EB (0.5–1% SDS and 0.1 M NaHCO$_3$) at 65 °C for 15 min. A 16 µl volume of reverse crosslinking salt mixture (250 mM Tris–HCl, pH 6.5, 62.5 mM EDTA pH 8.0, 1.25 M NaCl and 5 mg ml$^{-1}$ Proteinase K) was added, and samples were allowed to incubate at 65 °C overnight. The next day 284 µl TE was added to the reaction along with 400 µl phenol–chloroform–isoamyl alcohol (Invitrogen), and the solution was briefly vortexed and then centrifuged at 21,000*g* for 10 min at room temperature in phase lock tubes (VWR). After centrifugation, the aqueous phase was collected and combined with 20 µl 5 M NaCl, 1 µl 20 mg ml$^{-1}$ glycogen (Thermo Fisher Scientific) and 880 µl 100% ethanol. This solution was stored at −20 °C overnight and then centrifuged at 21,000*g* 4 °C for 1 h the next day. This was followed by two washes with 70% ethanol and elution of the pelleted DNA in 50 µl 1× TE. Libraries were prepared using NEBNext Ultra II DNA Library Prep Kit for Illumina (NEB) following the manufacturer's protocol. All libraries were sequenced using 100 bp paired-end sequencing (200 cycles kit) on a NovaSeq 6000 platform.

## WGBS data processing
Raw reads were subjected to adapter and quality trimming using cutadapt (version 2.4; parameters: –quality-cutoff 20 –overlap 5 minimum-length 25 –adapter AGATCGGAAGAGC -A AGATCGGAAGAGC), followed by trimming of 10 and 5 nucleotides from the 5′ and 3′ end of the first read and 15 and 5 nucleotides from the 5′ and 3′ end of the second read[115]. The trimmed reads were aligned to the mouse genome (mm10) using BSMAP (version 2.90; parameters: -v 0.1 -s 16 -q 20 -w 100 -S 1 -u -R) (ref. [116]). A sorted BAM file was obtained and indexed using samtools with the 'sort' and 'index' commands (version 1.10) (ref. [117]). Duplicates were removed using the 'MarkDuplicates' command from GATK (version 4.1.4.1) and default parameters[118]. Methylation rates were called using mcall from the MOABS package (version 1.3.2; default parameters)[119]. All analyses were restricted to autosomes, and only CpGs covered by at least 10 and at most 150 reads were considered for downstream analyses.

## RRBS data processing
Raw reads were subjected to adapter and quality trimming using cutadapt (version 2.4; parameters: –quality-cutoff 20 –overlap 5 –minimum-length 25 –adapter AGATCGGAAGAGC -A AGATCGGAAGAGC), followed by NuGEN diversity adapter trimming (https://

github.com/nugentechnologies/NuMetRRBS). The trimmed reads were aligned to the mouse genome (mm10) using BSMAP (version 2.90; parameters: -v 0.1 -s 12 -q 20 -w 100 -S 1 -u -R -D C-CGG). A sorted BAM file was obtained and indexed using samtools with the 'sort' and 'index' commands (version 1.10). Aligned reads were deduplicated on the basis of unique molecular identifiers (UMIs) using NuDup (https://github.com/nugentechnologies/nudup; parameters: start 6 –length 6). Methylation rates were called using mcall from the MOABS package (version 1.3.2; default parameters). All analyses were restricted to autosomes, and only CpGs covered by at least 10 and at maximum 150 reads were considered for downstream analyses.

## RNA-seq data processing
Raw reads were subjected to adapter and quality trimming with cutadapt (version 2.4; parameters: –quality-cutoff 20 overlap 5–minimum-length 25 –interleaved –adapter AGATCGGAAGAGC -A AGATCGGAAGAGC), followed by poly-A trimming with cutadapt (parameters: –interleaved –overlap 20 –minimum-length –adapter 'A[100]' –adapter 'T[100]'). Reads were aligned to the mouse reference genome (mm10) using STAR (version 2.7.5a; parameters: –runMode alignReads –chimSegmentMin 20 –outSAMstrandField intronMotif –quantMode GeneCounts)[120], and transcripts were quantified using stringtie (version 2.0.6; parameters: -e) (ref. [121]) with the GENCODE annotation (release VM19). For the repeat expression quantification, reads were re-aligned with additional parameters '–outFilterMultimapNmax 50'.

## Hi-C data processing
Raw reads were subjected to adapter and quality trimming with cutadapt (version 2.4; parameters: –quality-cutoff 20– –overlap 5 –minimum-length 25–adapter AGATCGGAAGAGC -A AGATCGGAAGAGC). Mates were separately aligned to the mouse genome (mm10) using bwa with the 'mem' command (version 0.7.17; parameters: -A 1 -B 4 -E 50 -L 0) (ref. [122]). Hi-C matrices for each replicate were built using HiCExplorer with the 'hicBuildMatrix' command (version 3.6; parameters: –binSize 5000 –restrictionSequence GATC –danglingSequence GATC–minMappingQuality 30) (ref. [123]). TSC and ESC replicates were merged respectively using 'hicSumMatrices'. TSC and ESC matrices were normalized together using 'hicNormalize' (parameters: –smallest) and corrected using 'hicCorrectMatrix' (parameters: –correctionMethod KR). For whole-chromosome representation and compartment analysis, bins of matrices at 5 kb resolution were merged into 100 kb bins using 'hicMergeMatrixBins' (parameters: –nb 20). Log$_2$ ratio matrices comparing TSCs with ESCs were generated with 'hicCompareMatrices' (parameters: –operation log2ratio). Full chromosomes were visualized using 'hicPlotMatrix' (parameters: –vMin 1 –vMax 100000 for corrected interaction matrices and –vMin -4 –vMax 4 for log$_2$ ratio matrices) and counts across different distances were compared with 'hicPlotDistVsCounts' (parameters: –maxdepth 80000000 –chromosomeExclude chrX chrY).

## Nanopore data processing
Nanopore data was processed using Nanopype (v1.1.0, base calling: guppy v4.0.11 with r9.4.1 high-accuracy configuration; alignment: minimap2 v2.10 with -ax map-ont -L –MD; methylation calling: nanopolish v0.13.2; reference genome: mm10) (refs. [124]–[126]). Only methylation calls with an absolute likelihood of at least 2.5 were used for downstream analyses. For genome browser tracks, only CpGs covered by at least ten reads were shown.

## ChIP–BS-seq processing
Raw reads of ESC and TSC H3K27me3 ChIP–BS-seq samples as well as their respective input samples were subjected to adapter and quality trimming with cutadapt (version 2.4; parameters: –quality-cutoff 20 –overlap 5 –minimum-length 25–adapter AGATCGGAAGAGC -A AGATCGGAAGAGC). Reads were aligned to the mouse genome (mm10) using BSMAP (version 2.90; parameters: -v 0.1 -s 16 -q 20 -w 100 -S 1 -u -R). A sorted BAM file was obtained and indexed using samtools with the 'sort' and 'index' commands (version 1.10). Duplicate reads were identified and removed using GATK (version 4.1.4.1) 'MarkDuplicates' and default parameters. After careful inspection and validation of high correlation, replicates of treatment and input samples were merged respectively using samtools 'merge'. Methylation rates were called using mcall from the MOABS package (version 1.3.2; default parameters). All analyses were restricted to autosomes, and only CpGs covered by at least 10 and at most 150 reads were considered for downstream analyses. Genome-wide coverage tracks for single and merged replicates normalized by library size were computed using deepTools bamCoverage (parameters: –normalizeUsing RPGC –extendReads –smoothLength 300). Coverage tracks were subtracted by the respective input using deeptools 'bigwigCompare'.

## ChIP–seq processing
Raw reads of ESC and TSC EED and publicly available ESC H3K27me3 ChIP–seq samples were subjected to adapter and quality trimming with cutadapt (version 2.4; parameters: –quality-cutoff 20 –overlap 5 –minimum-length 25 –adapter AGATCGGAAGAGC -A AGATCGGAAGAGC) as were their respective input samples. Reads were aligned to the mouse genome (mm10) using BWA with the 'mem' command (version 0.7.17, default parameters)[122]. A sorted BAM file was obtained and indexed using samtools with the 'sort' and 'index' commands (version 1.10) (ref. [117]). Duplicate reads were identified and removed using GATK (version 4.1.4.1) 'MarkDuplicates' and default parameters. After careful inspection and validation of high correlation, replicates of treatment and input samples were merged respectively using samtools 'merge'. H3K27me3 domains in ESCs were called for each sample with its respective input using peakranger 'bcp' (version 1.18) (ref. [127]). Only regions that were called as domain in at least two of the samples were considered for the final selection and merged using bedtools 'mergeBed' (parameters: -d 50) (ref. [128]). Retained regions smaller than 100 bp were removed from the set.

EED peaks were called using MACS2 'callpeak' (version 2.1.2; parameters: –bdg –SPMR –broad) based on merged replicates using the input samples as control samples[129], and only peaks with a $q$ value <0.01 were considered for downstream analyses. Genome-wide coverage tracks for single and merged replicates normalized by library size were computed using deepTools bamCoverage (parameters: –normalizeUsing RPGC –extendReads –smoothLength 300). Coverage tracks were subtracted by the respective input using deeptools 'bigwigCompare'.

## MINUTE-ChIP processing
MINUTE-ChIP multiplexed FASTQ files were processed using 'minute', a workflow implemented in Snakemake[130]. To ensure reproducibility, a conda environment was set up. Source code and documentation are fully available on GitHub: https://github.com/NBISweden/minute.

Main steps performed are described below.

Adaptor removal: Read pairs matching parts of the adaptor sequence (SBS3 or T7 promoter) in either read1 or read2 were removed using cutadapt v3.2.

Demultiplexing and deduplication: Reads were demultiplexed using cutadapt v3.2 allowing only one mismatch per barcode and written into sample-specific FASTQ files used for subsequent mapping.

Mapping: Sample-specific paired FASTQ files were mapped to the reference mm10 using bowtie2 v2.3.5.1 (ref. [131]) with –fast and –reorder parameter. Alignments were processed into sorted BAM files and replicates were pooled using samtools v1.10.

Deduplication: Duplicate reads are marked using UMI-sensitive deduplication tool je-suite (v2.0.RC) (https://github.com/gbcs-embl/Je/). Read pairs are marked as duplicates if their read1 (first-in-pair) sequences have the same UMI (allowing for one mismatch) and map to

the same location in the genome. Blacklisted regions as downloaded from ENCODE were then removed from BAM files using bedtools v2.30.

Generation of coverage tracks and quantitative scaling: Input coverage tracks with 1 bp resolution in bigWig format were generated from BAM files using deepTools (v3.5.0) (ref. [132]) bamCoverage and scaled to a reads-per-genome-coverage of one (1xRPGC, also referred to as '1× normalization') using the mm10 effective genome size. ChIP coverage tracks were generated from BAM files using deepTools (v3.5.0) bamCoverage. Quantitative scaling of the ChIP–seq tracks among conditions within each pool was based on their input-normalized mapped read count (INRC). INRC was calculated by dividing the number of unique reference-mapped reads by the respective number of input reads: #mapped[ChIP]/#mapped[Input]. This essentially corrects for an uneven representation of barcodes in the input. It has been previously shown that INRCs are proportional to the amount of epitope present in each condition[48]. Reference condition (TSC WT) was scaled to 1× coverage (also termed reads per genome coverage, RPGC). All other conditions within the same pool were scaled relative to the reference using the ratio of INRCs multiplied by the scaling factor determined for 1× normalization of the reference: (#mapped[ChIP]/#mapped[Input])/ (#mapped[ChIP_Reference]/#mapped[Input_Reference]) × scaling factor.

Quality control: FastQC was run on all FASTQ files to assess general sequencing quality. Picard (v2.24.1) was used to determine insert size distribution, duplication rate and estimated library size. Mapping stats were generated from BAM files using samtools (v1.10) idxstats and flagstat commands. Final reports with all the statistics generated throughout the pipeline execution are gathered with MultiQC[133].

### Feature annotation

One-kilobase genomic tiles were generated by segmenting the genome using bedtools makewindows (parameters: -w 1000 -s 1000). Annotations of highly methylated domains (HMDs) and PMDs in mm10 were downloaded from https://zwdzwd.github.io/pmd ref. [103]. The mm10 gene annotation was downloaded from GENCODE (VM19). Promoters were defined as 1,500 bp upstream and 500 bp downstream of the transcription start site.

Annotations of CGIs for mm10 were downloaded from UCSC. CGI shores were defined as the 2 kb flanking each island, while CGI shelves were defined as the 2 kb flanking the shores. CGIs were defined to be targeted by PRC2 in ESCs if at least 20% of the CGI overlapped with a H3K27me3 domain (see ChIP–seq processing). CGIs were defined as promoter CGI if at least 20% of the CGI or the promoter overlapped. CGIs were associated with EED peaks if at least 1 bp overlapped. The distance to the nearest transcription start site for all CGIs was calculated using bedtools 'closestBed'.

Annotations of repeats for mm10 were downloaded from the UCSC RepeatMasker. Full-length IAP elements were defined as described previously[134]: Elements annotated as inner parts (containing the keyword 'int') were merged if they belonged to the same subfamily and were located within maximal 200 base pairs of each other. Second, only the merged inner parts with an annotated IAP LTR within a distance of at most 50 base pairs on each side were selected as full-length element candidates. The subfamily per element was defined on the basis of the inner part.

### Definition of hyper CGIs

Mouse ExE: Hyper CGIs were defined using the methylation difference of mouse epiblast and ExE based on previous findings[23]. This previously reported set of CGIs was re-defined using higher-coverage WGBS data (GSE137337). CGIs were termed hyper CGIs if the difference of the average methylation of a CGI was more than 0.1 when comparing averaged WT ExE replicates with averaged WT epiblast replicates. Additionally, either more than half of the CpGs within a CGI were required to have a minimum difference of 0.1 or the CGI was required to contain a

differentially methylated region with higher methylation in the ExE. Differentially methylated regions were called on the basis of CpGs located in CGIs using metilene (version 0.2-8; parameters: -m 10 -d 0.1 -c 2 -f 1 -M 80 -v 0.7) (ref. [135]) and filtered for a q value <0.05. CGIs methylated in the epiblast (≥0.2) were excluded from the set.

TSC PRC KOs: CGIs hypermethylated in PRC KOs were defined as CGIs gaining at least 0.2 in average methylation in any of *Kdm2b* KO, *Rnf2* KO and *Eed* KO compared with the WT. Here, in contrast to the other set, we prioritized strong gain of methylation independent of the original WT levels for the definition in order to investigate regions that change drastically upon loss of PRC.

### Average feature methylation analysis

For every sample, the arithmetic mean was calculated across features (tiles, CGIs, shores, shelves, repeats, promoters, gene bodies). A feature was considered only if at least three CpGs were covered within a region. Replicates of WGBS WT epiblast and ExE were averaged per CpG first followed by the calculation of the arithmetic mean across features. For comparative analyses of multiple samples (within one figure panel) only features covered by all respective samples were used. The DNMT1i and EZH2i time course analyses are an exception. There, features covered by at least 80% of the samples were used to accommodate the large number of samples.

### Methylation entropy

Read-level DNA methylation statistics add another layer of information on top of the actual methylation rates per CpG with respect to cell population methylation heterogeneity. In the past, different metrics have been established to quantify heterogeneity across molecules based on single-read methylation patterns[37], and different groups, including our own, have developed tools to compute these metrics from high-throughput bisulfite sequencing data[37,40]. These read-level statistics include DNA methylation entropy, a measurement that is based on 4-mers of consecutive CpGs[38]. Entropy measures how heterogeneously each 4-mer is methylated on the basis of the patterns of so-called epialleles, which are generated from all reads that span the entire 4-mer. These epialleles represent the different possible configurations of methylated and unmethylated CpGs (16 epialleles possible for a 4-mer). If all reads show the same pattern across the four CpGs, the entropy would be equal to zero, while the entropy would be 1 if all 16 epialleles would be present at the same frequency. We can therefore use entropy as a potent indicator of dynamic turnover by measuring whether intermediate methylation in TSCs stems from the presence of several different subpopulations with specific methylation patterns (cellular heterogeneity) or whether each molecule is reflective of stochastically distributed methylated and unmethylated CpGs (allelic heterogeneity). Entropy per 4-mer of CpGs was calculated using RLM[40]. Mean entropy per CGI was calculated using the arithmetic mean. Only 4-mers covered by at least 10 and at most 150 reads were considered.

### Single cell sorted clone analysis

Methylation and entropy per 4-mer overlapping hyper CGIs was extracted and shown per ESC and TSC clone (Fig. 1d and Extended Data Fig. 2a–c). The in silico bulk was generated by adding the epiallele counts for all clones of one cell line and randomly subsampling 100 times from these epialleles using the average coverage per 4-mer across the clones. In each random sampling, the entropy and methylation of the 4-mer was calculated and the average per 4-mer across all sampling rounds was reported as the bulk value.

### Single-read analysis using nanopore data

Reads were overlapped with hyper CGIs (defined using the mouse ExE feature set), and only reads were retained that spanned at least two complete hyper CGIs. Hyper CGIs covered by fewer than ten reads were discarded from the analysis. For each pair of hyper CGIs, the number

of reads spanning both islands was extracted and used to calculate the fraction of concordant reads if at least ten reads spanned both islands. The fraction of concordant reads was calculated the following way: For each read $r$, the average methylation for both CGIs in the pair was calculated ($x_r, y_r$). These averages were compared with the median across the population for each CGI ($\bar{x}, \bar{y}$). The fraction of reads for each pair fulfilling

$$(x_r > \bar{x} \wedge y_r > \bar{y}) \vee (x_r \leq \bar{x} \wedge y_r \leq \bar{y})$$

was termed the fraction of concordant reads. To generate unphased, random control measurements, the average per read of the second CGI was randomly shuffled 100 times and each time the fraction of concordant reads was calculated. The average across all random samplings was reported per CGI pair.

### A/B compartment analysis

The first three eigenvectors were calculated using HiCExplorer's 'hicPCA' function using 100 kb resolution, KR-corrected matrices. For TSCs, compartments seemed to be defined by PC1 for all chromosomes while for ESCs compartments for chromosome 1 and 3 seemed to be represented by PC2 based on manual inspection. For the final compartment annotation, we picked the representative PC per chromosome and swapped the sign whenever applicable on the basis of gene density (positive for A compartment, negative for B compartment; higher gene density in A compartments expected). The A/B interaction ratio was calculated as described previously[104] using the $\log_2$ fold change of the average interaction frequency of each genomic 100 kb bin with other bins in A compared with B compartments based on the observed over expected matrix (generated using the function 'hicTransform' from HiCExplorer).

### Sequencing-based histone modification analysis

Enriched heat maps and profile plots of MINUTE-ChIP, ChIP–BS-seq and ChIP–seq signal were generated using the R package Enriched-Heatmap[136]. For this purpose, the signal was normalized to genomic features using the function 'normalizeToMatrix' (parameters: extend = c(5000, 5000), mean_mode = 'w0', w = 50, target_ratio = 0.25). The resulting data matrix was visualized using the function 'Enriched-Heatmap'. Density and scatter plots were generated by calculating the average signal across one kb genomic tiles or CGIs. Tiles were classified as overlapping with hyper CGIs (as defined on the basis of the ExE) if at least 20% of the CGI or 20% of the tile overlapped. For genome browser tracks, the RPGC signal was smoothed using 300 bp sliding windows.

### Read stack plot

To visualize the methylation rates of single ChIP–BS-seq reads (Fig. 3b and Extended Data Fig. 4c), the single read output from RLM was used (only reads spanning at least three CpGs were considered). Reads were coloured by their average methylation as reported by RLM and visualized by IGV limiting the number of reads shown to the first 20 rows[137].

### RNA-seq analysis

Transcripts per million (TPM) were obtained from the stringtie output. TPMs of replicates were averaged per gene. Correlation of RNA-seq samples was calculated using genes active in at least one averaged sample (TPM >2) based on the $\log_2$-transformed TPMs. Correlation and standardized expression values across samples were visualized using the R package pheatmap (ref. 138).

Differentially expressed genes were calculated using DEseq2 (ref. 139) considering only genes with a minimum total read count of 10 across all samples. Replicates for DNMT1i, EZH2i and double treatment were respectively tested against WT and all DMSO control replicates combined. Genes with an absolute $\log_2$ fold change >2 and an adjusted $P$ value <0.05 were termed differentially expressed. Only genes

active in at least two considered samples (TPM >2) were considered for downstream analyses.

### Overrepresentation analysis

Overrepresentation analysis of gene sets (up- and downregulated, hyper CGI associated, or proteins significantly interacting with EED as detected by MS) was conducted using WebGestaltR (parameters: minNum = 10, maxNum = 500, ref. 140) and the top 10 or 20 Gene Ontology (GO) terms for each set were visualized (for gene expression or MS, respectively).

### Repeat expression quantification

Global repeat expression quantification from RNA-seq was carried out as described previously[67]. Briefly, to estimate the expression for each retrotransposon subfamily without bias due to gene expression, only reads not overlapping any gene were considered for the analysis. Spliced reads as well as reads with a high poly-A content were also removed. The remaining reads were counted per subfamily only if they aligned uniquely or multiple times to elements of the same subfamily. Any annotated element of a specific subfamily from UCSC RepeatMasker was considered independent of our full-length IAP annotation. Reads aligning to multiple elements were only counted once. The overall read count per sample was then normalized by library size.

### Statistics and reproducibility

No statistical methods were used to pre-determine sample sizes, but our sample sizes are similar to those reported in previous publications[23,141–143]. Sample sizes are indicated in the figure panels or legends. No data were excluded. Four different TSC lines from two different labs and including female and male lines were profiled to confirm that TSCs exhibit an intermediate, stochastic methylome similar to that of the ExE. Single *Eed*, *Rnf2*, *Dnmt3b*, *Tet3* and *Kdm2b* KOs were generated (no replicates) and the effect of the *Eed* KO was verified using an inhibitor for EZH2. The effect of DNMT1i on TSCs was replicated in two different lines and three different experiments within the TSC1 line (different passages). MINUTE-ChIP experiments were performed in triplicate. RNA-seq, ChIP–BS-seq and EED ChIP–seq experiments were performed in duplicate. For RRBS, WGBS and nanopore experiments, single replicates per sample or timepoint were generated. Sex typing and western blots were repeated at least three times, and co-IP of EED and other proteins were repeated at least two times (one representative shown in this study). All attempts at replication were successful. Our genomic analyses are independent of human intervention. For experiments, no pre-selection was done on experimental versus control samples during culture, treatment, library synthesis or sequencing stages. Blinding was not relevant for this study since this is not an intervention study. However, our analytical pipeline followed uniform criteria applied to all samples, allowing us to analyse our data in an unbiased manner. All statistical tests were two-sided and were chosen as appropriate for data distribution.

### Reporting summary

Further information on research design is available in the Nature Portfolio Reporting Summary linked to this article.

## Data availability

Sequencing data that support the findings of this study have been deposited in the Gene Expression Omnibus (GEO) under accession code GSE166362. Previously published datasets that were re-analysed here are available under the following accession codes: WGBS datasets for WT as well as Polycomb KO mouse epiblast and ExE were obtained from GSE137337. WGBS for WT mESCs was used from GSE158460. ChIP–seq for H3K27me3 profiled in mESCs and respective input samples were obtained from GSE116603, GSE120376 and GSE49847 (refs. 144–146). mESC RNA-seq replicates are available under GSE159468. WGBS of E15

and E18 mouse placental tissue were obtained from GSE84350. Source data are provided with this paper or available at https://doi.org/10.5281/zenodo.7492144. Proteomics datasets have been deposited and are available at the ProteomeXchange Consortium under accession codes PXD039611 and PXD039719, and at the PeptideAtlas under accession codes PASS03804 and PASS03805. All other data supporting the findings of this study are available from the corresponding author on reasonable request.

## Code availability

Code is available at https://doi.org/10.5281/zenodo.7492144.

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

## Acknowledgements

We thank members of the Meissner and Smith laboratory for their support, in particular J. Batki, E. Lorentz, A. L. Mattei, C. Riemenschneider, S. Otto, M. Walther, R. Tornisie and J. Villagrana. We also thank B. Lukaszewska-McGreal for proteome sample preparation, the MPIMG Flow Cytometry Facility for assistance sorting cells, M. Zernicka-Goetz and her team for gifting a TSC line and the MPIMG Sequencing Core Facility, in particular S. Klages, M. Hochadel, N. Mages, S. Paturej and B. Timmermann. We thank R. Schöpflin for

helpful advice regarding Hi-C data processing and P. Giesselmann for assistance with Nanopore data processing. We thank R. McCarthy and K. S. Zaret for supporting initial ChIP–seq efforts. This work was supported by the German Federal Ministry for Research and Education (BMBF IntraEpiGliom, FKZ 13GW0347) (F.-J.M. and A.M.), the Deutsche Forschungsgemeinschaft (DFG, German Research Foundation) under Germany's Excellence Strategy (EXC 22167-390884018) (F.-J.M.), the Max Planck Society (A.M.) and NIH Early Innovators Award (DP2HD108774), the Mathers Foundation and the Chen Innovation Award (Z.D.S).

## Author contributions

R.W., S.H., Z.D.S. and A.M. designed and conceived the study. R.W., S.H. Z.D.S. and A.M. prepared the manuscript with the assistance of the other authors. R.W. performed all KO experiments. R.W. performed EZH2i and DNMT1i treatments with the help of N.B. S.H. performed initial processing of RRBS, WGBS, RNA-seq, Hi-C and Nanopore samples as well as downstream analyses of all sequencing datasets supervised by H.K. C.H. performed ChIP–BS-seq and EED ChIP–seq experiments. I.A.I. performed co-IPs supervised by T.A. P.Y.K.Y. did the MINUTE-ChIP experiments supervised by S.J.E. C.N. and S.J.E. performed the initial data processing and quality control for MINUTE-ChIP samples. R.W. and A.B. prepared RNA-seq libraries. A.S.K. derived TSC WT lines from mouse embryos and helped with imaging. C.A. prepared Hi-C libraries supervised by D.G.L. N.B. prepared the WGBS libraries. B.B. prepared nanopore libraries supervised by F.-J.M. D.M. performed MS experiments. Z.D.S and A.M. supervised the work.

## Funding

## Competing interests

A.M. and Z.D.S. are inventors on a patent related to hypermethylated CGI targets in cancer. Z.D.S. and A.M. are co-founders and scientific advisors of Harbinger Health. The remaining authors declare no competing interests.

## Additional information

**Extended data** is available for this paper at https://doi.org/10.1038/s41556-023-01114-y.

**Correspondence and requests for materials** should be addressed to Zachary D. Smith or Alexander Meissner.

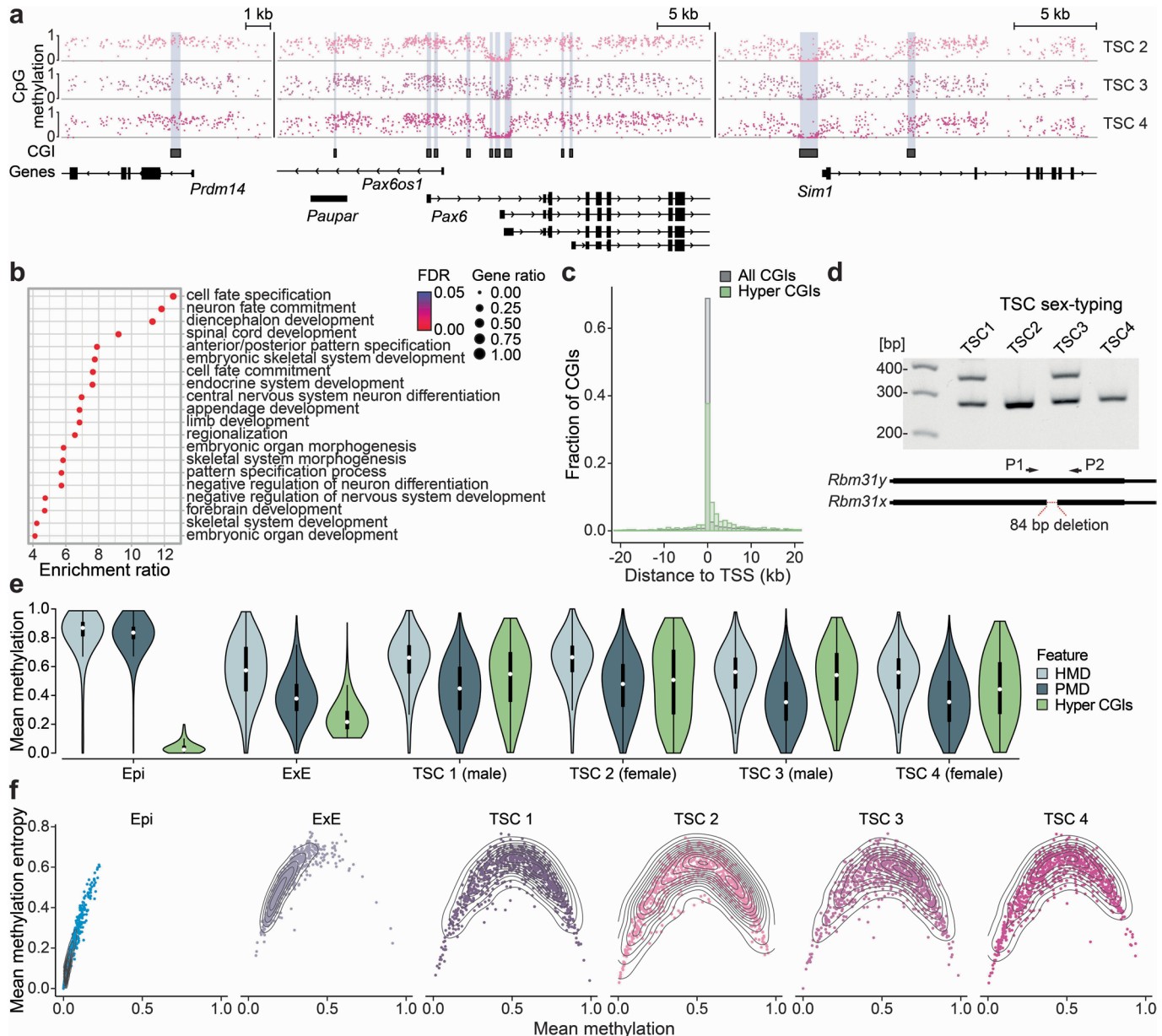

**Extended Data Fig. 1 | The intermediate methylome is stable and consistent across multiple TSC lines. a)** Genome browser tracks displaying CpG methylation for additional TSC lines. All lines exhibit a global decrease of methylation with select hypermethylation of CGIs to intermediate levels. **b)** Overrepresentation analysis of genes with hypermethylated CGI promoters in the ExE. Genes are enriched in developmental processes. **c)** Distribution of the distances to the nearest TSS for all CGIs and CGIs hypermethylated in the ExE. **d)** Genetic sex determination of wild type TSC lines 1-4 by simplex PCR. Primers differentiate X and Y chromosome homologues of the *Rbm31* gene. *Rbm31x* has an 84 bp deletion in comparison to *Rbm31y*. Amplicon size: *Rbm31x* = 269 bp, *Rbm31y* = 353 bp. **e)** Violin plots of average HMD, PMD and hyper CGI methylation

($n$ = 959,249, 954,783 and 1,102 features respectively) in epiblast, ExE and all TSC lines (single biological replicates for TSCs, two merged biological replicates for epiblast and ExE). White dots denote the median, edges denote the IQR and whiskers denote either 1.5 × IQR or minima/maxima (if no point exceeded 1.5 × IQR; minima/maxima are indicated by the violin plot range). **f)** Scatterplot showing the relationship between mean methylation entropy and mean CpG methylation at hypermethylated CGIs in epiblast, ExE and all TSC lines. CGIs are unmethylated in epiblast, which is associated with low entropy. Both ExE and TSC lines exhibit mostly intermediate methylation levels and high entropy (ExE shows comparatively lower intermediate methylation compared to TSCs).

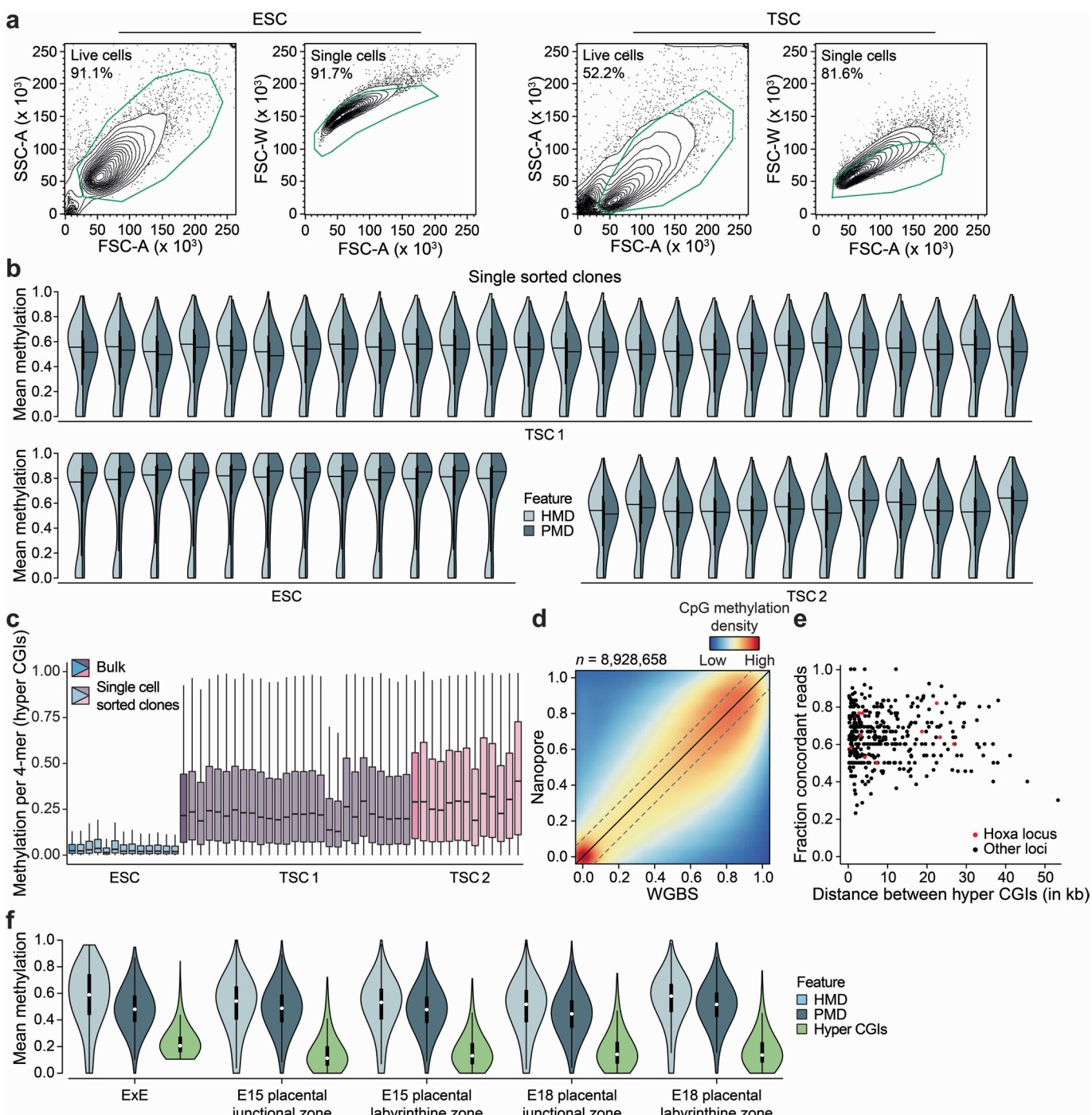

**Extended Data Fig. 2 | Single cell sorted clones re-gain bulk methylation levels and patterns. a)** Gating strategy for sorting single TSC and ESC clones. FSC-A/SSC-A was used to determine live cells, followed by FSC-A/FSC-W gating for single cells (1 cell/well of a 96 well plate). **b)** HMD and PMD violin plots (1 kb tiles, $n$ = 99,654 and 51,479 tiles respectively) for single sorted clones. Lines denote the median, edges the IQR and whiskers either 1.5 × IQR or minima/maxima (if no point exceeded 1.5 × IQR; minima/maxima are indicated by the violin plot range). **c)** 4-mer methylation ($n$ = 21,952) in hypermethylated CGIs for single cell-derived subclones (matching entropy boxplots in Fig. 1d). Subclones from the same cell type have similar methylation levels (low for ESCs, intermediate for TSCs), and resemble *in silico* generated averages. Lines denote the median, edges the IQR and whiskers either 1.5 × IQR or minima/maxima (if no point exceeded 1.5 × IQR; outliers were omitted). **d)** CpG-wise comparison of

WGBS and Nanopore methylation calls in the same TSC line (single biological replicates, ≥ 10x coverage in both). **e)** Relationship between CGI pair distance and the fraction of concordantly methylated reads (hypermethylated CGI pairs captured ≥10x). A read is termed concordant if paired CGIs both have methylation levels above or below their unphased averages. *Hoxa* locus CGI pairs (Fig. 1e) marked in red. The fraction of concordant reads does not appear to depend on distance between CGIs. **f)** Average HMD, PMD and hyper CGI methylation ($n$ = 274,371, 135,801 and 713 features respectively) values in epiblast, ExE and later placental tissues (E15, E18, two merged biological replicates for epiblast/ExE, six biological replicates for placental tissues). White dots denote the median, edges the IQR and whiskers either 1.5 × IQR or minima/maxima (if no point exceeded 1.5 × IQR; minima/maxima are indicated by the violin plot range). Data from Ref. 34.

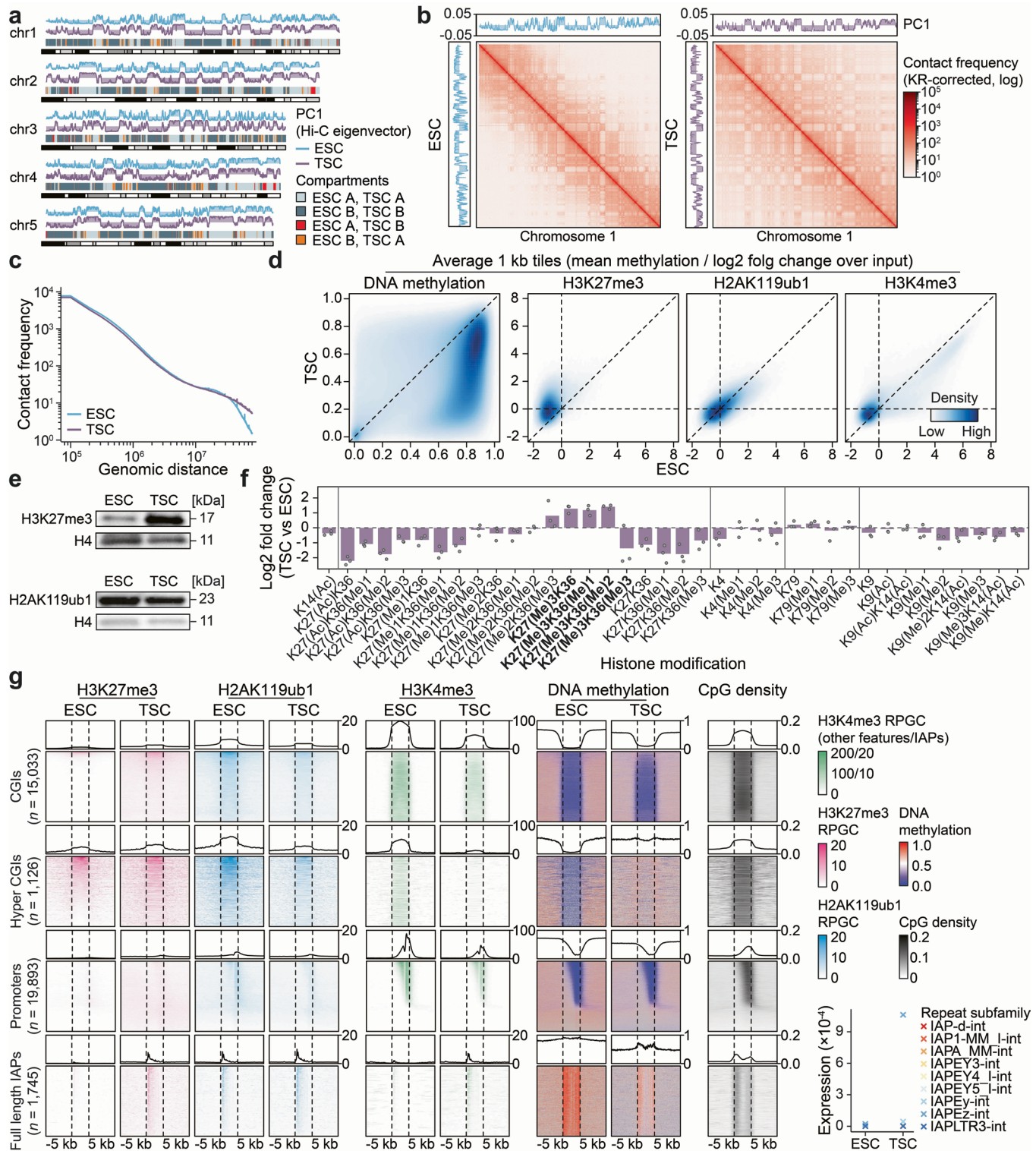

**Extended Data Fig. 3 | See next page for caption.**

**Extended Data Fig. 3 | H3K27me3 is globally enriched in TSCs compared to ESCs. a)** Compartments as defined by Hi-C (PC1 eigenvector, A > 0, B < 0) comparing ESCs and TSCs for the first five chromosomes. Few regions switch compartments and the overall distributions are highly comparable. **b)** Hi-C contact frequencies for ESCs and TSCs for chromosome 1 (100 kb bins), used to generate the comparative heatmap in Fig. 2a (two merged technical replicates per cell type). **c)** Comparison of contact frequencies across genomic distances between ESCs and TSCs. TSCs show an increase in very long-range contacts, but the effect is very small and imprecise. **d)** Density plots comparing DNA methylation and histone modification levels in one kb genomic tiles as measured using quantitative MINUTE-ChIP (log2 fold change over input, *n* = three merged biological replicates per cell type). TSC epigenomes are characterized by lower DNA methylation and higher K27me3. **e)** Western blot showing an increase of

H3K27me3 in TSCs compared to ESCs. **f)** log2 fold change of modified histone tails measured by mass spectrometry (*n* = three TSC biological replicates normalized against the mean of two biological ESC replicates). Histone tails carrying H3K27me3 are enriched in TSCs compared to ESCs, whereas unmodified K27 residues are depleted. **g)** Heatmaps of DNA methylation and histone signal across different genomic features for ESCs and TSCs. TSCs exhibit higher H3K27me3 levels across all feature groups including flanking genomic regions. ESCs show a higher H3K4me3 signal at protein-coding promoters. In contrast, TSCs show an increase of this active modification at full length IAP elements, which is accompanied by an increase in their expression (bottom right). Notably, H3K27me3 also appears to have specific enrichment at the promoters of these elements in TSCs, whereas H2AK119ub1 is present in both cell types. H2AK119ub1 levels appear to be increased in ESCs at CGIs and promoters.

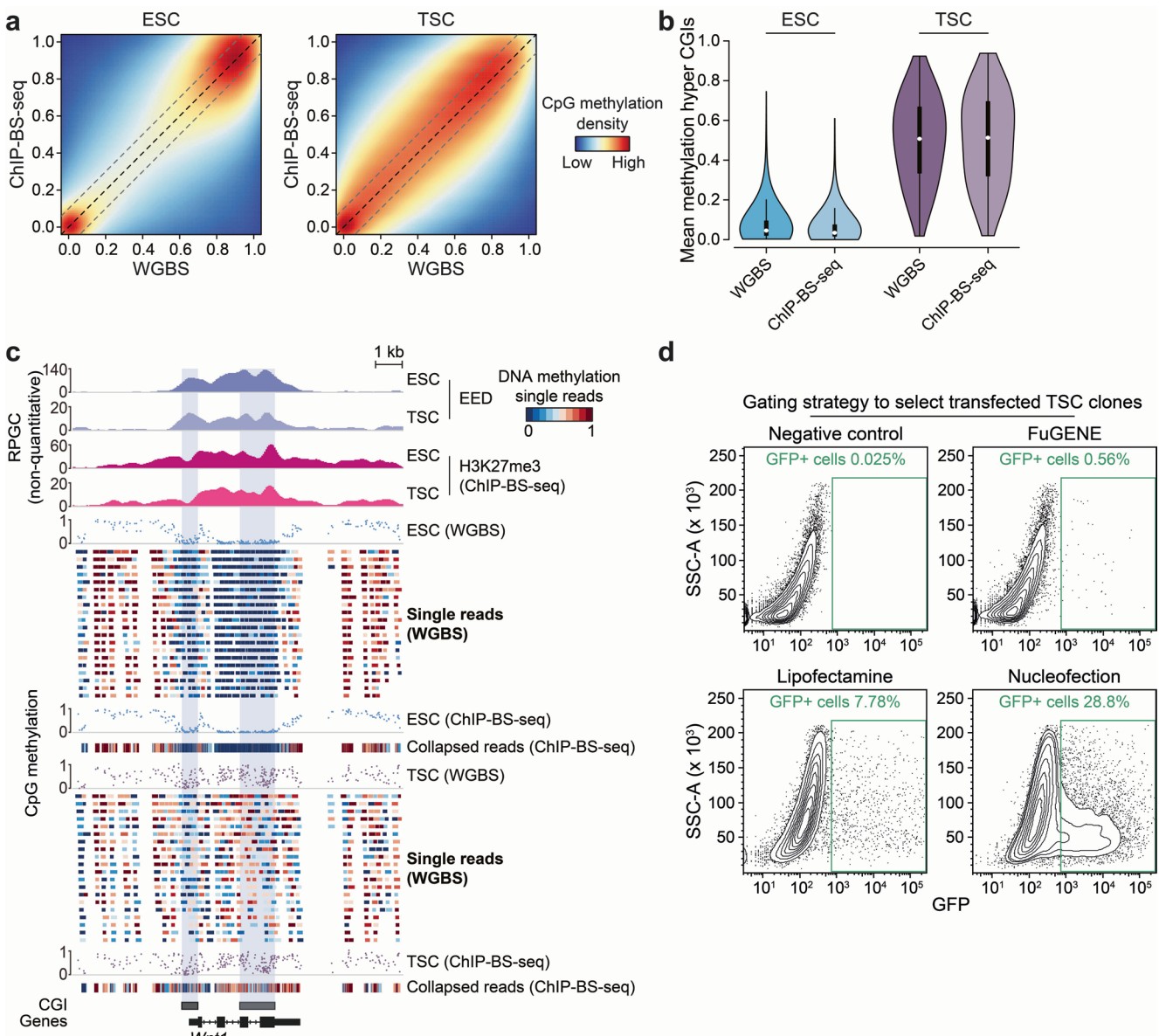

**Extended Data Fig. 4 | Intermediate DNA methylation and H3K27me3 co-occupy TSC chromatin. a)** CpG-wise comparison of ESCs and TSCs profiled with WGBS and ChIP-BS-seq ($n$ = two merged biological replicates for ChIP-BS-seq and single biological replicates for WGBS). **b)** Violin plots showing the methylation average of hyper CGIs in ESCs and TSCs as profiled by WGBS and ChIP-BS-seq. The high similarity between WGBS (unenriched background) and ChIP-BS-seq indicates that H3K27me3-modified nucleosomes carry intermediately methylated DNA as a steady state ($n$ = 939 CGIs). White dots denote the median, edges denote the IQR and whiskers denote either 1.5 × IQR or minima/maxima (if no point exceeded 1.5 × IQR; minima/maxima are indicated by the violin plot range). **c)** Genome browser track of the *Wnt1* locus in ESCs and TSCs showing EED localization and H3K27me3 (measured by ChIP-BS-seq) together with DNA methylation measured by WGBS and ChIP-BS-seq. Average methylation of single reads spanning at least three CpGs was visualized for WGBS using IGV (only the first 20 rows are shown). Read-level data expanded for the WGBS samples as a point of comparison for Fig. 3b. **d)** Gating strategy for selecting transfected clones for the TSC knockout lines. First, cells were gated according to the left panels to enrich for viable single cells, followed by sorting for GFP+ cells. WT TSCs were transfected with corresponding sgRNA/Cas9 plasmids expressing GFP. WT TSCs were used as negative control to set the GFP+ gate.

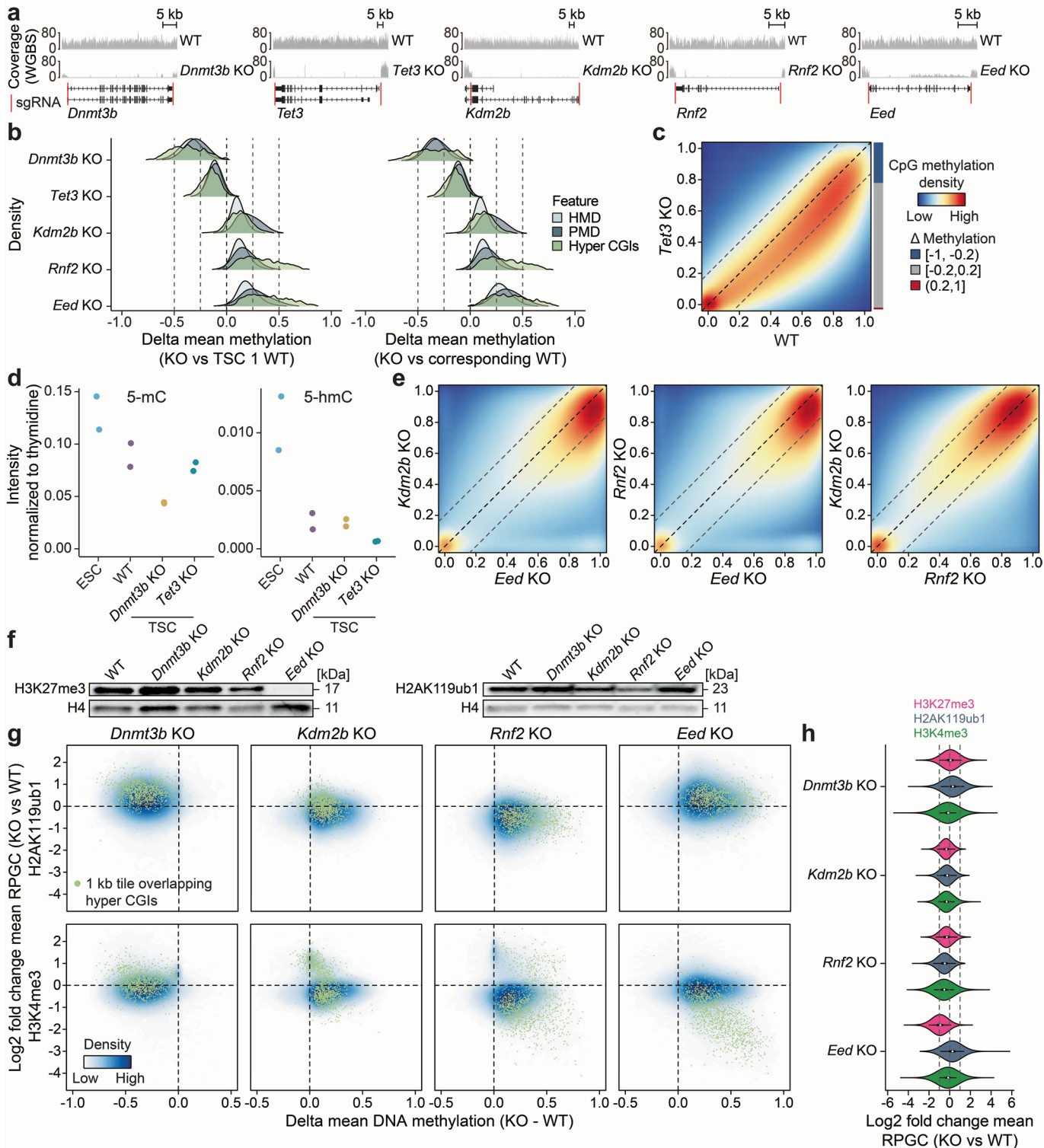

**Extended Data Fig. 5 | See next page for caption.**

**Extended Data Fig. 5 | Epigenome-wide shifts following the loss of epigenetic regulators. a)** Verification of the knockout strategy shown by read coverage of WGBS samples. **b)** Change in methylation for different knockout lines compared to wild type TSCs (left: TSC1, right: matching parental line, single biological replicates). **c)** CpG-wise density plot comparing *Tet3* KO with wild type TSCs show the overall similarity of these methylation landscapes (single biological replicates). **d)** 5-mC and 5-hmC levels as measured by Mass Spectrometry and normalized to thymidine, shown for ESCs as well as wild type, *Dnmt3b* KO and *Tet3* KO TSCs ($n$ = two independent biological samples, three technical replicates were conducted for each sample and averaged). These results confirm lower levels of both modifications in TSCs compared to ESCs, as well as the dependence of 5-mC on DNMT3B and 5-hmC on TET3. Overall, 5-hmC levels are lower in TSCs in comparison to ESCs even when accounting for lower global methylation levels in general (ratio of 5-hmC/5-mC = 8.7% in ESCs, 2.8% in TSCs). **e)** CpG-wise density plots comparing Polycomb (PRC) knockout TSCs. *Eed* KO triggers extreme genome-wide hypermethylation that is more pronounced compared to KOs of core or auxiliary PRC1 subunits (single biological replicates). **f)** Western blot showing H3K27me3 and H2AK119ub1 in WT and KO TSCs. **g)** Density plots depicting the relationship between DNA methylation (delta, single biological replicates) and either H2AK119ub1 or H3K4me3 (log2 fold change, three merged biological replicates) as they change between KO and WT TSCS (data is at one kb tile resolution). **h)** Log2 fold change for each histone modification in all TSC KOs compared to WT ($n$ = 1,700,932 one kb tiles, three merged biological replicates). White dots denote the median, edges the IQR and whiskers either 1.5 × IQR or minima/maxima (if no point exceeded 1.5 × IQR; minima/maxima are indicated by the violin plot range).

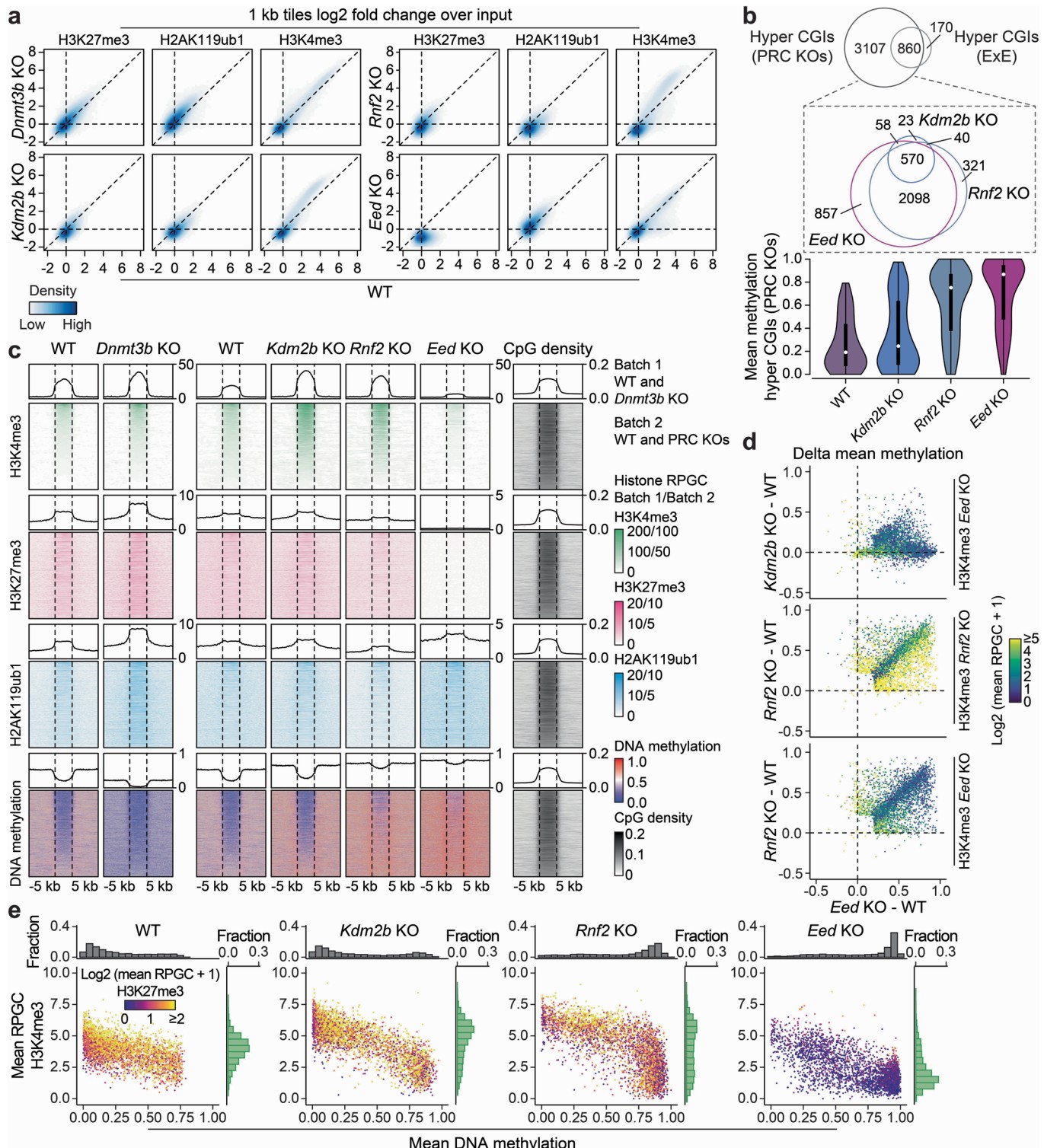

**Extended Data Fig. 6 | H3K4me3 shields CGIs from extreme hypermethylation. a)** Density plots comparing MINUTE-ChIP signal per one kb tile between WT and KO TSCs (log2 fold change over input, three merged biological replicates). **b)** Top: Overlap of CGIs hypermethylated in any PRC KO line (difference to WT > 0.2). *Kdm2b* KO cells show a diminished effect on CGI methylation in comparison to core regulators. Bottom: Mean methylation of the union of hypermethylated CGIs found in any of our PRC KOs (PRC hypermethylated CGIs, n = 3,967). White dots denote the median, edges the IQR and whiskers either 1.5 × IQR or minima/maxima (if no point exceeded 1.5 × IQR; minima/maxima are indicated by the violin plot range). **c)** Heatmaps of the histone modification and DNA methylation signal at CGIs hypermethylated in PRC KOs (matching the combined metaplots in Fig. 3h). Histone modifications

are quantitatively comparable as measured by MINUTE-ChIP within the same batch (*Dnmt3b* KO and PRC KOs were sequenced in two different batches and therefore each have a separate WT control, see Methods). **d)** Pairwise scatterplot comparing average delta methylation between PRC KOs with respect to the WT for PRC hypermethylated CGIs. Points are colored by H3K4me3 level in *Eed* KO (left and right) or *Rnf2* KO (mid) (log2-transformed). **e)** Scatterplot comparing mean methylation and H3K4me3 for PRC hypermethylated CGIs (samples all measured within the same MINUTE-ChIP batch). Histograms show the enrichment of CGIs for DNA methylation (x axis) and H3K4me3 (y axis), respectively. Color represents the average H3K27me3 signal per line (log2-transformed). DNA methylation increases from *Kdm2b* KO to *Eed* KO while H3K4me3 signal drops.

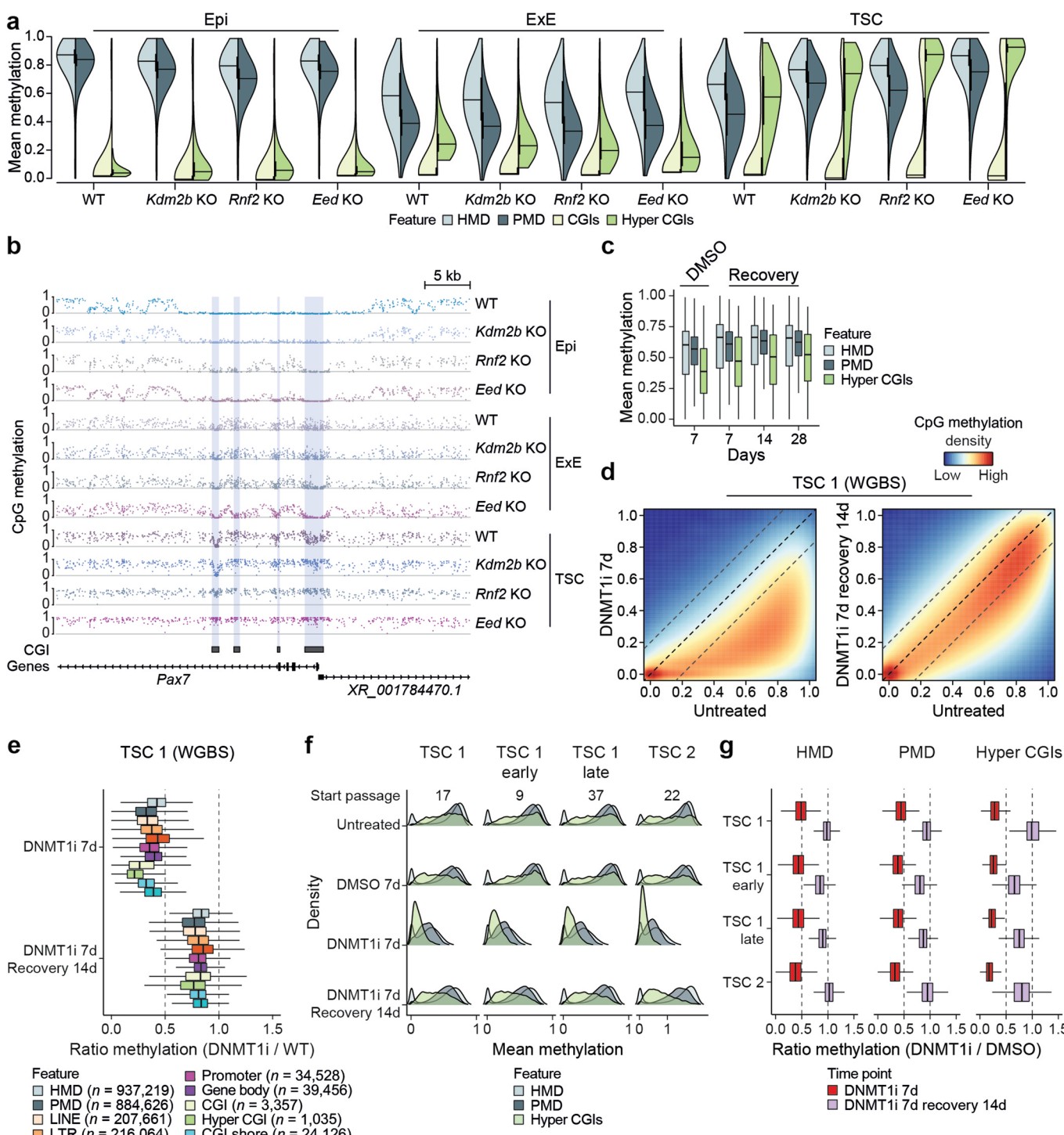

**Extended Data Fig. 7 | Differences between zygotic knockouts *in vivo* and acute knockouts in TSCs. a)** Feature-level violin plots for TSC and zygotic knockouts (n = 939,938 and 913,380 one kb tiles in HMDs and PMDs, n = 14,698 and 1,054 CGIs and hyper CGIs, two merged biological replicates for epiblast/ExE samples, single biological replicates otherwise). White dots denote the median, edges the IQR and whiskers either 1.5 × IQR or minima/maxima (if no point exceeded 1.5 × IQR; minima/maxima are indicated by the violin plot range). Data taken from Ref. 67. **b)** Genome browser track of the *Pax7* locus. **c)** Feature-level mean methylation across DMSO controls (RRBS time course in Fig. 5a, n = 117,477 and 62,118 one kb tiles in HMDs and PMDs, 970 hyper CGIs, single biological replicates). Lines denote the median, edges the IQR and whiskers either 1.5 × IQR or minima/maxima (if no point exceeded 1.5 × IQR; outliers omitted). **d)** CpG-wise density plot of DNMT1i-treated TSCs (left) and following withdrawal (right, single

biological replicates compared to WT, WGBS). **e)** Feature-level methylation changes after DNMT1i treatment or withdrawal (WGBS, features methylated > 0.2 in control were considered, single biological replicates, n = features). Lines denote the median, edges the IQR and whiskers either 1.5 × IQR or minima/maxima (if no point exceeded 1.5 × IQR; outliers omitted). **f)** Different TSC lines and passages show reproducible DNMT1i responses, highlighting the stability of this landscape over extended culture (feature n as in Extended Data Fig. 7c, single biological replicates). **g)** Methylation ratios between DNMT1i treatment or recovery (features methylated > 0.2 in DMSO control, n = 93,393 - 95,472 and 57,789 - 58,796 one kb tiles in HMDs and PMDs, 746 - 890 hyper CGIs, single biological replicates). Lines denote the median, edges the IQR and whiskers either 1.5 × IQR or minima/maxima (if no point exceeded 1.5 × IQR; outliers omitted).

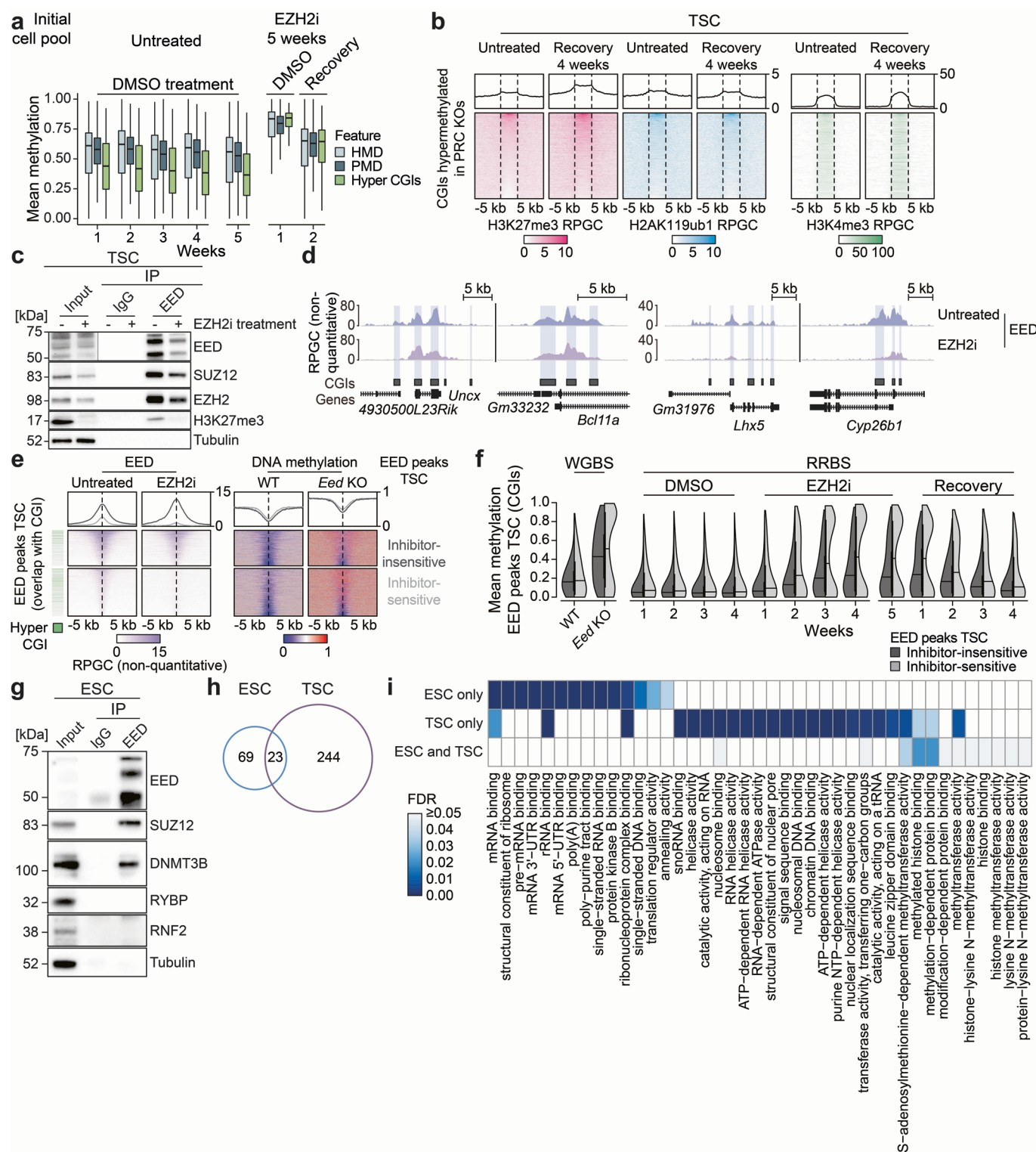

**Extended Data Fig. 8 | See next page for caption.**

**Extended Data Fig. 8 | Dynamic interactions between PRC2 and *de novo* methyltransferases. a)** Feature-level methylation across DMSO controls collected for Fig. 5b (*n* = 116,056 and 62,434 one kb tiles in HMDs and PMDs, 960 hyper CGIs, single biological replicates). X-axis breaks indicate different experiments (EZH2i treatment and DNMT1i pulse treatment). Lines denote the median, edges the IQR and whiskers either 1.5 × IQR or minima/maxima (if no point exceeded 1.5 × IQR; outliers were omitted). **b)** MINUTE-ChIP signal heatmaps for EZH2i-recovery and control TSCs (log2 fold change over input). Data is for PRC hypermethylated CGIS (see Extended Data Fig. 6b). H3K27me3 is fully regained after extensive periods of PRC2 inhibition. **c)** EED Co-IP of EZH2i-treated TSCs. EED is slightly downregulated and preserves interactions with core PRC2 components. Input lanes for the EED blot are taken from the same blot but shown for a higher exposure time given the intensity of the IP lanes. **d)** Representative genome browser tracks showing EED localization in -five week EZH2i-treated and control TSCs. Regions with strong EED enrichment maintain signal after EZH2i treatment whereas regions with low enrichment are generally depleted. **e)** EED signal heatmaps (ChIP-seq) in WT and EZH2i-treated TSCs, centered at EED peaks that overlap CGIs. DNA methylation in WT and *Eed* KO TSCs are also included. **f)** Methylation of inhibitor-insensitive and -sensitive EED peaks in WT and *Eed* KO TSCs (WGBS) as well as for our EZH2i experiments (RRBS, *n* = 2,868 inhibitor-sensitive and 2,202 -insensitive peaks). White dots denote the median, edges the IQR and whiskers either 1.5 × IQR or minima/maxima (if no point exceeded 1.5 × IQR; minima/maxima are indicated by the violin plot range). **g)** Co-IP of EED in WT ESCs. EED directly interacts with other components of PRC2 as well as DNMT3B, but not with PRC1 components. **h)** Overlap of significant EED interaction partners between ESCs and TSCs as determined by IP-MS. **i)** GO terms for significant EED interaction partners within ESCs and TSCs as determined by IP-MS.

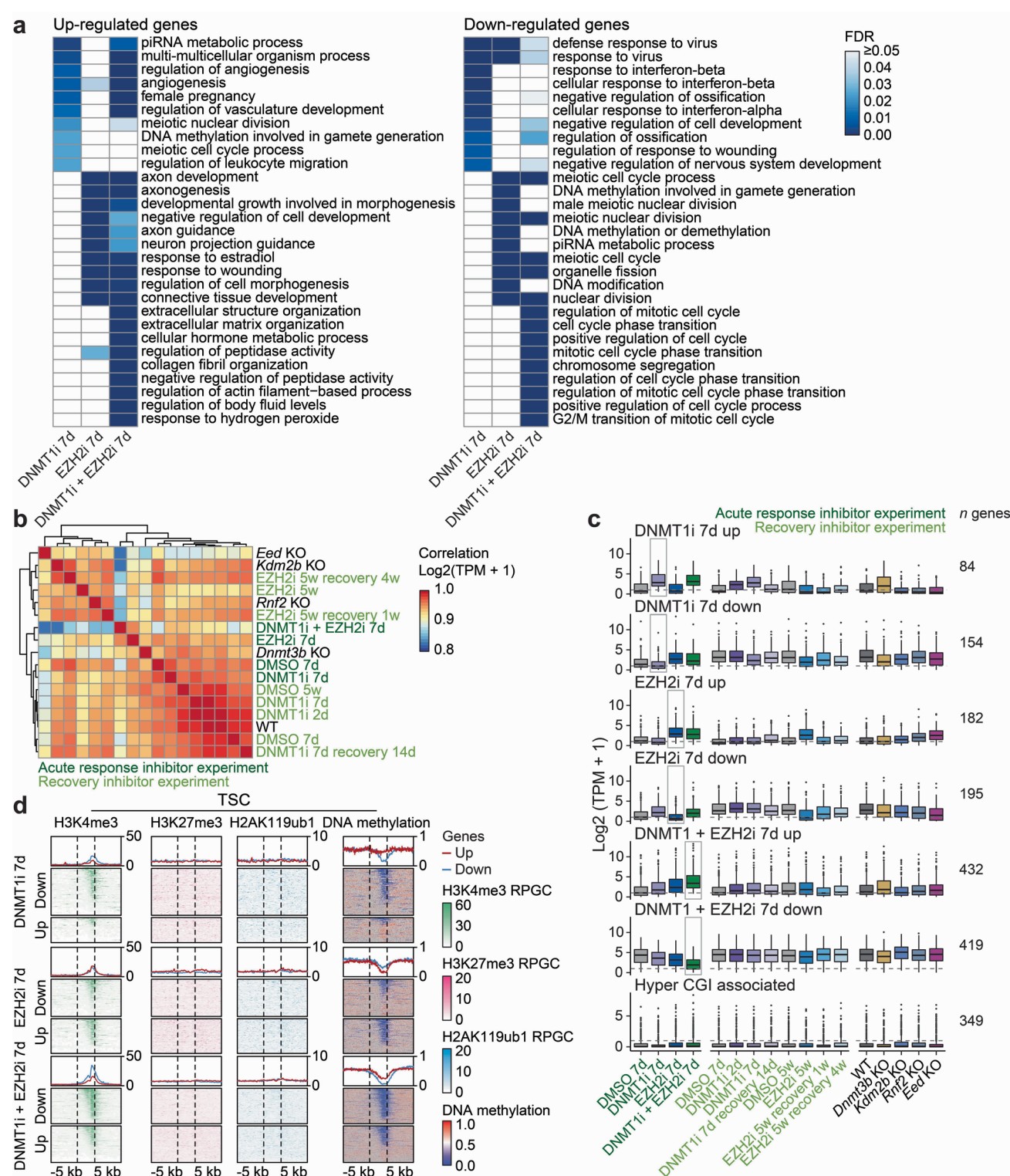

**Extended Data Fig. 9 | See next page for caption.**

**Extended Data Fig. 9 | Distinct transcriptional responses follow treatment with epigenetic inhibitors. a)** Overrepresented GO terms of biological processes for differentially up- or down-regulated gene sets following single and dual inhibitor treatments. Genes up-regulated upon DNMT1i treatment are enriched in germline-associated processes while genes up-regulated upon loss of H3K27me3 are associated with morphogenesis. Treatment with both inhibitors leads to a discrete response affecting genes involved in cell cycle regulation and chromosome segregation. **b)** Clustering of knockout, wild type and inhibitor samples based on their RNA-seq profiles (see Methods). **c)** Distribution of log2-transformed TPMs for specific gene sets (differentially expressed genes in our seven day inhibitor treatments or genes associated with hypermethylated CGIs, number of genes indicated in figure panel). *Eed* KO mimics the transcriptional response to the EZH2i treatment, as do our PRC1 knockouts. Loss of either or both repressive pathways does not lead to expression of genes associated with hypermethylated CGIs, although a subtle upward trend can be observed after double treatment. Lines denote the median, edges denote the IQR and whiskers denote either 1.5 × IQR and minima/maxima are represented by dots. **d)** Heatmaps of MINUTE-ChIP signal and DNA methylation in WT TSCs at significantly up- or down-regulated genes after 7 days of inhibitor treatment. Genes up-regulated after treatment with DNMT1i are mostly methylated in TSCs and become expressed after inhibitor-triggered loss of methylation. In contrast, neither EZH2i nor dual inhibitor treatment seem to affect the expression of genes with hypermethylated promoter CGIs. EZH2i sensitive genes show no substantial enrichment for H3K27me3, H2AK119ub1 or DNA methylation and therefore may be more indicative of indirect responses.

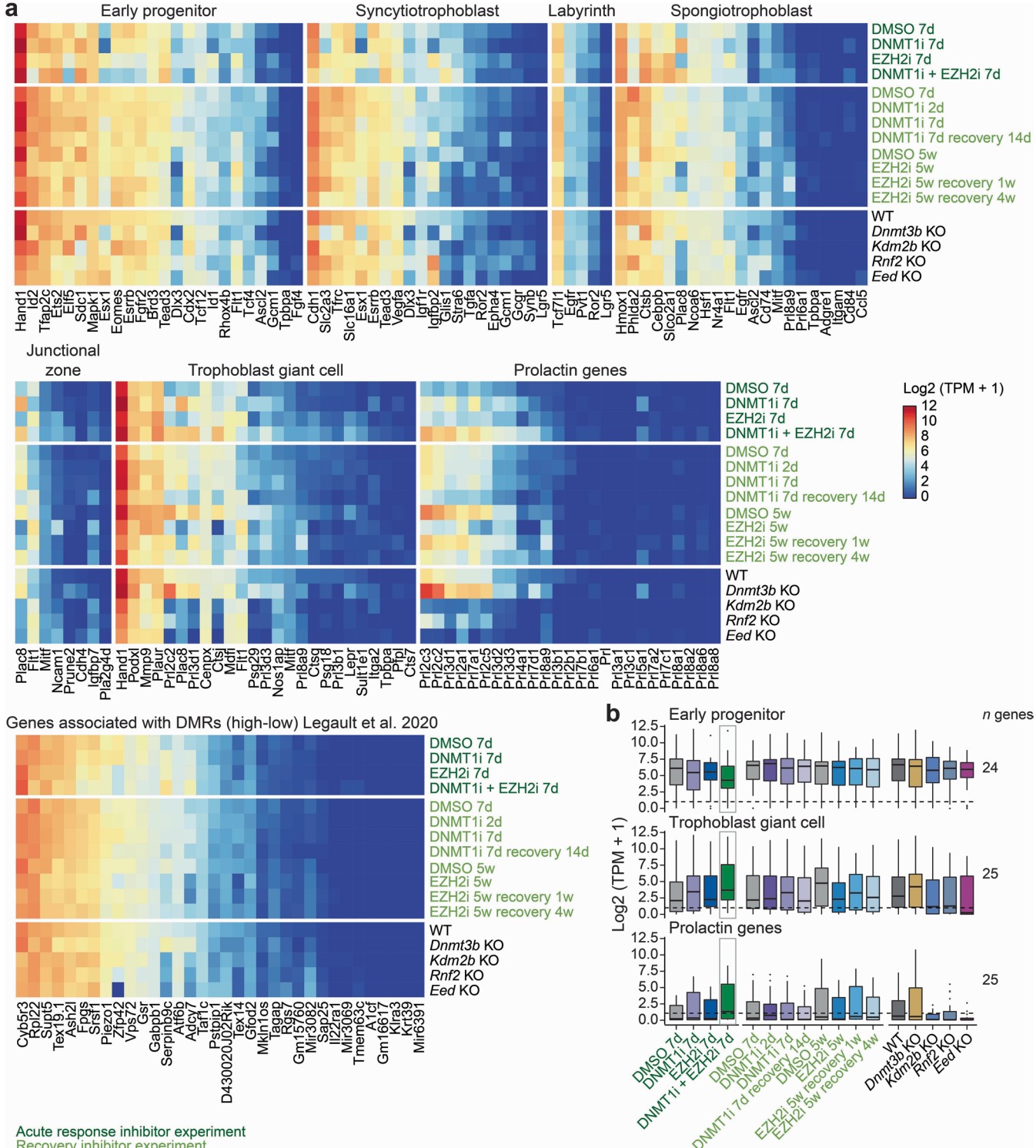

**Extended Data Fig. 10 | Examining the effects of disrupted epigenetic regulation on placental gene expression. a)** Heatmap of log2-transformed TPMs for marker gene sets specific to different placental cell types, including those associated with early progenitor states (trophoblast stem cells, the ExE and early chorion), as well as for the labyrinth, junctional and giant trophoblast lineages. Marker panels are collected from selected references and include those for the entire prolactin cluster and genes with shared gametogenic and placental functions[83–89]. Very minimal transcriptional changes are observed across these

sets, other than slight downregulation of progenitor markers and upregulation of giant cell markers when both DNA and PRC2 functions are dually inhibited. These signatures could easily be explained by low level spontaneous differentiation induced alongside rapid cell cycle arrest. **b)** Boxplot of log2-transformed TPMs for marker gene sets that exhibit subtle but notable dynamics, including those for progenitor, trophoblast giant cell and prolactin genes (number of genes indicated in figure panel). Lines denote the median, edges denote the IQR and whiskers denote 1.5 × IQR and minima/maxima are represented by dots.

# nature research

# Reporting Summary

Nature Research wishes to improve the reproducibility of the work that we publish. This form provides structure for consistency and transparency in reporting. For further information on Nature Research policies, see our Editorial Policies and the Editorial Policy Checklist.

## Statistics

For all statistical analyses, confirm that the following items are present in the figure legend, table legend, main text, or Methods section.

| n/a | Confirmed | |
|---|---|---|
| ☐ | ☒ | The exact sample size (*n*) for each experimental group/condition, given as a discrete number and unit of measurement |
| ☐ | ☒ | A statement on whether measurements were taken from distinct samples or whether the same sample was measured repeatedly |
| ☐ | ☒ | The statistical test(s) used AND whether they are one- or two-sided<br>*Only common tests should be described solely by name; describe more complex techniques in the Methods section.* |
| ☐ | ☒ | A description of all covariates tested |
| ☐ | ☒ | A description of any assumptions or corrections, such as tests of normality and adjustment for multiple comparisons |
| ☐ | ☒ | A full description of the statistical parameters including central tendency (e.g. means) or other basic estimates (e.g. regression coefficient) AND variation (e.g. standard deviation) or associated estimates of uncertainty (e.g. confidence intervals) |
| ☐ | ☒ | For null hypothesis testing, the test statistic (e.g. *F*, *t*, *r*) with confidence intervals, effect sizes, degrees of freedom and *P* value noted<br>*Give P values as exact values whenever suitable.* |
| ☒ | ☐ | For Bayesian analysis, information on the choice of priors and Markov chain Monte Carlo settings |
| ☒ | ☐ | For hierarchical and complex designs, identification of the appropriate level for tests and full reporting of outcomes |
| ☒ | ☐ | Estimates of effect sizes (e.g. Cohen's *d*, Pearson's *r*), indicating how they were calculated |

*Our web collection on statistics for biologists contains articles on many of the points above.*

## Software and code

Policy information about availability of computer code

| Data collection | cutadapt (version 2.4), BSMAP (version 2.90),GATK (version 4.1.4.1), MOABS (version 1.3.2), Nugene diversity adapter trimming (https://github.com/nugentechnologies/NuMetRRBS), NuDup (https://github.com/nugentechnologies/nudup), BWA (version 0.7.17), samtools (version 1.10), MACS2 (version 2.1.2), peakranger (version 1.18), deepTools (version 2.4.1), STAR (version 2.7.5a), stringtie (version 2.0.6), bowtie2 (version 2.3.5.1), HiCExplorer (version 3.6), Nanopype (version 1.1.0), guppy (v4.0.11), minimap2 (v2.10) , nanopolish (v0.13.2)<br><br>- Western blot images were collected using Image Lab software (version 6.1.0 build 7) (Bio-Rad)<br>- Single cell clones (ESC+TSC) and TSC knockout clones (GFP+) were sorted with FACS Diva software (BD Biosciences, v8.0.1) using BD FACS Aria II and BD FACSAria Fusion<br>- Brightfield images were collected using the Z1 Axio Observer and Zen Blue (2.3.69.1016) software (Zeiss) |
|---|---|
| Data analysis | R (version 3.6.3), metilene (version 0.2-8), bedtools (version 2.29.2), deepTools (version 2.4.1), pheatmap (version 1.0.12), EnrichedHeatmap (1.19.2), RLM (version 1.0.0), DESeq2 (version 1.26.0)<br>Custom code is available at 10.5281/zenodo.7492144.<br>- FACS data was analyzed with FlowJo (v10.7)<br>- GraphPad Prism (V 9.2.0) was used for the generation of bar plots and growth curves |

For manuscripts utilizing custom algorithms or software that are central to the research but not yet described in published literature, software must be made available to editors and reviewers. We strongly encourage code deposition in a community repository (e.g. GitHub). See the Nature Research guidelines for submitting code & software for further information.

## Data

Policy information about availability of data

All manuscripts must include a data availability statement. This statement should provide the following information, where applicable:
- Accession codes, unique identifiers, or web links for publicly available datasets
- A list of figures that have associated raw data
- A description of any restrictions on data availability

Sequencing data that support the findings of this study have been deposited in the Gene Expression Omnibus (GEO) under accession code GSE166362. Previously published data sets that were re-analyzed here are available under the following accession codes: WGBS data sets for wild type as well as Polycomb knockout mouse epiblast and extraembryonic ectoderm were obtained from GSE137337. WGBS for wild type mESCs was used from GSE158460. ChIP-seq for H3K27me3 profiled in mESCs and respective input samples were obtained from GSE116603, GSE120376 and GSE49847. mESC RNAseq replicates are available under GSE159468. WGBS of E15 and E18 mouse placental tissue were obtained from GSE84350. Source data are either provided with this study or available at 10.5281/zenodo.7492144. All other data supporting the findings of this study are available from the corresponding author on reasonable request.

# Field-specific reporting

Please select the one below that is the best fit for your research. If you are not sure, read the appropriate sections before making your selection.

☒ Life sciences        ☐ Behavioural & social sciences        ☐ Ecological, evolutionary & environmental sciences

For a reference copy of the document with all sections, see nature.com/documents/nr-reporting-summary-flat.pdf

# Life sciences study design

All studies must disclose on these points even when the disclosure is negative.

| | |
|---|---|
| Sample size | No statistical methods were used to predetermine sample sizes but our sample sizes are similar to those reported in previous publications (Smith et al. Nature 2017, Schoenfelder et al. Nature Communications 2018, Haggerty et al. Nature Structural and Molecular Biology 2021, Kumar et al. Nature Cell Biology 2022). Sample sizes are indicated in the figure panels or legends. |
| Data exclusions | No data was excluded. |
| Replication | Four different TSC lines from two different labs and including female and male lines were profiled in order to confirm that TSCs exhibit an intermediate, stochastic methylome similar to that of the extraembryonic ectoderm. Single Eed, Rnf2, Dnmt3b, Tet3 and Kdm2b knockouts were generated (no replicates) and the effect of the Eed knockout was verified using an inhibitor for EZH2. The effect of DNMT1i on TSCs was replicated in two different lines and three different experiments within the TSC1 line (at different passages). MINUTE-ChIP experiments were performed in triplicates. RNAseq, ChIP-BS-seq and EED ChIP-seq experiments were performed in duplicates. For RRBS, WGBS and Nanopore experiments single replicates per sample or time point were generated. Sex-typing and Western Blots were repeated at least three times, co-immunoprecipitation of EED and other proteins were repeated at least two times (one representative shown in this study). All attempts at replication were successful. |
| Randomization | Our genomic analyses are independent of human intervention and analyze each sample equally and in an unbiased fashion. For experiments, no pre-selection was done on experimental vs control samples during culture, treatment, library synthesis, or sequencing stages. |
| Blinding | Blinding was not relevant for this study since this is not an intervention study. However, our analytical pipeline followed uniform criteria applied to all samples, allowing us to analyze our data in an unbiased manner. |

# Reporting for specific materials, systems and methods

We require information from authors about some types of materials, experimental systems and methods used in many studies. Here, indicate whether each material, system or method listed is relevant to your study. If you are not sure if a list item applies to your research, read the appropriate section before selecting a response.

## Materials & experimental systems

| n/a | Involved in the study |
|---|---|
| ☐ | ☒ Antibodies |
| ☐ | ☒ Eukaryotic cell lines |
| ☒ | ☐ Palaeontology and archaeology |
| ☐ | ☒ Animals and other organisms |
| ☒ | ☐ Human research participants |
| ☒ | ☐ Clinical data |
| ☒ | ☐ Dual use research of concern |

## Methods

| n/a | Involved in the study |
|---|---|
| ☐ | ☒ ChIP-seq |
| ☐ | ☒ Flow cytometry |
| ☒ | ☐ MRI-based neuroimaging |

# Antibodies

**Antibodies used**

Supplier name, catalog number, clone/lot number
Western Blot and Co-IP/MS
Rabbit anti-H3K27me3: Cell Signaling Technology, #9733S, C36B11
Rabbit anti-H2AK119Ub1: Cell Signaling Technology, #8240S, D27C4
Rabbit anti-Histone H4: Cell Signaling Technology, #2592
Mouse anti-Rabbit-IgG HRP: Jackson, cat# 211-032-171, clone: 5A6-1D10
Rabbit anti-IgG: Cell Signaling Technology, #2729, lot:10
Rabbit anti-EED: Abcam, cat# ab4469, lot:GR3207387-1
Rabbit anti-Rnf2: Cell Signaling Technology, #5694, clone: D22F2
Rabbit anti-DNMT3B: Cell Signaling Technology, #48488S, clone: E4I40
Rabbit anti-RYBP: Millipore, #AB3637
Rabbit anti-Suz12: Cell Signaling Technology, #3737T, clone: 8
Mouse anti-Tubulin: Santa Cruz, #sc-32293, clone: DM1A
MINUTE ChIP:
Rabbit anti-H3K4me3: Millipore, #04-745, clone MC315
Rabbit anti-H3K27me3: Cell Signaling Technology, #9733, clone: C36B11
Rabbit anti-H2AK119ub: Cell Signaling Technology, #8240S, clone: D27C4
ChIP-BS-seq:
Rabbit anti-H3K27me3: Thermo Fisher, MA5-11198, lot: WH3366172
ChIP-seq:
Rabbit anti-EED: Abcam, ab240650, EPR23043-5, lot: GR3427609-2

**Validation**

Antibodies were validated by comparing immunofluorescence data and western blot in wild type and knockout trophoblast stem cells (data not shown). Additionally, all antibodies were validated by the provider and cited in numerous publications (information below)

The rabbit anti-H3K27me3 (CST) antibody has been validated using SimpleChIP® Enzymatic Chromatin IP Kits.
https://www.cellsignal.de/products/primary-antibodies/tri-methyl-histone-h3-lys27-c36b11-rabbit-mab/9733

The rabbit anti-H2AK119Ub1 antibody has been validated using SimpleChIP® Enzymatic Chromatin IP Kits.
https://www.cellsignal.com/products/primary-antibodies/ubiquityl-histone-h2a-lys119-d27c4-xp-rabbit-mab/8240

The rabbit anti-Histone H4 antibody has been validated by WB.
https://www.cellsignal.de/products/primary-antibodies/histone-h4-antibody/2592

The rabbit anti-IgG antibody has been validated by ChIP and IP.
https://www.cellsignal.de/products/primary-antibodies/normal-rabbit-igg/2729

The rabbit anti-EED (abcam, #ab4469) antibody has been validated for WB and ICC/IF. This antibody is additionally knockout validated.
https://www.abcam.com/eed-antibody-ab4469.html

The rabbit anti-Rnf2 antibody has been validated using SimpleChIP® Enzymatic Chromatin IP Kits.
https://www.cellsignal.de/products/primary-antibodies/ring1b-d22f2-xp-rabbit-mab/5694

The rabbit anti-DNMT3B antibody is highly specific and rigorously validated by the provider for ChIP, IP, WB and IF.
https://www.cellsignal.com/products/primary-antibodies/dnmt3b-e4i4o-rabbit-mab-mouse-specific/48488

The rabbit anti-RYBP antibody has been validated for use in WB.
https://www.merckmillipore.com/DE/de/product/Anti-DEDAF-Antibody,MM_NF-AB3637?ReferrerURL=https%3A%2F%2Fwww.google.com%2F

The rabbit anti-Suz12 antibody has been validated using SimpleChIP® Enzymatic Chromatin IP Kits.
https://www.cellsignal.com/products/primary-antibodies/suz12-d39f6-xp-rabbit-mab/3737

The mouse anti-Tubulin antibody has been validated for WB and IF.
https://www.scbt.com/p/alpha-tubulin-antibody-dm1a?productCanUrl=alpha-tubulin-antibody-dm1a&_requestid=1049815

The rabbit anti-H3K4me3 antibody has been validated for WB, ChIP, DB, Mplex and ChIP-seq.
https://www.merckmillipore.com/DE/de/product/Anti-trimethyl-Histone-H3-Lys4-Antibody-clone-MC315-rabbit-monoclonal,MM_NF-04-745

The rabbit anti-H3K27me3 (Thermo Fisher) antibody has been extensively validated for specificity using SNAP-ChIP™ spike-in and peptide array.
https://www.thermofisher.com/antibody/product/H3K27me3-Antibody-clone-G-299-10-Monoclonal/MA5-11198

The rabbit anti-EED (abcam, #ab240650) antibody has been validated in WB, ChIP-seq and IP. This antibody is additionally knockout validated.
https://www.abcam.com/eed-antibody-epr23043-5-chip-grade-ab240650.html

# Eukaryotic cell lines

Policy information about cell lines

| | |
|---|---|
| Cell line source(s) | Trophoblast Stem Cell (TSC) lines were derived from CD-1 strain blastocysts. Pronuclei Stage 3 (PN3) zygotes were isolated from natural mating of CD-1 strain mice (Charles river). More detailed info is provided in Methods under: "Derivation of Trophoblast stem cells".<br>V6.5 mouse embryonic stem cell line (mESCs), source: Konrad Hochedlinger lab |
| Authentication | The identity of V6.5 mESCs and mTSCs including all cell lines derived from them have been validated using morphological characteristics, immunofluorescence, marker gene expression (RNAseq), but have not been authenticated. Knockout cell lines were validated by genotyping, western blotting, Sanger sequencing and next generation sequencing (RNAseq/RRBS/WGBS). |
| Mycoplasma contamination | All cell lines are negative for mycoplasma contamination and were regularly tested throughout the study. |
| Commonly misidentified lines (See ICLAC register) | No commonly misidentified cell lines were used. |

# Animals and other organisms

Policy information about studies involving animals; ARRIVE guidelines recommended for reporting animal research

| | |
|---|---|
| Laboratory animals | CD-1 strain Mus musculus domesticus animals (male and female) were used to generate blastocysts for TSC derivation. |
| Wild animals | *Provide details on animals observed in or captured in the field; report species, sex and age where possible. Describe how animals were caught and transported and what happened to captive animals after the study (if killed, explain why and describe method; if released, say where and when) OR state that the study did not involve wild animals.* |
| Field-collected samples | *For laboratory work with field-collected samples, describe all relevant parameters such as housing, maintenance, temperature, photoperiod and end-of-experiment protocol OR state that the study did not involve samples collected from the field.* |
| Ethics oversight | All research described here complies with the relevant ethical regulations at the respective institutions. Work at the Max Planck Institute was approved by the LAGESO. |

Note that full information on the approval of the study protocol must also be provided in the manuscript.

# ChIP-seq

## Data deposition

☒ Confirm that both raw and final processed data have been deposited in a public database such as GEO.

☒ Confirm that you have deposited or provided access to graph files (e.g. BED files) for the called peaks.

| | |
|---|---|
| Data access links<br>*May remain private before publication.* | Datasets generated in this study have been deposited in the Gene Expression Omnibus under accession number GSE166362. |
| Files in database submission | MINUTE_ESC_WT_H2AK119ub1_Pool1_Rep1_R1.fastq.gz MINUTE_ESC_WT_H2AK119ub1_Pool1_Rep1_R2.fastq.gz<br>MINUTE_ESC_WT_H2AK119ub1_Pool1_Rep2_R1.fastq.gz MINUTE_ESC_WT_H2AK119ub1_Pool1_Rep2_R2.fastq.gz<br>MINUTE_ESC_WT_H2AK119ub1_Pool1_Rep3_R1.fastq.gz MINUTE_ESC_WT_H2AK119ub1_Pool1_Rep3_R2.fastq.gz<br>MINUTE_ESC_WT_H3K27me3_Pool1_Rep1_R1.fastq.gz MINUTE_ESC_WT_H3K27me3_Pool1_Rep1_R2.fastq.gz<br>MINUTE_ESC_WT_H3K27me3_Pool1_Rep2_R1.fastq.gz MINUTE_ESC_WT_H3K27me3_Pool1_Rep2_R2.fastq.gz<br>MINUTE_ESC_WT_H3K27me3_Pool1_Rep3_R1.fastq.gz MINUTE_ESC_WT_H3K27me3_Pool1_Rep3_R2.fastq.gz<br>MINUTE_ESC_WT_H3K4me3_Pool1_Rep1_R1.fastq.gz MINUTE_ESC_WT_H3K4me3_Pool1_Rep1_R2.fastq.gz<br>MINUTE_ESC_WT_H3K4me3_Pool1_Rep2_R1.fastq.gz MINUTE_ESC_WT_H3K4me3_Pool1_Rep2_R2.fastq.gz<br>MINUTE_ESC_WT_H3K4me3_Pool1_Rep3_R1.fastq.gz MINUTE_ESC_WT_H3K4me3_Pool1_Rep3_R2.fastq.gz<br>MINUTE_ESC_WT_Input_Pool1_Rep1_R1.fastq.gz MINUTE_ESC_WT_Input_Pool1_Rep1_R2.fastq.gz<br>MINUTE_ESC_WT_Input_Pool1_Rep2_R1.fastq.gz MINUTE_ESC_WT_Input_Pool1_Rep2_R2.fastq.gz<br>MINUTE_ESC_WT_Input_Pool1_Rep3_R1.fastq.gz MINUTE_ESC_WT_Input_Pool1_Rep3_R2.fastq.gz<br>MINUTE_TSC1_KDM2BKO_H2AK119ub1_Pool2_Rep1_R1.fastq.gz<br>MINUTE_TSC1_KDM2BKO_H2AK119ub1_Pool2_Rep1_R2.fastq.gz<br>MINUTE_TSC1_KDM2BKO_H2AK119ub1_Pool2_Rep2_R1.fastq.gz<br>MINUTE_TSC1_KDM2BKO_H2AK119ub1_Pool2_Rep2_R2.fastq.gz<br>MINUTE_TSC1_KDM2BKO_H2AK119ub1_Pool2_Rep3_R1.fastq.gz<br>MINUTE_TSC1_KDM2BKO_H2AK119ub1_Pool2_Rep3_R2.fastq.gz<br>MINUTE_TSC1_KDM2BKO_H3K27me3_Pool2_Rep1_R1.fastq.gz<br>MINUTE_TSC1_KDM2BKO_H3K27me3_Pool2_Rep1_R2.fastq.gz<br>MINUTE_TSC1_KDM2BKO_H3K27me3_Pool2_Rep2_R1.fastq.gz<br>MINUTE_TSC1_KDM2BKO_H3K27me3_Pool2_Rep2_R2.fastq.gz<br>MINUTE_TSC1_KDM2BKO_H3K27me3_Pool2_Rep3_R1.fastq.gz<br>MINUTE_TSC1_KDM2BKO_H3K27me3_Pool2_Rep3_R2.fastq.gz<br>MINUTE_TSC1_KDM2BKO_H3K4me3_Pool2_Rep1_R1.fastq.gz MINUTE_TSC1_KDM2BKO_H3K4me3_Pool2_Rep1_R2.fastq.gz |

```
MINUTE_TSC1_KDM2BKO_H3K4me3_Pool2_Rep2_R1.fastq.gz MINUTE_TSC1_KDM2BKO_H3K4me3_Pool2_Rep2_R2.fastq.gz
MINUTE_TSC1_KDM2BKO_H3K4me3_Pool2_Rep3_R1.fastq.gz MINUTE_TSC1_KDM2BKO_H3K4me3_Pool2_Rep3_R2.fastq.gz
MINUTE_TSC1_KDM2BKO_Input_Pool2_Rep1_R1.fastq.gz MINUTE_TSC1_KDM2BKO_Input_Pool2_Rep1_R2.fastq.gz
MINUTE_TSC1_KDM2BKO_Input_Pool2_Rep2_R1.fastq.gz MINUTE_TSC1_KDM2BKO_Input_Pool2_Rep2_R2.fastq.gz
MINUTE_TSC1_KDM2BKO_Input_Pool2_Rep3_R1.fastq.gz MINUTE_TSC1_KDM2BKO_Input_Pool2_Rep3_R2.fastq.gz
MINUTE_TSC1_RNF2KO_H2AK119ub1_Pool2_Rep1_R1.fastq.gz
MINUTE_TSC1_RNF2KO_H2AK119ub1_Pool2_Rep1_R2.fastq.gz
MINUTE_TSC1_RNF2KO_H2AK119ub1_Pool2_Rep2_R1.fastq.gz
MINUTE_TSC1_RNF2KO_H2AK119ub1_Pool2_Rep2_R2.fastq.gz
MINUTE_TSC1_RNF2KO_H2AK119ub1_Pool2_Rep3_R1.fastq.gz
MINUTE_TSC1_RNF2KO_H2AK119ub1_Pool2_Rep3_R2.fastq.gz
MINUTE_TSC1_RNF2KO_H3K27me3_Pool2_Rep1_R1.fastq.gz MINUTE_TSC1_RNF2KO_H3K27me3_Pool2_Rep1_R2.fastq.gz
MINUTE_TSC1_RNF2KO_H3K27me3_Pool2_Rep2_R1.fastq.gz MINUTE_TSC1_RNF2KO_H3K27me3_Pool2_Rep2_R2.fastq.gz
MINUTE_TSC1_RNF2KO_H3K27me3_Pool2_Rep3_R1.fastq.gz MINUTE_TSC1_RNF2KO_H3K27me3_Pool2_Rep3_R2.fastq.gz
MINUTE_TSC1_RNF2KO_H3K4me3_Pool2_Rep1_R1.fastq.gz MINUTE_TSC1_RNF2KO_H3K4me3_Pool2_Rep1_R2.fastq.gz
MINUTE_TSC1_RNF2KO_H3K4me3_Pool2_Rep2_R1.fastq.gz MINUTE_TSC1_RNF2KO_H3K4me3_Pool2_Rep2_R2.fastq.gz
MINUTE_TSC1_RNF2KO_H3K4me3_Pool2_Rep3_R1.fastq.gz MINUTE_TSC1_RNF2KO_H3K4me3_Pool2_Rep3_R2.fastq.gz
MINUTE_TSC1_RNF2KO_Input_Pool2_Rep1_R1.fastq.gz MINUTE_TSC1_RNF2KO_Input_Pool2_Rep1_R2.fastq.gz
MINUTE_TSC1_RNF2KO_Input_Pool2_Rep2_R1.fastq.gz MINUTE_TSC1_RNF2KO_Input_Pool2_Rep2_R2.fastq.gz
MINUTE_TSC1_RNF2KO_Input_Pool2_Rep3_R1.fastq.gz MINUTE_TSC1_RNF2KO_Input_Pool2_Rep3_R2.fastq.gz
MINUTE_TSC1_WT_EZH2i_5w_recovery_4w_H2AK119ub1_Pool2_Rep1_R1.fastq.gz
MINUTE_TSC1_WT_EZH2i_5w_recovery_4w_H2AK119ub1_Pool2_Rep1_R2.fastq.gz
MINUTE_TSC1_WT_EZH2i_5w_recovery_4w_H2AK119ub1_Pool2_Rep2_R1.fastq.gz
MINUTE_TSC1_WT_EZH2i_5w_recovery_4w_H2AK119ub1_Pool2_Rep2_R2.fastq.gz
MINUTE_TSC1_WT_EZH2i_5w_recovery_4w_H2AK119ub1_Pool2_Rep3_R1.fastq.gz
MINUTE_TSC1_WT_EZH2i_5w_recovery_4w_H2AK119ub1_Pool2_Rep3_R2.fastq.gz
MINUTE_TSC1_WT_EZH2i_5w_recovery_4w_H3K27me3_Pool2_Rep1_R1.fastq.gz
MINUTE_TSC1_WT_EZH2i_5w_recovery_4w_H3K27me3_Pool2_Rep1_R2.fastq.gz
MINUTE_TSC1_WT_EZH2i_5w_recovery_4w_H3K27me3_Pool2_Rep2_R1.fastq.gz
MINUTE_TSC1_WT_EZH2i_5w_recovery_4w_H3K27me3_Pool2_Rep2_R2.fastq.gz
MINUTE_TSC1_WT_EZH2i_5w_recovery_4w_H3K27me3_Pool2_Rep3_R1.fastq.gz
MINUTE_TSC1_WT_EZH2i_5w_recovery_4w_H3K27me3_Pool2_Rep3_R2.fastq.gz
MINUTE_TSC1_WT_EZH2i_5w_recovery_4w_H3K4me3_Pool2_Rep1_R1.fastq.gz
MINUTE_TSC1_WT_EZH2i_5w_recovery_4w_H3K4me3_Pool2_Rep1_R2.fastq.gz
MINUTE_TSC1_WT_EZH2i_5w_recovery_4w_H3K4me3_Pool2_Rep2_R1.fastq.gz
MINUTE_TSC1_WT_EZH2i_5w_recovery_4w_H3K4me3_Pool2_Rep2_R2.fastq.gz
MINUTE_TSC1_WT_EZH2i_5w_recovery_4w_H3K4me3_Pool2_Rep3_R1.fastq.gz
MINUTE_TSC1_WT_EZH2i_5w_recovery_4w_H3K4me3_Pool2_Rep3_R2.fastq.gz
MINUTE_TSC1_WT_EZH2i_5w_recovery_4w_Input_Pool2_Rep1_R1.fastq.gz
MINUTE_TSC1_WT_EZH2i_5w_recovery_4w_Input_Pool2_Rep1_R2.fastq.gz
MINUTE_TSC1_WT_EZH2i_5w_recovery_4w_Input_Pool2_Rep2_R1.fastq.gz
MINUTE_TSC1_WT_EZH2i_5w_recovery_4w_Input_Pool2_Rep2_R2.fastq.gz
MINUTE_TSC1_WT_EZH2i_5w_recovery_4w_Input_Pool2_Rep3_R1.fastq.gz
MINUTE_TSC1_WT_EZH2i_5w_recovery_4w_Input_Pool2_Rep3_R2.fastq.gz
MINUTE_TSC1_WT_H2AK119ub1_Pool1_Rep1_R1.fastq.gz MINUTE_TSC1_WT_H2AK119ub1_Pool1_Rep1_R2.fastq.gz
MINUTE_TSC1_WT_H2AK119ub1_Pool1_Rep2_R1.fastq.gz MINUTE_TSC1_WT_H2AK119ub1_Pool1_Rep2_R2.fastq.gz
MINUTE_TSC1_WT_H2AK119ub1_Pool1_Rep3_R1.fastq.gz MINUTE_TSC1_WT_H2AK119ub1_Pool1_Rep3_R2.fastq.gz
MINUTE_TSC1_WT_H2AK119ub1_Pool2_Rep1_R1.fastq.gz MINUTE_TSC1_WT_H2AK119ub1_Pool2_Rep1_R2.fastq.gz
MINUTE_TSC1_WT_H2AK119ub1_Pool2_Rep2_R1.fastq.gz MINUTE_TSC1_WT_H2AK119ub1_Pool2_Rep2_R2.fastq.gz
MINUTE_TSC1_WT_H2AK119ub1_Pool2_Rep3_R1.fastq.gz MINUTE_TSC1_WT_H2AK119ub1_Pool2_Rep3_R2.fastq.gz
MINUTE_TSC1_WT_H3K27me3_Pool1_Rep1_R1.fastq.gz MINUTE_TSC1_WT_H3K27me3_Pool1_Rep1_R2.fastq.gz
MINUTE_TSC1_WT_H3K27me3_Pool1_Rep2_R1.fastq.gz MINUTE_TSC1_WT_H3K27me3_Pool1_Rep2_R2.fastq.gz
MINUTE_TSC1_WT_H3K27me3_Pool1_Rep3_R1.fastq.gz MINUTE_TSC1_WT_H3K27me3_Pool1_Rep3_R2.fastq.gz
MINUTE_TSC1_WT_H3K27me3_Pool2_Rep1_R1.fastq.gz MINUTE_TSC1_WT_H3K27me3_Pool2_Rep1_R2.fastq.gz
MINUTE_TSC1_WT_H3K27me3_Pool2_Rep2_R1.fastq.gz MINUTE_TSC1_WT_H3K27me3_Pool2_Rep2_R2.fastq.gz
MINUTE_TSC1_WT_H3K27me3_Pool2_Rep3_R1.fastq.gz MINUTE_TSC1_WT_H3K27me3_Pool2_Rep3_R2.fastq.gz
MINUTE_TSC1_WT_H3K4me3_Pool1_Rep1_R1.fastq.gz MINUTE_TSC1_WT_H3K4me3_Pool1_Rep1_R2.fastq.gz
MINUTE_TSC1_WT_H3K4me3_Pool1_Rep2_R1.fastq.gz MINUTE_TSC1_WT_H3K4me3_Pool1_Rep2_R2.fastq.gz
MINUTE_TSC1_WT_H3K4me3_Pool1_Rep3_R1.fastq.gz MINUTE_TSC1_WT_H3K4me3_Pool1_Rep3_R2.fastq.gz
MINUTE_TSC1_WT_H3K4me3_Pool2_Rep1_R1.fastq.gz MINUTE_TSC1_WT_H3K4me3_Pool2_Rep1_R2.fastq.gz
MINUTE_TSC1_WT_H3K4me3_Pool2_Rep2_R1.fastq.gz MINUTE_TSC1_WT_H3K4me3_Pool2_Rep2_R2.fastq.gz
MINUTE_TSC1_WT_H3K4me3_Pool2_Rep3_R1.fastq.gz MINUTE_TSC1_WT_H3K4me3_Pool2_Rep3_R2.fastq.gz
MINUTE_TSC1_WT_Input_Pool1_Rep1_R1.fastq.gz MINUTE_TSC1_WT_Input_Pool1_Rep1_R2.fastq.gz
MINUTE_TSC1_WT_Input_Pool1_Rep2_R1.fastq.gz MINUTE_TSC1_WT_Input_Pool1_Rep2_R2.fastq.gz
MINUTE_TSC1_WT_Input_Pool1_Rep3_R1.fastq.gz MINUTE_TSC1_WT_Input_Pool1_Rep3_R2.fastq.gz
MINUTE_TSC1_WT_Input_Pool2_Rep1_R1.fastq.gz MINUTE_TSC1_WT_Input_Pool2_Rep1_R2.fastq.gz
MINUTE_TSC1_WT_Input_Pool2_Rep2_R1.fastq.gz MINUTE_TSC1_WT_Input_Pool2_Rep2_R2.fastq.gz
MINUTE_TSC1_WT_Input_Pool2_Rep3_R1.fastq.gz MINUTE_TSC1_WT_Input_Pool2_Rep3_R2.fastq.gz
MINUTE_TSC2_3BKO_H2AK119ub1_Pool1_Rep1_R1.fastq.gz MINUTE_TSC2_3BKO_H2AK119ub1_Pool1_Rep1_R2.fastq.gz
MINUTE_TSC2_3BKO_H2AK119ub1_Pool1_Rep2_R1.fastq.gz MINUTE_TSC2_3BKO_H2AK119ub1_Pool1_Rep2_R2.fastq.gz
MINUTE_TSC2_3BKO_H2AK119ub1_Pool1_Rep3_R1.fastq.gz MINUTE_TSC2_3BKO_H2AK119ub1_Pool1_Rep3_R2.fastq.gz
MINUTE_TSC2_3BKO_H3K27me3_Pool1_Rep1_R1.fastq.gz MINUTE_TSC2_3BKO_H3K27me3_Pool1_Rep1_R2.fastq.gz
MINUTE_TSC2_3BKO_H3K27me3_Pool1_Rep2_R1.fastq.gz MINUTE_TSC2_3BKO_H3K27me3_Pool1_Rep2_R2.fastq.gz
MINUTE_TSC2_3BKO_H3K27me3_Pool1_Rep3_R1.fastq.gz MINUTE_TSC2_3BKO_H3K27me3_Pool1_Rep3_R2.fastq.gz
MINUTE_TSC2_3BKO_H3K4me3_Pool1_Rep1_R1.fastq.gz MINUTE_TSC2_3BKO_H3K4me3_Pool1_Rep1_R2.fastq.gz
MINUTE_TSC2_3BKO_H3K4me3_Pool1_Rep2_R1.fastq.gz MINUTE_TSC2_3BKO_H3K4me3_Pool1_Rep2_R2.fastq.gz
```

```
MINUTE_TSC2_3BKO_H3K4me3_Pool1_Rep3_R1.fastq.gz MINUTE_TSC2_3BKO_H3K4me3_Pool1_Rep3_R2.fastq.gz
MINUTE_TSC2_3BKO_Input_Pool1_Rep1_R1.fastq.gz MINUTE_TSC2_3BKO_Input_Pool1_Rep1_R2.fastq.gz
MINUTE_TSC2_3BKO_Input_Pool1_Rep2_R1.fastq.gz MINUTE_TSC2_3BKO_Input_Pool1_Rep2_R2.fastq.gz
MINUTE_TSC2_3BKO_Input_Pool1_Rep3_R1.fastq.gz MINUTE_TSC2_3BKO_Input_Pool1_Rep3_R2.fastq.gz
MINUTE_TSC3_EEDKO_H2AK119ub1_Pool2_Rep1_R1.fastq.gz MINUTE_TSC3_EEDKO_H2AK119ub1_Pool2_Rep1_R2.fastq.gz
MINUTE_TSC3_EEDKO_H2AK119ub1_Pool2_Rep2_R1.fastq.gz MINUTE_TSC3_EEDKO_H2AK119ub1_Pool2_Rep2_R2.fastq.gz
MINUTE_TSC3_EEDKO_H2AK119ub1_Pool2_Rep3_R1.fastq.gz MINUTE_TSC3_EEDKO_H2AK119ub1_Pool2_Rep3_R2.fastq.gz
MINUTE_TSC3_EEDKO_H3K27me3_Pool2_Rep1_R1.fastq.gz MINUTE_TSC3_EEDKO_H3K27me3_Pool2_Rep1_R2.fastq.gz
MINUTE_TSC3_EEDKO_H3K27me3_Pool2_Rep2_R1.fastq.gz MINUTE_TSC3_EEDKO_H3K27me3_Pool2_Rep2_R2.fastq.gz
MINUTE_TSC3_EEDKO_H3K27me3_Pool2_Rep3_R1.fastq.gz MINUTE_TSC3_EEDKO_H3K27me3_Pool2_Rep3_R2.fastq.gz
MINUTE_TSC3_EEDKO_H3K4me3_Pool2_Rep1_R1.fastq.gz MINUTE_TSC3_EEDKO_H3K4me3_Pool2_Rep1_R2.fastq.gz
MINUTE_TSC3_EEDKO_H3K4me3_Pool2_Rep2_R1.fastq.gz MINUTE_TSC3_EEDKO_H3K4me3_Pool2_Rep2_R2.fastq.gz
MINUTE_TSC3_EEDKO_H3K4me3_Pool2_Rep3_R1.fastq.gz MINUTE_TSC3_EEDKO_H3K4me3_Pool2_Rep3_R2.fastq.gz
MINUTE_TSC3_EEDKO_Input_Pool2_Rep1_R1.fastq.gz MINUTE_TSC3_EEDKO_Input_Pool2_Rep1_R2.fastq.gz
MINUTE_TSC3_EEDKO_Input_Pool2_Rep2_R1.fastq.gz MINUTE_TSC3_EEDKO_Input_Pool2_Rep2_R2.fastq.gz
MINUTE_TSC3_EEDKO_Input_Pool2_Rep3_R1.fastq.gz MINUTE_TSC3_EEDKO_Input_Pool2_Rep3_R2.fastq.gz
MINUTE_ESC_WT_H2AK119ub1_Pool1_Rep1_RPGC_mm10_scaled.bw
MINUTE_ESC_WT_H2AK119ub1_Pool1_Rep2_RPGC_mm10_scaled.bw
MINUTE_ESC_WT_H2AK119ub1_Pool1_Rep3_RPGC_mm10_scaled.bw
MINUTE_ESC_WT_H3K27me3_Pool1_Rep1_RPGC_mm10_scaled.bw
MINUTE_ESC_WT_H3K27me3_Pool1_Rep2_RPGC_mm10_scaled.bw
MINUTE_ESC_WT_H3K27me3_Pool1_Rep3_RPGC_mm10_scaled.bw
MINUTE_ESC_WT_H3K4me3_Pool1_Rep1_RPGC_mm10_scaled.bw
MINUTE_ESC_WT_H3K4me3_Pool1_Rep2_RPGC_mm10_scaled.bw
MINUTE_ESC_WT_H3K4me3_Pool1_Rep3_RPGC_mm10_scaled.bw
MINUTE_ESC_WT_Input_Pool1_Rep1_RPGC_mm10_unscaled.bw
MINUTE_ESC_WT_Input_Pool1_Rep2_RPGC_mm10_unscaled.bw
MINUTE_ESC_WT_Input_Pool1_Rep3_RPGC_mm10_unscaled.bw
MINUTE_TSC1_KDM2BKO_H2AK119ub1_Pool2_Rep1_RPGC_mm10_scaled.bw
MINUTE_TSC1_KDM2BKO_H2AK119ub1_Pool2_Rep2_RPGC_mm10_scaled.bw
MINUTE_TSC1_KDM2BKO_H2AK119ub1_Pool2_Rep3_RPGC_mm10_scaled.bw
MINUTE_TSC1_KDM2BKO_H3K27me3_Pool2_Rep1_RPGC_mm10_scaled.bw
MINUTE_TSC1_KDM2BKO_H3K27me3_Pool2_Rep2_RPGC_mm10_scaled.bw
MINUTE_TSC1_KDM2BKO_H3K27me3_Pool2_Rep3_RPGC_mm10_scaled.bw
MINUTE_TSC1_KDM2BKO_H3K4me3_Pool2_Rep1_RPGC_mm10_scaled.bw
MINUTE_TSC1_KDM2BKO_H3K4me3_Pool2_Rep2_RPGC_mm10_scaled.bw
MINUTE_TSC1_KDM2BKO_H3K4me3_Pool2_Rep3_RPGC_mm10_scaled.bw
MINUTE_TSC1_KDM2BKO_Input_Pool2_Rep1_RPGC_mm10_unscaled.bw
MINUTE_TSC1_KDM2BKO_Input_Pool2_Rep2_RPGC_mm10_unscaled.bw
MINUTE_TSC1_KDM2BKO_Input_Pool2_Rep3_RPGC_mm10_unscaled.bw
MINUTE_TSC1_RNF2KO_H2AK119ub1_Pool2_Rep1_RPGC_mm10_scaled.bw
MINUTE_TSC1_RNF2KO_H2AK119ub1_Pool2_Rep2_RPGC_mm10_scaled.bw
MINUTE_TSC1_RNF2KO_H2AK119ub1_Pool2_Rep3_RPGC_mm10_scaled.bw
MINUTE_TSC1_RNF2KO_H3K27me3_Pool2_Rep1_RPGC_mm10_scaled.bw
MINUTE_TSC1_RNF2KO_H3K27me3_Pool2_Rep2_RPGC_mm10_scaled.bw
MINUTE_TSC1_RNF2KO_H3K27me3_Pool2_Rep3_RPGC_mm10_scaled.bw
MINUTE_TSC1_RNF2KO_H3K4me3_Pool2_Rep1_RPGC_mm10_scaled.bw
MINUTE_TSC1_RNF2KO_H3K4me3_Pool2_Rep2_RPGC_mm10_scaled.bw
MINUTE_TSC1_RNF2KO_H3K4me3_Pool2_Rep3_RPGC_mm10_scaled.bw
MINUTE_TSC1_RNF2KO_Input_Pool2_Rep1_RPGC_mm10_unscaled.bw
MINUTE_TSC1_RNF2KO_Input_Pool2_Rep2_RPGC_mm10_unscaled.bw
MINUTE_TSC1_RNF2KO_Input_Pool2_Rep3_RPGC_mm10_unscaled.bw
MINUTE_TSC1_WT_EZH2i_5w_recovery_4w_H2AK119ub1_Pool2_Rep1_RPGC_mm10_scaled.bw
MINUTE_TSC1_WT_EZH2i_5w_recovery_4w_H2AK119ub1_Pool2_Rep2_RPGC_mm10_scaled.bw
MINUTE_TSC1_WT_EZH2i_5w_recovery_4w_H2AK119ub1_Pool2_Rep3_RPGC_mm10_scaled.bw
MINUTE_TSC1_WT_EZH2i_5w_recovery_4w_H3K27me3_Pool2_Rep1_RPGC_mm10_scaled.bw
MINUTE_TSC1_WT_EZH2i_5w_recovery_4w_H3K27me3_Pool2_Rep2_RPGC_mm10_scaled.bw
MINUTE_TSC1_WT_EZH2i_5w_recovery_4w_H3K27me3_Pool2_Rep3_RPGC_mm10_scaled.bw
MINUTE_TSC1_WT_EZH2i_5w_recovery_4w_H3K4me3_Pool2_Rep1_RPGC_mm10_scaled.bw
MINUTE_TSC1_WT_EZH2i_5w_recovery_4w_H3K4me3_Pool2_Rep2_RPGC_mm10_scaled.bw
MINUTE_TSC1_WT_EZH2i_5w_recovery_4w_H3K4me3_Pool2_Rep3_RPGC_mm10_scaled.bw
MINUTE_TSC1_WT_EZH2i_5w_recovery_4w_Input_Pool2_Rep1_RPGC_mm10_unscaled.bw
MINUTE_TSC1_WT_EZH2i_5w_recovery_4w_Input_Pool2_Rep2_RPGC_mm10_unscaled.bw
MINUTE_TSC1_WT_EZH2i_5w_recovery_4w_Input_Pool2_Rep3_RPGC_mm10_unscaled.bw
MINUTE_TSC1_WT_H2AK119ub1_Pool1_Rep1_RPGC_mm10_scaled.bw
MINUTE_TSC1_WT_H2AK119ub1_Pool1_Rep2_RPGC_mm10_scaled.bw
MINUTE_TSC1_WT_H2AK119ub1_Pool1_Rep3_RPGC_mm10_scaled.bw
MINUTE_TSC1_WT_H2AK119ub1_Pool2_Rep1_RPGC_mm10_scaled.bw
MINUTE_TSC1_WT_H2AK119ub1_Pool2_Rep2_RPGC_mm10_scaled.bw
MINUTE_TSC1_WT_H2AK119ub1_Pool2_Rep3_RPGC_mm10_scaled.bw
MINUTE_TSC1_WT_H3K27me3_Pool1_Rep1_RPGC_mm10_scaled.bw
MINUTE_TSC1_WT_H3K27me3_Pool1_Rep2_RPGC_mm10_scaled.bw
MINUTE_TSC1_WT_H3K27me3_Pool1_Rep3_RPGC_mm10_scaled.bw
MINUTE_TSC1_WT_H3K27me3_Pool2_Rep1_RPGC_mm10_scaled.bw
MINUTE_TSC1_WT_H3K27me3_Pool2_Rep2_RPGC_mm10_scaled.bw
MINUTE_TSC1_WT_H3K27me3_Pool2_Rep3_RPGC_mm10_scaled.bw
```

```
MINUTE_TSC1_WT_H3K4me3_Pool1_Rep1_RPGC_mm10_scaled.bw
MINUTE_TSC1_WT_H3K4me3_Pool1_Rep2_RPGC_mm10_scaled.bw
MINUTE_TSC1_WT_H3K4me3_Pool1_Rep3_RPGC_mm10_scaled.bw
MINUTE_TSC1_WT_H3K4me3_Pool2_Rep1_RPGC_mm10_scaled.bw
MINUTE_TSC1_WT_H3K4me3_Pool2_Rep2_RPGC_mm10_scaled.bw
MINUTE_TSC1_WT_H3K4me3_Pool2_Rep3_RPGC_mm10_scaled.bw
MINUTE_TSC1_WT_Input_Pool1_Rep1_RPGC_mm10_unscaled.bw
MINUTE_TSC1_WT_Input_Pool1_Rep2_RPGC_mm10_unscaled.bw
MINUTE_TSC1_WT_Input_Pool1_Rep3_RPGC_mm10_unscaled.bw
MINUTE_TSC1_WT_Input_Pool2_Rep1_RPGC_mm10_unscaled.bw
MINUTE_TSC1_WT_Input_Pool2_Rep2_RPGC_mm10_unscaled.bw
MINUTE_TSC1_WT_Input_Pool2_Rep3_RPGC_mm10_unscaled.bw
MINUTE_TSC2_3BKO_H2AK119ub1_Pool1_Rep1_RPGC_mm10_scaled.bw
MINUTE_TSC2_3BKO_H2AK119ub1_Pool1_Rep2_RPGC_mm10_scaled.bw
MINUTE_TSC2_3BKO_H2AK119ub1_Pool1_Rep3_RPGC_mm10_scaled.bw
MINUTE_TSC2_3BKO_H3K27me3_Pool1_Rep1_RPGC_mm10_scaled.bw
MINUTE_TSC2_3BKO_H3K27me3_Pool1_Rep2_RPGC_mm10_scaled.bw
MINUTE_TSC2_3BKO_H3K27me3_Pool1_Rep3_RPGC_mm10_scaled.bw
MINUTE_TSC2_3BKO_H3K4me3_Pool1_Rep1_RPGC_mm10_scaled.bw
MINUTE_TSC2_3BKO_H3K4me3_Pool1_Rep2_RPGC_mm10_scaled.bw
MINUTE_TSC2_3BKO_H3K4me3_Pool1_Rep3_RPGC_mm10_scaled.bw
MINUTE_TSC2_3BKO_Input_Pool1_Rep1_RPGC_mm10_unscaled.bw
MINUTE_TSC2_3BKO_Input_Pool1_Rep2_RPGC_mm10_unscaled.bw
MINUTE_TSC2_3BKO_Input_Pool1_Rep3_RPGC_mm10_unscaled.bw
MINUTE_TSC3_EEDKO_H2AK119ub1_Pool2_Rep1_RPGC_mm10_scaled.bw
MINUTE_TSC3_EEDKO_H2AK119ub1_Pool2_Rep2_RPGC_mm10_scaled.bw
MINUTE_TSC3_EEDKO_H2AK119ub1_Pool2_Rep3_RPGC_mm10_scaled.bw
MINUTE_TSC3_EEDKO_H3K27me3_Pool2_Rep1_RPGC_mm10_scaled.bw
MINUTE_TSC3_EEDKO_H3K27me3_Pool2_Rep2_RPGC_mm10_scaled.bw
MINUTE_TSC3_EEDKO_H3K27me3_Pool2_Rep3_RPGC_mm10_scaled.bw
MINUTE_TSC3_EEDKO_H3K4me3_Pool2_Rep1_RPGC_mm10_scaled.bw
MINUTE_TSC3_EEDKO_H3K4me3_Pool2_Rep2_RPGC_mm10_scaled.bw
MINUTE_TSC3_EEDKO_H3K4me3_Pool2_Rep3_RPGC_mm10_scaled.bw
MINUTE_TSC3_EEDKO_Input_Pool2_Rep1_RPGC_mm10_unscaled.bw
MINUTE_TSC3_EEDKO_Input_Pool2_Rep2_RPGC_mm10_unscaled.bw
MINUTE_TSC3_EEDKO_Input_Pool2_Rep3_RPGC_mm10_unscaled.bw
MINUTE_ESC_WT_H2AK119ub1_Pool1_merged_mm10_scaled.bw
MINUTE_ESC_WT_H3K27me3_Pool1_merged_mm10_scaled.bw
MINUTE_ESC_WT_H3K4me3_Pool1_merged_mm10_scaled.bw
MINUTE_ESC_WT_Input_Pool1_merged_RPGC_mm10_unscaled.bw
MINUTE_TSC1_KDM2BKO_H2AK119ub1_Pool2_merged_mm10_scaled.bw
MINUTE_TSC1_KDM2BKO_H3K27me3_Pool2_merged_mm10_scaled.bw
MINUTE_TSC1_KDM2BKO_H3K4me3_Pool2_merged_mm10_scaled.bw
MINUTE_TSC1_KDM2BKO_Input_Pool2_merged_RPGC_mm10_unscaled.bw
MINUTE_TSC1_RNF2KO_H2AK119ub1_Pool2_merged_mm10_scaled.bw
MINUTE_TSC1_RNF2KO_H3K27me3_Pool2_merged_mm10_scaled.bw
MINUTE_TSC1_RNF2KO_H3K4me3_Pool2_merged_mm10_scaled.bw
MINUTE_TSC1_RNF2KO_Input_Pool2_merged_RPGC_mm10_unscaled.bw
MINUTE_TSC1_WT_EZH2i_5w_recovery_4w_H2AK119ub1_Pool2_merged_mm10_scaled.bw
MINUTE_TSC1_WT_EZH2i_5w_recovery_4w_H3K27me3_Pool2_merged_mm10_scaled.bw
MINUTE_TSC1_WT_EZH2i_5w_recovery_4w_H3K4me3_Pool2_merged_mm10_scaled.bw
MINUTE_TSC1_WT_EZH2i_5w_recovery_4w_Input_Pool2_merged_RPGC_mm10_unscaled.bw
MINUTE_TSC1_WT_H2AK119ub1_Pool1_merged_mm10_scaled.bw
MINUTE_TSC1_WT_H2AK119ub1_Pool2_merged_mm10_scaled.bw
MINUTE_TSC1_WT_H3K27me3_Pool1_merged_mm10_scaled.bw
MINUTE_TSC1_WT_H3K27me3_Pool2_merged_mm10_scaled.bw
MINUTE_TSC1_WT_H3K4me3_Pool1_merged_mm10_scaled.bw
MINUTE_TSC1_WT_H3K4me3_Pool2_merged_mm10_scaled.bw
MINUTE_TSC1_WT_Input_Pool1_merged_RPGC_mm10_unscaled.bw
MINUTE_TSC1_WT_Input_Pool2_merged_RPGC_mm10_unscaled.bw
MINUTE_TSC2_3BKO_H2AK119ub1_Pool1_merged_mm10_scaled.bw
MINUTE_TSC2_3BKO_H3K27me3_Pool1_merged_mm10_scaled.bw
MINUTE_TSC2_3BKO_H3K4me3_Pool1_merged_mm10_scaled.bw
MINUTE_TSC2_3BKO_Input_Pool1_merged_RPGC_mm10_unscaled.bw
MINUTE_TSC3_EEDKO_H2AK119ub1_Pool2_merged_mm10_scaled.bw
MINUTE_TSC3_EEDKO_H3K27me3_Pool2_merged_mm10_scaled.bw
MINUTE_TSC3_EEDKO_H3K4me3_Pool2_merged_mm10_scaled.bw
MINUTE_TSC3_EEDKO_Input_Pool2_merged_RPGC_mm10_unscaled.bw
ChIP-BS-seq_ESC_WT_H3K27me3_Input_Rep1_R1.fastq.gz ChIP-BS-seq_ESC_WT_H3K27me3_Input_Rep1_R2.fastq.gz
ChIP-BS-seq_ESC_WT_H3K27me3_Input_Rep2_R1.fastq.gz ChIP-BS-seq_ESC_WT_H3K27me3_Input_Rep2_R2.fastq.gz
ChIP-BS-seq_ESC_WT_H3K27me3_Rep1_R1.fastq.gz ChIP-BS-seq_ESC_WT_H3K27me3_Rep1_R2.fastq.gz
ChIP-BS-seq_ESC_WT_H3K27me3_Rep2_R1.fastq.gz ChIP-BS-seq_ESC_WT_H3K27me3_Rep2_R2.fastq.gz
ChIP-BS-seq_TSC1_WT_H3K27me3_Input_Rep1_R1.fastq.gz ChIP-BS-seq_TSC1_WT_H3K27me3_Input_Rep1_R2.fastq.gz
ChIP-BS-seq_TSC1_WT_H3K27me3_Input_Rep2_R1.fastq.gz ChIP-BS-seq_TSC1_WT_H3K27me3_Input_Rep2_R2.fastq.gz
ChIP-BS-seq_TSC1_WT_H3K27me3_Rep1_R1.fastq.gz ChIP-BS-seq_TSC1_WT_H3K27me3_Rep1_R2.fastq.gz
ChIP-BS-seq_TSC1_WT_H3K27me3_Rep2_R1.fastq.gz ChIP-BS-seq_TSC1_WT_H3K27me3_Rep2_R2.fastq.gz
```

ChIPseq_ESC_WT_EED_Input_Rep1_R1.fastq.gz ChIPseq_ESC_WT_EED_Input_Rep1_R2.fastq.gz
ChIPseq_ESC_WT_EED_Input_Rep2_R1.fastq.gz ChIPseq_ESC_WT_EED_Input_Rep2_R2.fastq.gz
ChIPseq_ESC_WT_EED_Rep1_R1.fastq.gz ChIPseq_ESC_WT_EED_Rep1_R2.fastq.gz
ChIPseq_ESC_WT_EED_Rep2_R1.fastq.gz ChIPseq_ESC_WT_EED_Rep2_R2.fastq.gz
ChIPseq_TSC1_WT_EED_Input_Rep1_R1.fastq.gz ChIPseq_TSC1_WT_EED_Input_Rep1_R2.fastq.gz
ChIPseq_TSC1_WT_EED_Input_Rep2_R1.fastq.gz ChIPseq_TSC1_WT_EED_Input_Rep2_R2.fastq.gz
ChIPseq_TSC1_WT_EED_Rep1_R1.fastq.gz ChIPseq_TSC1_WT_EED_Rep1_R2.fastq.gz
ChIPseq_TSC1_WT_EED_Rep2_R1.fastq.gz ChIPseq_TSC1_WT_EED_Rep2_R2.fastq.gz
ChIPseq_TSC1_WT_EZH2i_5w_4d_EED_Input_R1.fastq.gz ChIPseq_TSC1_WT_EZH2i_5w_4d_EED_Input_R2.fastq.gz
ChIPseq_TSC1_WT_EZH2i_5w_4d_EED_Rep1_R1.fastq.gz ChIPseq_TSC1_WT_EZH2i_5w_4d_EED_Rep1_R2.fastq.gz
ChIPseq_TSC1_WT_EZH2i_5w_4d_EED_Rep2_R1.fastq.gz ChIPseq_TSC1_WT_EZH2i_5w_4d_EED_Rep2_R2.fastq.gz
ChIP-BS-seq_ESC_WT_H3K27me3_Input_Rep1_RPGC_mm10.bw
ChIP-BS-seq_ESC_WT_H3K27me3_Input_Rep2_RPGC_mm10.bw
ChIP-BS-seq_ESC_WT_H3K27me3_Rep1_RPGC_mm10_input_subtracted.bw
ChIP-BS-seq_ESC_WT_H3K27me3_Rep2_RPGC_mm10_input_subtracted.bw
ChIP-BS-seq_ESC_WT_H3K27me3_merged_RPGC_mm10_input_subtracted.bw
ChIP-BS-seq_TSC1_WT_H3K27me3_Input_Rep1_RPGC_mm10.bw
ChIP-BS-seq_TSC1_WT_H3K27me3_Input_Rep2_RPGC_mm10.bw
ChIP-BS-seq_TSC1_WT_H3K27me3_Rep1_RPGC_mm10_input_subtracted.bw
ChIP-BS-seq_TSC1_WT_H3K27me3_Rep2_RPGC_mm10_input_subtracted.bw
ChIP-BS-seq_TSC1_WT_H3K27me3_merged_RPGC_mm10_input_subtracted.bw
ChIPseq_ESC_WT_EED_Input_Rep1_RPGC_mm10.bw
ChIPseq_ESC_WT_EED_Input_Rep2_RPGC_mm10.bw
ChIPseq_ESC_WT_EED_Rep1_RPGC_mm10_input_subtracted.bw
ChIPseq_ESC_WT_EED_Rep2_RPGC_mm10_input_subtracted.bw
ChIPseq_ESC_WT_EED_merged_RPGC_mm10_input_subtracted.bw
ChIPseq_TSC1_WT_EED_Input_Rep1_RPGC_mm10.bw
ChIPseq_TSC1_WT_EED_Input_Rep2_RPGC_mm10.bw
ChIPseq_TSC1_WT_EED_Rep1_RPGC_mm10_input_subtracted.bw
ChIPseq_TSC1_WT_EED_Rep2_RPGC_mm10_input_subtracted.bw
ChIPseq_TSC1_WT_EED_merged_RPGC_mm10_input_subtracted.bw
ChIPseq_TSC1_WT_EZH2i_5w_4d_EED_Input_RPGC_mm10.bw
ChIPseq_TSC1_WT_EZH2i_5w_4d_EED_Rep1_RPGC_mm10_input_subtracted.bw
ChIPseq_TSC1_WT_EZH2i_5w_4d_EED_Rep2_RPGC_mm10_input_subtracted.bw
ChIPseq_TSC1_WT_EZH2i_5w_4d_EED_merged_RPGC_mm10_input_subtracted.bw
ChIPseq_TSC1_WT_EED_merged_peaks_broad_mm10.bed
ChIPseq_TSC1_WT_EZH2i_5w_4d_EED_merged_peaks_broad_mm10.bed

**Genome browser session**
(e.g. UCSC)

No longer applicable.

## Methodology

**Replicates**

MINUTE-ChIP experiments were performed in triplicates. ChIP-BS-seq and EED ChIP-seq experiments were performed in duplicates.

**Sequencing depth**

All MINUTE-ChIP, ChIP-BS-seq and ChIP-seq samples were sequenced using 100 bp paired-end reads.

Sample - Number of reads - Number of reads aligned
MINUTE_ESC_WT_H2AK119ub1_Pool1_merged 141845280 101941533
MINUTE_ESC_WT_H2AK119ub1_Pool1_Rep1 50065548 36684163
MINUTE_ESC_WT_H2AK119ub1_Pool1_Rep2 41595303 30350659
MINUTE_ESC_WT_H2AK119ub1_Pool1_Rep3 50184429 34906711
MINUTE_TSC2_3BKO_H2AK119ub1_Pool1_merged 243588532 168567006
MINUTE_TSC2_3BKO_H2AK119ub1_Pool1_Rep1 80878090 55134033
MINUTE_TSC2_3BKO_H2AK119ub1_Pool1_Rep2 113100219 78952552
MINUTE_TSC2_3BKO_H2AK119ub1_Pool1_Rep3 49610223 34480421
MINUTE_TSC1_WT_H2AK119ub1_Pool1_merged 211698526 138859385
MINUTE_TSC1_WT_H2AK119ub1_Pool1_Rep1 78924631 53817493
MINUTE_TSC1_WT_H2AK119ub1_Pool1_Rep2 52123133 32688503
MINUTE_TSC1_WT_H2AK119ub1_Pool1_Rep3 80650762 52353389
MINUTE_ESC_WT_Input_Pool1_merged 343753429 151525584
MINUTE_ESC_WT_Input_Pool1_Rep1 102646276 54188917
MINUTE_ESC_WT_Input_Pool1_Rep2 118151325 43566752
MINUTE_ESC_WT_Input_Pool1_Rep3 122955828 53769915
MINUTE_TSC2_3BKO_Input_Pool1_merged 355408705 172183271
MINUTE_TSC2_3BKO_Input_Pool1_Rep1 99916131 53923869
MINUTE_TSC2_3BKO_Input_Pool1_Rep2 144545605 81663723
MINUTE_TSC2_3BKO_Input_Pool1_Rep3 110946969 36595679
MINUTE_TSC1_WT_Input_Pool1_merged 596699276 208711190
MINUTE_TSC1_WT_Input_Pool1_Rep1 273763826 88273170
MINUTE_TSC1_WT_Input_Pool1_Rep2 145001817 46244776
MINUTE_TSC1_WT_Input_Pool1_Rep3 177933633 74193244
MINUTE_ESC_WT_H3K27me3_Pool1_merged 32719139 19339355
MINUTE_ESC_WT_H3K27me3_Pool1_Rep1 11262487 7003712
MINUTE_ESC_WT_H3K27me3_Pool1_Rep2 8720460 5328786
MINUTE_ESC_WT_H3K27me3_Pool1_Rep3 12736192 7006857

```
MINUTE_TSC2_3BKO_H3K27me3_Pool1_merged 233603994 157500268
MINUTE_TSC2_3BKO_H3K27me3_Pool1_Rep1 76548171 50633401
MINUTE_TSC2_3BKO_H3K27me3_Pool1_Rep2 105486623 71866746
MINUTE_TSC2_3BKO_H3K27me3_Pool1_Rep3 51569200 35000121
MINUTE_TSC1_WT_H3K27me3_Pool1_merged 238672876 158569921
MINUTE_TSC1_WT_H3K27me3_Pool1_Rep1 87076293 60105343
MINUTE_TSC1_WT_H3K27me3_Pool1_Rep2 63859004 40560155
MINUTE_TSC1_WT_H3K27me3_Pool1_Rep3 87737579 57904423
MINUTE_ESC_WT_H3K4me3_Pool1_merged 186788627 148862331
MINUTE_ESC_WT_H3K4me3_Pool1_Rep1 67127140 55120682
MINUTE_ESC_WT_H3K4me3_Pool1_Rep2 51277688 39859044
MINUTE_ESC_WT_H3K4me3_Pool1_Rep3 68383799 53882605
MINUTE_TSC2_3BKO_H3K4me3_Pool1_merged 215014259 156808573
MINUTE_TSC2_3BKO_H3K4me3_Pool1_Rep1 68458242 50269827
MINUTE_TSC2_3BKO_H3K4me3_Pool1_Rep2 96640512 71477965
MINUTE_TSC2_3BKO_H3K4me3_Pool1_Rep3 49915505 35060781
MINUTE_TSC1_WT_H3K4me3_Pool1_merged 275362302 191912576
MINUTE_TSC1_WT_H3K4me3_Pool1_Rep1 111144193 79228100
MINUTE_TSC1_WT_H3K4me3_Pool1_Rep2 68590298 44988541
MINUTE_TSC1_WT_H3K4me3_Pool1_Rep3 95627811 67695935
MINUTE_TSC1_WT_EZH2i_5w_recovery_4w_H2AK119ub1_Pool2_merged 168298808 90733792
MINUTE_TSC1_WT_EZH2i_5w_recovery_4w_H2AK119ub1_Pool2_Rep1 57427876 31694730
MINUTE_TSC1_WT_EZH2i_5w_recovery_4w_H2AK119ub1_Pool2_Rep2 59219965 30329923
MINUTE_TSC1_WT_EZH2i_5w_recovery_4w_H2AK119ub1_Pool2_Rep3 51650967 28709139
MINUTE_TSC1_WT_H2AK119ub1_Pool2_merged 213901307 106497392
MINUTE_TSC1_WT_H2AK119ub1_Pool2_Rep1 65674827 34936402
MINUTE_TSC1_WT_H2AK119ub1_Pool2_Rep2 40497725 20878610
MINUTE_TSC1_WT_H2AK119ub1_Pool2_Rep3 107728755 50682380
MINUTE_TSC3_EEDKO_H2AK119ub1_Pool2_merged 218043557 126532156
MINUTE_TSC3_EEDKO_H2AK119ub1_Pool2_Rep1 73153206 44177404
MINUTE_TSC3_EEDKO_H2AK119ub1_Pool2_Rep2 60947244 34445412
MINUTE_TSC3_EEDKO_H2AK119ub1_Pool2_Rep3 83943107 47909340
MINUTE_TSC1_KDM2BKO_H2AK119ub1_Pool2_merged 153309045 89700547
MINUTE_TSC1_KDM2BKO_H2AK119ub1_Pool2_Rep1 56510675 34377032
MINUTE_TSC1_KDM2BKO_H2AK119ub1_Pool2_Rep2 42440202 23235706
MINUTE_TSC1_KDM2BKO_H2AK119ub1_Pool2_Rep3 54358168 32087809
MINUTE_TSC1_RNF2KO_H2AK119ub1_Pool2_merged 75530458 42988929
MINUTE_TSC1_RNF2KO_H2AK119ub1_Pool2_Rep1 19802076 10471992
MINUTE_TSC1_RNF2KO_H2AK119ub1_Pool2_Rep2 29662427 17982267
MINUTE_TSC1_RNF2KO_H2AK119ub1_Pool2_Rep3 26065955 14534670
MINUTE_TSC1_WT_Input_Pool2_merged 328722233 132495967
MINUTE_TSC1_WT_Input_Pool2_Rep1 100487642 46157234
MINUTE_TSC1_WT_Input_Pool2_Rep2 90249694 26241606
MINUTE_TSC1_WT_Input_Pool2_Rep3 137984897 60097127
MINUTE_TSC3_EEDKO_Input_Pool2_merged 296078609 119642065
MINUTE_TSC3_EEDKO_Input_Pool2_Rep1 87229727 41504715
MINUTE_TSC3_EEDKO_Input_Pool2_Rep2 99055499 31977906
MINUTE_TSC3_EEDKO_Input_Pool2_Rep3 109793383 46159444
MINUTE_TSC1_WT_EZH2i_5w_recovery_4w_Input_Pool2_merged 364789499 139552924
MINUTE_TSC1_WT_EZH2i_5w_recovery_4w_Input_Pool2_Rep1 124130256 47985369
MINUTE_TSC1_WT_EZH2i_5w_recovery_4w_Input_Pool2_Rep2 128732348 49841001
MINUTE_TSC1_WT_EZH2i_5w_recovery_4w_Input_Pool2_Rep3 111926895 41726554
MINUTE_TSC1_KDM2BKO_Input_Pool2_merged 390301863 168868264
MINUTE_TSC1_KDM2BKO_Input_Pool2_Rep1 160768318 67200838
MINUTE_TSC1_KDM2BKO_Input_Pool2_Rep2 108897239 42000159
MINUTE_TSC1_KDM2BKO_Input_Pool2_Rep3 120636306 59667267
MINUTE_TSC1_RNF2KO_Input_Pool2_merged 272464823 121777220
MINUTE_TSC1_RNF2KO_Input_Pool2_Rep1 62685809 30861253
MINUTE_TSC1_RNF2KO_Input_Pool2_Rep2 92580072 49249174
MINUTE_TSC1_RNF2KO_Input_Pool2_Rep3 117198942 41666793
MINUTE_TSC1_WT_EZH2i_5w_recovery_4w_H3K27me3_Pool2_merged 236356961 130738765
MINUTE_TSC1_WT_EZH2i_5w_recovery_4w_H3K27me3_Pool2_Rep1 73809591 41069794
MINUTE_TSC1_WT_EZH2i_5w_recovery_4w_H3K27me3_Pool2_Rep2 88303582 46927267
MINUTE_TSC1_WT_EZH2i_5w_recovery_4w_H3K27me3_Pool2_Rep3 74243788 42741704
MINUTE_TSC1_WT_H3K27me3_Pool2_merged 245067144 123141814
MINUTE_TSC1_WT_H3K27me3_Pool2_Rep1 73096464 39140708
MINUTE_TSC1_WT_H3K27me3_Pool2_Rep2 48309268 25453463
MINUTE_TSC1_WT_H3K27me3_Pool2_Rep3 123661412 58547643
MINUTE_TSC3_EEDKO_H3K27me3_Pool2_merged 34483059 14090270
MINUTE_TSC3_EEDKO_H3K27me3_Pool2_Rep1 10833672 4652161
MINUTE_TSC3_EEDKO_H3K27me3_Pool2_Rep2 8603252 3401434
MINUTE_TSC3_EEDKO_H3K27me3_Pool2_Rep3 15046135 6036675
MINUTE_TSC1_KDM2BKO_H3K27me3_Pool2_merged 165233328 98350986
MINUTE_TSC1_KDM2BKO_H3K27me3_Pool2_Rep1 59795267 36638260
MINUTE_TSC1_KDM2BKO_H3K27me3_Pool2_Rep2 46351310 25983293
MINUTE_TSC1_KDM2BKO_H3K27me3_Pool2_Rep3 59086751 35729433
```

```
MINUTE_TSC1_RNF2KO_H3K27me3_Pool2_merged 128076569 77314618
MINUTE_TSC1_RNF2KO_H3K27me3_Pool2_Rep1 30589049 17320342
MINUTE_TSC1_RNF2KO_H3K27me3_Pool2_Rep2 50906169 32635015
MINUTE_TSC1_RNF2KO_H3K27me3_Pool2_Rep3 46581351 27359261
MINUTE_TSC1_WT_EZH2i_5w_recovery_4w_H3K4me3_Pool2_merged 226774674 128311244
MINUTE_TSC1_WT_EZH2i_5w_recovery_4w_H3K4me3_Pool2_Rep1 78229362 43866195
MINUTE_TSC1_WT_EZH2i_5w_recovery_4w_H3K4me3_Pool2_Rep2 77863358 43801880
MINUTE_TSC1_WT_EZH2i_5w_recovery_4w_H3K4me3_Pool2_Rep3 70681954 40643169
MINUTE_TSC1_WT_H3K4me3_Pool2_merged 266836087 139662202
MINUTE_TSC1_WT_H3K4me3_Pool2_Rep1 81635204 45806411
MINUTE_TSC1_WT_H3K4me3_Pool2_Rep2 52228585 27050950
MINUTE_TSC1_WT_H3K4me3_Pool2_Rep3 132972298 66804841
MINUTE_TSC3_EEDKO_H3K4me3_Pool2_merged 175443574 100432930
MINUTE_TSC3_EEDKO_H3K4me3_Pool2_Rep1 59024385 35500868
MINUTE_TSC3_EEDKO_H3K4me3_Pool2_Rep2 47322664 25918081
MINUTE_TSC3_EEDKO_H3K4me3_Pool2_Rep3 69096525 39013981
MINUTE_TSC1_KDM2BKO_H3K4me3_Pool2_merged 255355318 160441580
MINUTE_TSC1_KDM2BKO_H3K4me3_Pool2_Rep1 99107491 63055348
MINUTE_TSC1_KDM2BKO_H3K4me3_Pool2_Rep2 66902809 39917029
MINUTE_TSC1_KDM2BKO_H3K4me3_Pool2_Rep3 89345018 57469203
MINUTE_TSC1_RNF2KO_H3K4me3_Pool2_merged 149092308 99126770
MINUTE_TSC1_RNF2KO_H3K4me3_Pool2_Rep1 37149959 23977642
MINUTE_TSC1_RNF2KO_H3K4me3_Pool2_Rep2 58733615 40704248
MINUTE_TSC1_RNF2KO_H3K4me3_Pool2_Rep3 53208734 34444880
ChIP-BS-seq_ESC_WT_H3K27me3_Rep1 164101626 140738559
ChIP-BS-seq_ESC_WT_H3K27me3_Input_Rep1 203837836 171835894
ChIP-BS-seq_ESC_WT_H3K27me3_Rep2 130882570 111365421
ChIP-BS-seq_ESC_WT_H3K27me3_Input_Rep2 150804558 126779101
ChIP-BS-seq_TSC1_WT_H3K27me3_Rep1 151500914 130089160
ChIP-BS-seq_TSC1_WT_H3K27me3_Input_Rep1 135742638 108544707
ChIP-BS-seq_TSC1_WT_H3K27me3_Rep2 133212844 116169501
ChIP-BS-seq_TSC1_WT_H3K27me3_Input_Rep2 138307810 113594936
ChIPseq_ESC_WT_EED_Rep1 110872064 109275287
ChIPseq_ESC_WT_EED_Input_Rep1 176371316 173280612
ChIPseq_ESC_WT_EED_Rep2 95414054 93546165
ChIPseq_ESC_WT_EED_Input_Rep2 113403308 112652634
ChIPseq_TSC1_WT_EED_Rep1 138446050 136848653
ChIPseq_TSC1_WT_EED_Input_Rep1 131513134 129550215
ChIPseq_TSC1_WT_EED_Rep2 106175636 104559800
ChIPseq_TSC1_WT_EED_Input_Rep2 101350538 100553303
ChIPseq_TSC1_WT_EZH2i_5w_4d_EED_Rep1 134135868 132238906
ChIPseq_TSC1_WT_EZH2i_5w_4d_EED_Rep2 136643690 134987043
ChIPseq_TSC1_WT_EZH2i_5w_4d_EED_Input 94794548 94117771
```

| | |
|---|---|
| Antibodies | MINUTE-ChIP:<br>Rabbit anti-H3K4me3: Millipore, #04-745, clone: MC315<br>Rabbit anti-H3K27me3: Cell Signaling, #9733, clone: C36B11<br>Rabbit anti-H2AK119ub: Cell Signaling, #8240, clone: D27C4<br><br>ChIP-BS-seq:<br>Rabbit anti-H3K27me3: Thermo Fisher, MA5-11198, lot: WH3366172<br><br>ChIP-seq:<br>Rabbit anti-EED: ab240650, EPR23043-5, lot: GR3427609-2 |
| Peak calling parameters | Peaks for EED ChIPs were called using MACS2 'callpeak' (version 2.1.2; parameters --bdg --SPMR --broad) based on merged replicates using the input samples as control samples and only peaks with a q-value < 0.01 were considered for downstream analyses. No peak calling was performed for MINUTE-ChIP data generated within this study. Instead the scaled RPGC values were used to calculate the average intensity across CpG islands or one kb tiles. The tracks were generated as described below (Software). |
| Data quality | FastQC was run on all FASTQ files to assess general sequencing quality. Picard was used to determine insert size distribution, duplication rate, estimated library size. Mapping stats were generated from BAM files using samtools idxstats and flagstat commands. Final reports with all the statistics generated throughout the pipeline execution are gathered with MultiQC.<br>For EED ChIP-seq samples, peaks were called on the merged replicates. For WT TSCs 19,775 broad peaks were called with an FDR < 0.01. For WT TSCs treated with EZH2i 7,553 broad peaks were called with an FDR < 0.01. |
| Software | MINUTE-ChIP processing:<br>MINUTE-ChIP multiplexed FASTQ files were processed using minute, a workflow implemented in Snakemake. In order to ensure reproducibility, a conda environment was set up. Source code and documentation are fully available on GitHub: https://github.com/NBISweden/minute. Main steps performed are described below.<br>Adaptor removal: Read pairs matching parts of the adaptor sequence (SBS3 or T7 promoter) in either read1 or read2 were removed using cutadapt v3.2. |

Demultiplexing and deduplication: Reads were demultiplexed using cutadapt v3.2 allowing only one mismatch per barcode and written into sample-specific FASTQ files used for subsequent mapping.

Mapping: Sample-specific paired FASTQ files were mapped to the reference mm10 using bowtie2 v2.3.5.1 with --fast and --reorder parameter. Alignments were processed into sorted BAM files and replicates were pooled using samtools v1.10.

Deduplication: Duplicate reads are marked using UMI-sensitive deduplication tool je-suite (v2.0.RC) (https://github.com/gbcs-embl/Je/). Read pairs are marked as duplicates if their read1 (first-in-pair) sequences have the same UMI (allowing for 1 mismatch) and map to the same location in the genome. Blacklisted regions as downloaded from ENCODE were then removed from BAM files using bedtools v2.30.

Generation of coverage tracks and quantitative scaling: Input coverage tracks with 1 bp resolution in bigWig format were generated from BAM files using deepTools v3.5.0 bamCoverage and scaled to a reads-per-genome-coverage of one (1xRPGC, also referred to as '1x normalization') using the mm10 effective genome size. ChIP coverage tracks were generated from BAM files using deepTools (v3.5.0) bamCoverage. Quantitative scaling of the ChIP-Seq tracks amongst conditions within each pool was based on their Input-Normalized Mapped Read Count (INRC). INRC was calculated by dividing the number of unique reference-mapped reads by the respective number of Input reads: #mapped[ChIP] / #mapped[Input]. This essentially corrects for an uneven representation of barcodes in the Input. It has been previously shown that INRCs are proportional to the amount of epitope present in each condition. Reference condition (TSC WT) was scaled to 1x coverage (also termed Reads per Genome Coverage, RPGC). All other conditions within the same pool were scaled relative to the reference using the ratio of INRCs multiplied by the scaling factor determined for 1x normalization of the reference: ( #mapped[ChIP] / #mapped[Input] ) / ( #mapped[ChIP_Reference] / #mapped[Input_Reference] ) * scaling factor.

ChIP-BS-seq processing:

Raw reads of ESC and TSC H3K27me3 ChIP-BS-seq samples as well as their respective input samples were subjected to adapter and quality trimming with cutadapt (version 2.4; parameters: --quality-cutoff 20 --overlap 5 --minimum-length 25 --adapter AGATCGGAAGAGC -A AGATCGGAAGAGC). Reads were aligned to the mouse genome (mm10) using BSMAP (version 2.90; parameters: -v 0.1 -s 16 -q 20 -w 100 -S 1 -u -R). A sorted BAM file was obtained and indexed using samtools with the 'sort' and 'index' commands (version 1.10). Duplicate reads were identified and removed using GATK (version 4.1.4.1) 'MarkDuplicates' and default parameters. After careful inspection and validation of high correlation, replicates of treatment and input samples were merged respectively using samtools 'merge'. Methylation rates were called using mcall from the MOABS package (version 1.3.2; default parameters). All analyses were restricted to autosomes and only CpGs covered by at least 10 and at most 150 reads were considered for downstream analyses. Genome-wide coverage tracks for single and merged replicates normalized by library size were computed using deepTools bamCoverage (parameters: --normalizeUsing RPGC --extendReads --smoothLength 300). Coverage tracks were subtracted by the respective input using deeptools 'bigwigCompare'.

ChIP-seq processing:

Raw reads of ESC and TSC EED and publicly available ESC H3K27me3 ChIP-seq samples as well as their respective input samples were subjected to adapter and quality trimming with cutadapt (version 2.4; parameters: --quality-cutoff 20 --overlap 5 --minimum-length 25 --adapter AGATCGGAAGAGC -A AGATCGGAAGAGC). Reads were aligned to the mouse genome (mm10) using BWA with the 'mem' command (version 0.7.17, default parameters)61. A sorted BAM file was obtained and indexed using samtools with the 'sort' and 'index' commands (version 1.10)66. Duplicate reads were identified and removed using GATK (version 4.1.4.1) 'MarkDuplicates' and default parameters. After careful inspection and validation of high correlation, replicates of treatment and input samples were merged respectively using samtools 'merge'. Domains for public H3K27me3 ESC samples were called for each sample with its respective input using peakranger 'bcp' (version 1.18) 67. Only regions called a domain in at least two of the samples were considered for the final selection and merged using bedtools 'mergeBed' (parameters: -d 50). Retained regions smaller than 100 bp were removed from the set. Peaks for EED ChIPs were called using MACS2 'callpeak' (version 2.1.2; parameters --bdg --SPMR --broad) based on merged replicates using the input samples as control samples and only peaks with a q-value < 0.01 were considered for downstream analyses. Genome-wide coverage tracks for single and merged replicates normalized by library size were computed using deepTools bamCoverage (parameters: --normalizeUsing RPGC --extendReads --smoothLength 300). Coverage tracks were subtracted by the respective input using deeptools 'bigwigCompare'.

# Flow Cytometry

## Plots

Confirm that:

☒ The axis labels state the marker and fluorochrome used (e.g. CD4-FITC).

☒ The axis scales are clearly visible. Include numbers along axes only for bottom left plot of group (a 'group' is an analysis of identical markers).

☒ All plots are contour plots with outliers or pseudocolor plots.

☒ A numerical value for number of cells or percentage (with statistics) is provided.

## Methodology

| | |
|---|---|
| Sample preparation | Cells were detached using Trypsin-EDTA 0.05 % for 10 minutes. Subsequently, trypsinization was stopped by addition of ESC/TSC medium containing FBS and cells were dissociated to generate a single cell suspension. Cells were spun down, washed once with PBS and passed through a FACS tube with cell strainer just before the sort with the flow cytometer |
| Instrument | BD FACS Aria II and BD FACS Fusion |
| Software | FACS Diva (BD Biosciences) for collection and FlowJo (v1.07) for analysis |

| Cell population abundance | The overall cell population was calculated using forward and side-scatter patterns. The abundance of cells in a population is represented as the normalized mode. |
| Gating strategy | Gating for negative and positive population was determined with untreated or isotype controls. |

☒ Tick this box to confirm that a figure exemplifying the gating strategy is provided in the Supplementary Information.

