## [Peer Review File · Nature Cell Biology]

Peer Review Information

Journal: Nature Cell Biology

Manuscript Title: Dynamic antagonism between key repressive pathways maintains the placental epigenome

Corresponding author name(s): Professor Zachary D Smith Meissner

Editorial Notes:

Reviewer Comments & Decisions:

Decision Letter, initial version:
--

Message: *Please delete the link to your author homepage if you wish to forward this email to co-authors.

Dear Alex,

I hope this message finds you well. Your manuscript, "Dynamic antagonism between key repressive pathways in the placental epigenome", has now been seen by 3 referees, who are experts in epigenetics, DNA methylation, embryogenesis (referee 1); trophoblast stem cells (referee 2); and DNA methylation, polycomb, biochemistry (referee 3). As you will see from their comments (attached below) they find this work of potential interest, but have raised substantial concerns, which in our view would need to be addressed with considerable revisions before we can consider publication in Nature Cell Biology.

Nature Cell Biology editors discuss the referee reports in detail within the editorial team, including the chief editor, to identify key referee points that should be addressed with priority. To guide the scope of the revisions, I have listed these points below. We are committed to providing a fair and constructive peer-review process, so please feel free to contact me if you would like to discuss any of the referee comments further.

In particular, it would be essential to:

(A) Further test the unique genome co-occurrence of DNA methylation and PRC2-related H3K27me3 in TSCs compared to ESCs, as indicated by:

Reviewer 1:

"One major claim of the manuscript is the unique genome co-occurrence of DNA

methylation and PRC2-related H3K27me3 in TSCs compared to ESCs, confirming what the authors had previously observed in dissected extraembryonic (EXE) tissues at E6.5. However, to demonstrate that the two marks really exist at the same locus in the same cell and not in separate cells, the authors need to perform ChIP-seq followed by bisulfite sequencing. This was challenging in in vivo EXE tissues, regarding the low amount of material, but this should not be a limitation anymore in cultured TSCs. Results from this experiment would be equally interesting, whether they confirm direct co-occupancy or reveal differential occupancy in distinct cells. And this would provide clearer insights into the relationship between the two marks".

Reviewer 3:

"The chromatin colocalization profile of DNMT3B and PRC2 is absent. DNMT3B and PRC2 ChIP-seq should be included. Also, while the co-occurrence of DNA methylation and H3K27 methylation were analyzed in bulk level, it's better to perform bisulfite sequencing using the DNA retrieved from H3K27me3 ChIP which would be a direct evidence showing the co-occurrence of the two modifications".

(B) Further investigate and clarify the mechanism claimed and strengthen the rigor of current datasets, as indicated by:

Reviewer 1:

"One limitation also is that we finally do not learn how the interplay between DNA methylation and H3K27me3 operates in TSCs, while this was the original and announced goal of the study. It is still unclear as to why the co-occurrence of the two marks is tolerated genome-wide in TSCs, even at hypermethylated CGIs. Maybe different flavors of the PRC2 complex exist in this cell type, for exemple, but this is not studied here".

"One key question is whether the H3K27me3 mark or EED/the PRC2 complex itself is required for generating intermediate DNA methylation levels. Because DNA methylation gain is equally observed in Eed-KO, where both H3K27marks are abolished and the PRC2 complex is destabilized, and upon treatment with inhibitor of EZH2 activity, the authors conclude-in their discussion-that the mark itself may be responsible for a competition with DNMT3B. As a support, they mention that "the EZH2i is expected to preserve the complex and only block its catalytic activity". How solid is this statement? It is not backed up by publication references. Do we know indeed that the PRC2 complex is not altered in its composition upon EZH2i? Because the authors reported direct interactions between EED/PRC2 and DNMT3B, the question remains open".

"To document whether the co-occurrence of H3K27me3 and DNA methylation in TSCs results from antagonistic relationships between EED/PRC2 and DNMT3B that are driven by direct physical interactions between these two entities, the authors used EDD-centered co-IP and mass spec. However, one necessary and essential control that is missing here would to carry out such direct interaction assay in ESCs, where the two marks are usually exclusive".

Reviewer 2:

"It would also be useful to have more data from the current study on the changes in morphology and gene expression when TS cells were treated with inhibitors. Fig 5a shows that TS cells change morphology after treatment with inhibitors- it is stated that they flatten and senesce. But looking at the images suggests they might be ceasing mitosis and becoming trophoblast giant cells- the terminal differentiation stage for TS cells. They state that genes involved in core cellular functions are affected- but did they look to see if cells differentiated? And what happens to the DMRs when the TS cells differentiate?"

Reviewer 3:

"4 different TSC lines were used in this study. However, genetic disruptions were performed with different lines: TSCDnmt3bKO from TSC2 (female), TSCRnf2KO and TSCKdm2bKO from TSC1 (male), TSCEedKO from TSC3 (male). Violin plots in ED Fig. 1d lacks TSC1 cells.

Because TSCRnf2KO and TSCEedKO were derived from different parental cells, it is not so solid to conclude that "PRC2 knockout cells showed the most dramatic increase in DNA methylation" without illustrating the parental epigenome status respectively".

"Do the Tet enzymes play any roles in the heterogeneity and dynamics of the methylome in TSCs? This needs to be discussed at least".

(C) Highlight the biological implication and significance of your findings, as indicated by reviewer 2:

"My only major query relates to the lack of much discussion or experimentation relevant to the biological significance of this intermediate methylation state. The authors state that there is little work on the epigenetic regulation of TS cells and, while this may be generally true, there are a number of groups who have contributed to this area in both stem cells and in the embryo. Miriam Hemberger's group in particular- note Senner et al, Stem Cells 2012- Branco et al Dev Cell 2016. Also note Legault et al 2020 Epigenetics: Developmental genome-wide DNA methylation asymmetry between mouse placenta and embryo. This study identifies specific DMRs between embryo and placenta in early development and shows that hypomethylated DMRs in the placenta are enriched in genes involved in reproductive and germline functions. They also look at the roles of the DNMTs in the process. It would be good to integrate the findings in the current study more carefully with their potential biological significance as derived from previous studies".

(D) All other referee concerns pertaining to strengthening existing data, providing controls, methodological details, clarifications and textual changes, should also be addressed.

(E) Finally please pay close attention to our guidelines on statistical and methodological reporting (listed below) as failure to do so may delay the reconsideration of the revised manuscript. In particular please provide:

We would be happy to consider a revised manuscript that would satisfactorily address these points, unless a similar paper is published elsewhere, or is accepted for publication in Nature Cell Biology in the meantime.

- ensure that it conforms to our format instructions and publication policies (see below and www.nature.com/nature/authors/).

- provide a point-by-point rebuttal to the full referee reports verbatim, as provided at the end of this letter.

- provide the completed Editorial Policy Checklist (found here <https://www.nature.com/authors/policies/Policy.pdf>), and Reporting Summary (found here <https://www.nature.com/authors/policies/ReportingSummary.pdf>). This is essential for reconsideration of the manuscript and these documents will be available to editors and referees in the event of peer review. For more information see <http://www.nature.com/authors/policies/availability.html> or contact me.

Nature Cell Biology is committed to improving transparency in authorship. As part of our efforts in this direction, we are now requesting that all authors identified as 'corresponding author' on published papers create and link their Open Researcher and Contributor Identifier (ORCID) with their account on the Manuscript Tracking System (MTS), prior to acceptance. ORCID helps the scientific community achieve unambiguous attribution of all scholarly contributions. You can create and link your ORCID from the home page of the MTS by clicking on 'Modify my Springer Nature account'. For more information please visit please visit www.springernature.com/orcid.

[Redacted]

We would like to receive a revised submission within six months. We would be happy to consider a revision even after this timeframe, however if the resubmission deadline is missed and the paper is eventually published, the submission date will be the date when

the revised manuscript was received.

We hope that you will find our referees' comments, and editorial guidance helpful. Please do not hesitate to contact me if there is anything you would like to discuss.

Best wishes,

Stelios

Stylios Lefkopoulos, PhD
He/him/his
Associate Editor
Nature Cell Biology
Springer Nature
Heidelberger Platz 3, 14197 Berlin, Germany

E-mail: stylios.lefkopoulos@springernature.com

Twitter: @s_lefkopoulos

Reviewers' Comments:

Reviewer #1:

Remarks to the Author:

Mammalian extraembryonic tissues exhibit distinct chromatin characteristics compared to embryonic tissues, which may be relevant for their function. In particular, DNA methylation exists at intermediate levels (at so called Partially Methylated Domains, PMDs) and co-occur with usually antagonistic Polycomb-associated H3K27me3 marks. The reasons for these non-canonical epigenomic patterns are unknown. To gain mechanistic insights, Weigert et al. used here cultured trophoblast stem cell lines (TSCs), allowing them to perturb the dynamics of DNA methylation and H3K27me3 at will. Using four independent TSC lines (2 females and 2 males), they could conclude that cultured TSCs indeed reproduce the extraembryonic-specific chromatin features observed in vivo, using both short and long-read sequencing. They further demonstrated that intermediate DNA methylation levels in TSCs 1-reflect continuous opposition between DNA methylation gain and loss-rather than a failure to maintain DNA methylation across cell divisions-that relates to a functional interplay between DNMT3B and PRC2, which is reflected by direct physical interactions between DDE and DNMT3B. Finally, using inhibitors of DNMT1 and EZH2, they went into deeper details into the relative dynamics of each mark, and further demonstrated that one or the other mark is sufficient to maintain transcriptional patterns of gene showing co-enrichment of the two marks, while ablation of both marks strongly impacted on core cellular function, decreased proliferation and induced senescence. This is an elegant study and clearly written study, which makes the best usage of both genetic and chemical alterations to highlight the genetic link between DNA methylation and PRC2 in shaping the extraembryonic epigenome, and its remarkable ability to be autonomously restored after perturbations of one or the other components. Functionally, it also reveals the need of both marks for TSC functioning, although this is not a specific feature of TSCs, as it is known ESCs are also profoundly impacted by the lack of both marks. This is one of the weaknesses of the study, as the usage of DNMT and EZH2 inhibitors induces large scale genomic effects, which may not be related at all to the

regulation of TSC-specific domains of DNA methylation and H3K27me3 co-enriched territories. One limitation also is that we finally do not learn how the interplay between DNA methylation and H3K27me3 operates in TSCs, while this was the original and announced goal of the study. It is still unclear as to why the co-occurrence of the two marks is tolerated genome-wide in TSCs, even at hypermethylated CGIs. Maybe different flavors of the PRC2 complex exist in this cell type, for example, but this is not studied here.

My major concerns are highlighted below, and these are experimentally addressable:

- One major claim of the manuscript is the unique genome co-occurrence of DNA methylation and PRC2-related H3K27me3 in TSCs compared to ESCs, confirming what the authors had previously observed in dissected extraembryonic (EXE) tissues at E6.5. However, to demonstrate that the two marks really exist at the same locus in the same cell and not in separate cells, the authors need to perform ChIP-seq followed by bisulfite sequencing. This was challenging in *in vivo* EXE tissues, regarding the low amount of material, but this should not be a limitation anymore in cultured TSCs. Results from this experiment would be equally interesting, whether they confirm direct co-occupancy or reveal differential occupancy in distinct cells. And this would provide clearer insights into the relationship between the two marks.
- To document whether the co-occurrence of H3K27me3 and DNA methylation in TSCs results from antagonistic relationships between EED/PRC2 and DNMT3B that are driven by direct physical interactions between these two entities, the authors used EED-centered co-IP and mass spec. However, one necessary and essential control that is missing here would be to carry out such direct interaction assay in ESCs, where the two marks are usually exclusive.
- In their former *in vivo* paper (Smith et al. 2017), the authors reported that Eed-KO EXE tissues did not show changes in genome-wide DNA methylation levels and loss of DNA methylation at hypermethylated CGIs and concluded that *de novo* DNA methylation of CGIs required PRC2 in this tissue. Here, in cultured TSCs, Eed-KO rather results in genome-wide gain of DNA methylation, including at PMDs and hyper CGIs, therefore highlighting antagonistic relationships. How can the authors reconcile these opposite results? It is possible that I missed differences in the genomic sites that were assessed in the two studies.
- One key question is whether the H3K27me3 mark or EED/the PRC2 complex itself is required for generating intermediate DNA methylation levels. Because DNA methylation gain is equally observed in Eed-KO, where both H3K27marks are abolished and the PRC2 complex is destabilized, and upon treatment with inhibitor of EZH2 activity, the authors conclude in their discussion that the mark itself may be responsible for a competition with DNMT3B. As a support, they mention that “the EZH2i is expected to preserve the complex and only block its catalytic activity”. How solid is this statement? It is not backed up by publication references. Do we know indeed that the PRC2 complex is not altered in its composition upon EZH2i? Because the authors reported direct interactions between EED/PRC2 and DNMT3B, the question remains open.

Minor comments:

- Compared to male ESCs, female ESCs tend to lose DNA methylation genome-wide. Although it does not seem to be the case from the presented figures, the authors should mention in their manuscript whether female and male TSCs show the same DNA methylation levels.

Reviewer #2:

Remarks to the Author:

In this study, the authors undertake a detailed analysis of the status of DNA methylation and Polycomb regulation in mouse trophoblast stem cells. They confirm that the intermediate state of DNA methylation observed in trophoblast in vivo is maintained in the in vitro cell lines. They show that there is a balanced stable antagonistic relationship between the two repressive pathways in maintaining this unusual intermediate state. They use appropriate sequencing tools and algorithms to assess the epigenetic landscape in the cells and they also use a combination of genetic mutations and chemical inhibitors to verify the roles of the different DNMTs and Polycomb genes in this process.

Overall, the quality of the data presented seems high and the datasets will be a strong resource for the field. My only major query relates to the lack of much discussion or experimentation relevant to the biological significance of this intermediate methylation state. The authors state that there is little work on the epigenetic regulation of TS cells and, while this may be generally true, there are a number of groups who have contributed to this area in both stem cells and in the embryo. Miriam Hemberger's group in particular- note Senner et al, Stem Cells 2012- Branco et al Dev Cell 2016. Also note Legault et al 2020 Epigenetics: Developmental genome-wide DNA methylation asymmetry between mouse placenta and embryo. This study identifies specific DMRs between embryo and placenta in early development and shows that hypomethylated DMRs in the placenta are enriched in genes involved in reproductive and germline functions. They also look at the roles of the DNMTs in the process. It would be good to integrate the findings in the current study more carefully with their potential biological significance as derived from previous studies.

It would also be useful to have more data from the current study on the changes in morphology and gene expression when TS cells were treated with inhibitors. Fig 5a shows that TS cells change morphology after treatment with inhibitors- it is stated that they flatten and senesce. But looking at the images suggests they might be ceasing mitosis and becoming trophoblast giant cells- the terminal differentiation stage for TS cells. They state that genes involved in core cellular functions are affected- but did they look to see if cells differentiated? And what happens to the DMRs when the TS cells differentiate?

Reviewer #3:

Remarks to the Author:

Extraembryonic epigenome is less studied compared to embryonic cells. In this manuscript, the authors utilized trophoblast stem cells (TSCs) as a model to explore the extraembryonic lineages characteristics. They found TSCs preserved hypermethylated CGIs, which also contained high levels of H3K27me3. These non-canonical intermediate methylated CGIs were targets of DNMT3B and PRC2, which physically interact and antagonize each other, ensuring the special DNA methylation landscape of TSCs. The epigenome of extraembryonic lineages is indeed worth profiled. However, the relationship between methylome and H3K27me3 distribution seems not to be consistent between TSCs and extraembryonic ectoderm (ExE). The main shortage of the study is the mechanism underlying the co-occupancy of H3K27 methylation and DNA methylation. The authors provided some circumstantial evidence arguing that DNMT3B and PRC2 may directly interact. This really needs to be solidly clarified before publication.

Major points:

1. 4 different TSC lines were used in this study. However, genetic disruptions were performed with different lines: TSCDnmt3bKO from TSC2 (female), TSCRnf2KO and TSCKdm2bKO from TSC1 (male), TSCEedKO from TSC3 (male). Violin plots in ED Fig. 1d lacks TSC1 cells.
Because TSCRnf2KO and TSCEedKO were derived from different parental cells, it is not so solid to conclude that "PRC2 knockout cells showed the most dramatic increase in DNA methylation" without illustrating the parental epigenome status respectively.
2. To my understanding, TSC cells were explored to mimic the in vivo system of extraembryonic lineages. In this manuscript, eed-null results in global increase of DNA methylation as shown in Fig 3c, 3d, 3h and ED Fig. 4b-f, except canonical CGIs. However, in the Nature paper published by the authors in 2017, global DNAm patterns were preserved in eed-null ExE, while CGI methylation was disrupted. What is the explanation for the opposite patterns?
3. Fig. 2g, and ED Fig. 3g, the author concluded that "H3K4me3 retained its expected negative correlation with DNA methylation". However, DNA methylation level of HMD, PMD and Hyper CGIs were elevated in TSCKdm2bKO cells in Fig 3c and 3d? In regions of "CGIs hypermethylated in PRC KOs" in ED Fig. 5c, DNA methylation level also got enhanced along with the increased H3K4me3 signals in TSCKdm2bKO cells. Any explanations?
4. The chromatin colocalization profile of DNMT3B and PRC2 is absent. DNMT3B and PRC2 ChIP-seq should be included. Also, while the co-occurrence of DNA methylation and H3K27 methylation were analyzed in bulk level, it's better to perform bisulfite sequencing using the DNA retrieved from H3K27me3 ChIP which would be a direct evidence showing the co-occurrence of the two modifications.
5. Fig. 4d, co-IP, which confirmed the physical interaction between PRC2 and DNMT3B, was performed with 110 mM NaCl, a salt concentration far below the standard line, why? Can the authors repeat this experiment with higher salt concentration?
6. DNA methylation and H3K27me3 co-existed in hyper CGIs. Does this reflect cell/allele heterogeneity?
7. Do the Tet enzymes play any roles in the heterogeneity and dynamics of the methylome in TSCs? This needs to be discussed at least.

Minor points:

1. Fig. 4b, why was mean methylation level in the right EZH2i-5weeks higher than the left EZH2i-5 weeks? DNA methylation levels increased to ~85-90% after 4 or 5-weeks EZH2i treatment in Fig. 4b and ED Fig. 7a, not ~100% as described in the text.
2. Typo "Mass Spectrometry" page 5, line 13.
3. Fig. 4e, two dots were labeled as EZH2.

Methods should be written concisely, but should contain all elements necessary to allow interpretation and replication of the results. As a guideline, Methods sections typically do not exceed 3,000 words. The Methods should be divided into subsections listing reagents and techniques. When citing previous methods, accurate references should be provided and any alterations should be noted. Information must be provided about: antibody dilutions, company names, catalogue numbers and clone numbers for monoclonal antibodies; sequences of RNAi and cDNA probes/primers or company names and catalogue numbers if reagents are commercial; cell line names, sources and information on cell line identity and authentication. Animal studies and experiments involving human subjects must be reported in detail, identifying the committees approving the protocols. For studies involving human subjects/samples, a statement must be included confirming that informed consent was obtained. Statistical analyses and information on the reproducibility of experimental results should be provided in a section titled "Statistics and Reproducibility".

All Nature Cell Biology manuscripts submitted on or after March 21 2016 must include a Data availability statement at the end of the Methods section. For Springer Nature policies on data availability see <http://www.nature.com/authors/policies/availability.html>; for more information on this particular policy see <http://www.nature.com/authors/policies/data/data-availability-statements-data-citations.pdf>. The Data availability statement should include:

- Accession codes for primary datasets (generated during the study under consideration and designated as "primary accessions") and secondary datasets (published datasets reanalysed during the study under consideration, designated as "referenced accessions"). For primary accessions data should be made public to coincide with publication of the manuscript. A list of data types for which submission to community-endorsed public repositories is mandated (including sequence, structure, microarray, deep sequencing data) can be found here <http://www.nature.com/authors/policies/availability.html#data>.
- Unique identifiers (accession codes, DOIs or other unique persistent identifier) and hyperlinks for datasets deposited in an approved repository, but for which data deposition is not mandated (see here for details <http://www.nature.com/sdata/data-policies/repositories>).

- At a minimum, please include a statement confirming that all relevant data are available from the authors, and/or are included with the manuscript (e.g. as source data or supplementary information), listing which data are included (e.g. by figure panels and data types) and mentioning any restrictions on availability.
- If a dataset has a Digital Object Identifier (DOI) as its unique identifier, we strongly encourage including this in the Reference list and citing the dataset in the Methods.

We recommend that you upload the step-by-step protocols used in this manuscript to the Protocol Exchange. More details can found at www.nature.com/protocolexchange/about.

All imaging data should be accompanied by scale bars, which should be defined in the legend.

Cropped images of gels/blots are acceptable, but need to be accompanied by size markers, and to retain visible background signal within the linear range (i.e. should not be saturated). The boundaries of panels with low background have to be demarked with black lines. Splicing of panels should only be considered if unavoidable, and must be clearly marked on the figure, and noted in the legend with a statement on whether the samples were obtained and processed simultaneously. Quantitative comparisons between samples on different gels/blots are discouraged; if this is unavoidable, it should only be performed for samples derived from the same experiment with gels/blots were processed in parallel, which needs to be stated in the legend.

Supplementary items should relate to a main text figure, wherever possible, and should be

mentioned sequentially in the main manuscript, designated as Supplementary Figure, Table, Video, or Note, and numbered continuously (e.g. Supplementary Figure 1, Supplementary Figure 2, Supplementary Table 1, Supplementary Table 2 etc.).

The total number of Supplementary Figures (not including the “unprocessed scans” Supplementary Figure) should not exceed the number of main display items (figures and/or tables (see our Guide to Authors and March 2012 editorial <http://www.nature.com/ncb/authors/submit/index.html#suppinfo>; <http://www.nature.com/ncb/journal/v14/n3/index.html#ed>). No restrictions apply to Supplementary Tables or Videos, but we advise authors to be selective in including supplemental data.

GUIDELINES FOR EXPERIMENTAL AND STATISTICAL REPORTING

REPORTING REQUIREMENTS – To improve the quality of methods and statistics reporting in our papers we have recently revised the reporting checklist we introduced in 2013. We are now asking all life sciences authors to complete two items: an Editorial Policy Checklist (found here <https://www.nature.com/authors/policies/Policy.pdf>) that verifies compliance with all required editorial policies and a reporting summary (found here <https://www.nature.com/authors/policies/ReportingSummary.pdf>) that collects information on experimental design and reagents. These documents are available to referees to aid the evaluation of the manuscript. Please note that these forms are dynamic ‘smart pdfs’ and must therefore be downloaded and completed in Adobe Reader. We will then flatten them for ease of use by the reviewers. If you would like to reference the guidance text as you complete the template, please access these flattened versions at <http://www.nature.com/authors/policies/availability.html>.

STATISTICS – Wherever statistics have been derived the legend needs to provide the n number (i.e. the sample size used to derive statistics) as a precise value (not a range), and define what this value represents. Error bars need to be defined in the legends (e.g. SD, SEM) together with a measure of centre (e.g. mean, median). Box plots need to be defined in terms of minima, maxima, centre, and percentiles. Ranges are more appropriate than standard errors for small data sets. Wherever statistical significance has been derived, precise p values need to be provided and the statistical test used needs to

be stated in the legend. Statistics such as error bars must not be derived from $n < 3$. For sample sizes of $n < 5$ please plot the individual data points rather than providing bar graphs. Deriving statistics from technical replicate samples, rather than biological replicates is strongly discouraged. Wherever statistical significance has been derived, precise p values need to be provided and the statistical test stated in the legend.

Author Rebuttal to Initial comments

We would like to thank the reviewers for their helpful and constructive feedback. We now provide multiple additional experiments and analyses to address all points raised. A detailed point-by-point response is provided below.

We also thank the editor for highlighting the main areas that needed to be strengthened. Here we provide a brief response with more details in the full response below.

The editorial team suggested it would be essential to address the following **Main Points**:

(A) Further test the unique genome co-occurrence of DNA methylation and PRC2-related H3K27me3 in TSCs compared to ESCs (Reviewer 1 and 3).

We fully agree that this is indeed a central point and have therefore added several new experiments to highlight that H3K27me3 and DNA methylation co-exist at the same loci within TSCs. These include:

- 1) Sequential Chromatin ImmunoPrecipitation followed by Bisulfite Sequencing (ChIP-BS-seq)^{1,2} for ESCs and TSCs (4 new data sets). Our analysis confirms that the DNA surrounding H3K27me3-modified nucleosomes have identical DNA methylation levels and patterns in TSCs as they do within the unenriched background (as measured by WGBS). In contrast, ESCs exhibit the canonical inverse relationship between DNA methylation and H3K27me3 (see **Figures 2h-j, ED Figure 4**), where H3K27me3

enriched CpG-rich regions remain free of methylation. Taken together, these new data clearly demonstrate that DNA and H3K27 methylation co-occupy the same genomic loci in TSCs.

- 2) We have also added EED ChIP-Seq data for ESCs and TSCs, which confirms the continued occupancy of PRC2 at target regions despite their intermediate DNA methylation status (see **Figures 2h,i, ED Figure 4**). Combined with quantitative mass spectrometry data and WB analyses that show a global enrichment for H3K27me3 in TSCs (**ED Figure 3f,g**), these data highlight the continued regulation of developmental gene promoters by PRC2 in TSCs, despite the novel co-occupancy with DNA methylation.
- 3) We also generated additional co-IP and Mass Spectrometry data to further strengthen the context for these interactions, including IP Western and Mass Spec data for mESCs. In particular we would like to point out that, although embryonic and trophoblast stem cells both exhibit biochemical interactions between the core PRC2 complex and DNMT3B (see **Figure 4d, ED Figure 8g** and Refs 3,4), the global interactomes for this

regulator are quantitatively different between the two cell types (**ED Figure 8h,i**) in a manner that is consistent with expanded global functions within TSCs.

(B) Further investigate and clarify the mechanism claimed and strengthen the rigor of current datasets (Reviewers 1-3)

In line with the reviewer suggestions, we generated additional data sets including the above-mentioned ChIP-BS, EED ChIP-seq, and IP-MS data, all of which confirm the interaction between DNMT3B and PRC2 in TSCs, as well as the dual presence of their modifications in chromatin.

Furthermore, we have generated an additional knockout TSC line for the most expressed DNA dioxygenase in TSCs (TET3) and performed WGBS as well as quantitative MS of 5-methyl and 5-hydroxymethylcytosine (5-mC and 5-hmC). These data confirm that TET3 is largely responsible for 5-hmC within TSCs, but that this modification is not notably enriched in TSCs compared to mESCs (see **ED Figure 5d** as well as references 5,6, now included, for further clarification). Moreover, TET3 disruption does not have a notable effect on DNA methylation levels, either globally or at hypermethylated CpG islands. Thus, our new data establish that 5-hmC is globally depleted in the absence of TET3, confirming its predominant enzymatic role within TSCs. However, depletion of 5-hmC does not affect global DNA methylation levels. As we see little effect on the global DNA methylation landscape, we can conclude that the dynamic regulation we describe is largely independent of enzymatic DNA demethylation.

We have further added new EED ChIPs in WT ESCs and TSCs to confirm that the PRC2 complex itself continues to occupy intermediately methylated chromatin within TSCs. We also provide new results that validate our EZH2 inhibitor's effects on PRC2 complex formation, genomic occupancy, and catalytic activity (see new **ED Figures 8c-f**). In line with our original discussion points, these experiments argue that the EZH2i's effects are predominantly to impede catalytic methylation of H3K27 without substantially impacting complex formation. Simultaneously, compared to the global depletion of H3K27me3 during treatment, we see a more subtle loss of PRC2 genomic occupancy (see **ED Figures 8d and e**).

Finally, we have also included better descriptions of the different TSC lines tested and ensured that parent lines for independent knockouts are highlighted and included in all extended data as a matched reference for the dynamics we describe.

Together this brings the number of genetic perturbations to five knockout TSC lines, which have each undergone comprehensive molecular and biochemical characterization. These cell lines and data should be of value to other groups with interest in epigenetics and placental biology as well.

(C) Highlight the biological implication and significance of your findings (Reviewer 2).

Although our manuscript is largely focused on the genomic nature of this non-canonical interaction between PRC2 and DNA methylation within TSCs, we very much appreciate that our findings benefit from additional developmental context. To this end, we have added substantial additional analyses to incorporate our findings alongside public in vivo data sets (**ED Figures 2e and 7a,b**), as well as to interrogate the transcriptional responses of TSCs to global epigenetic regulator disruption from a developmental perspective (**ED Figure 10**). Finally, we have also added language within our results and discussion highlighting the relevance of these findings to both genome and developmental biology (Pages 4, 9 and 11).

We believe our findings have several implications. Most importantly, the placenta is an essential and relatively understudied organ with a rather unique and incompletely understood form of epigenome regulation, most frequently characterized by global intermediate methylation. To provide better context and acknowledge the work of other groups in this field, we have improved our presentation of prior work, added 7 new references **7-13** and integrated their data into our analysis (Page 9, **ED Figure 10**).

Based on our current understanding of the 200 or more somatic cell types analyzed to date, all use a canonical form of genome regulation that is well characterized. Most fundamentally, embryonic cells utilize PRC1 and 2 to regulate discrete domains around the promoters of developmental genes and preserve these in an unmethylated state. As these regions are methylated in the TSC epigenome, the prevailing view is that maintenance of this landscape proceeds independently of H3K27me3 through pathways that utilize DNA methylation. From this perspective, the genomic finding that these modifications co-occupy developmental gene promoters, and more broadly maintain the genome in a dynamic methylation state, is highly novel.

In our revised manuscript, we now include analysis of E6.5 ExE and E15 and E18 labyrinth and junctional zone DNA methylation profiles¹⁴ (GSE84350), which confirm that the epigenetic landscape of TSCs is globally preserved for the duration of the placental life span, including the maintenance of an intermediately methylation state consistent with dynamic turnover (**ED Figure 2e**). These results provide in vivo validation of our initial findings in TSCs, including the ability to maintain dynamic/intermediate methylation indefinitely (as shown for our extended passaging, subcloning and DNMTi pulse experiments in **Figure 1d and ED Figures 2b, and 7f,g**).

We also provide additional analysis to explain the divergence in epigenetic effects of zygotic *Eed* disruption (described in our prior publication Ref 15) and acute knockout in TSCs (described here). Specifically, we demonstrate that zygotic *Eed* KO appears to impact the ability for placental or embryonic progenitors to establish distinct epigenetic states at implantation, leading to characteristic methylation of CpG island shores and protection of CpG islands within both lineages (see **ED Figure 7a,b**). The contrasting effects of acute TSC knockout after the extraembryonic landscape has been established is a novel finding that necessitates further investigation *in vivo*, which we now highlight in greater detail within an expanded discussion section.

We also address reviewer comments regarding the effects of DNMT or EZH2 inhibition on gene expression of key markers associated with placental cell types, including those of the labyrinth and junctional zones as well as for trophoblast giant cells (see new **ED Figure 10**, curated from Refs 7-11,13). Of these lineages, we see very minimal significant deviation from the TSC transcriptome, other than minor downregulation of TSC associated genes that would be generally consistent with cell cycle arrest. The directed differentiation of TSCs remain less well characterized than ESCs, but we feel this additional diligence highlights the striking effect of dual inhibition on basic cellular processes (cell cycle progression) as well as the marked resilience of these cells to inhibition of either pathway in isolation.

Finally, we highlight a higher-level implication in our closing paragraphs, namely the possibility that some of these principles may also hold for abnormal disease states including tumorigenesis. As our paper is exclusively focused on the murine TSC model, we would like to keep that section at the end but not overstate the connection based on the developmental data presented in this manuscript.

Minor points:

(D) All other referee concerns pertaining to strengthening existing data, providing controls, methodological details, clarifications and textual changes, should also be addressed.

We have done a thorough pass to address all points and improved/expanded the text throughout.

(E) Finally, please pay close attention to our guidelines on statistical and methodological reporting (listed below) as failure to do so may delay the reconsideration of the revised manuscript.

As requested, we have added uncropped images of all gels/blots as Source Data files and provide a comprehensive summary of all data.

Comments to the Reviewers:

Reviewer #1:

Remarks to the Author:

Mammalian extraembryonic tissues exhibit distinct chromatin characteristics compared to embryonic tissues, which may be relevant for their function. In particular, DNA methylation exists at intermediate levels (at so called Partially Methylated Domains, PMDs) and co-occur with usually antagonistic Polycomb-associated H3K27me3 marks. The reasons for these non-canonical epigenomic patterns are unknown. To gain mechanistic insights, Weigert et al. used here cultured trophoblast stem cell lines (TSCs), allowing them to perturb the dynamics of DNA methylation and H3K27me3 at will. Using four independent TSC lines (2 females and 2 males), they could conclude that cultured TSCs indeed reproduce the extraembryonic-specific chromatin features observed in vivo, using both short and long-read sequencing. They further demonstrated that intermediate DNA methylation levels in TSCs 1-reflect continuous opposition between DNA methylation gain and loss-rather than a failure to maintain DNA methylation across cell divisions-that relates to a functional interplay between DNMT3B and PRC2, which is reflected by direct physical interactions between DDE and DNMT3B. Finally, using inhibitors of DNMT1 and EZH2, they went into deeper details into the relative dynamics of each mark, and further demonstrated that one or the other mark is sufficient to maintain transcriptional patterns of gene showing co-enrichment of the two marks, while ablation of both marks strongly impacted on core cellular function, decreased proliferation and induced senescence.

This is an elegant study and clearly written study, which makes the best usage of both genetic and chemical alterations to highlight the genetic link between DNA methylation and PRC2 in shaping the extraembryonic epigenome, and its remarkable ability to be autonomously restored after perturbations of one or the other components.

We appreciate this comment and find it is a well-articulated summary of the central (and unexpected) findings. The plastic or elastic nature of this regulation is indeed remarkable and we have added substantial additional details to confirm it is an intrinsic and global feature of genome regulation within TSCs.

Functionally, it also reveals the need of both marks for TSC functioning, although this is not a specific feature of TSCs, as it is known ESCs are also profoundly impacted by the lack of both marks. This is one of the weaknesses of the study, as the usage of DNMT and EZH2 inhibitors induces large scale genomic effects, which may not be related at all to the regulation of TSC-specific domains of DNA methylation and H3K27me3 co-enriched territories.

The reviewer makes another key point here that aspects of the developmental reasons for this epigenetic landscape are still missing, and we have made efforts to strengthen our mechanistic insights to highlight the unusual nature of this landscape as motivation for future research in this area. Regarding impact, we would like to highlight that, although mESCs also show a joint dependency on these modifications, they can compensate for one another in pluripotent cells but are still largely mutually exclusive in terms of genomic occupancy (eg. Ref 16). That TSCs are resistant to individual loss of DNA or H3K27me3 through enzymatic inhibition or genetic

mutation may be a shared feature between them and mESCs, but we believe this is in fact a notable finding in and of itself given the many unique aspects of the placental lineage. Furthermore, the persistence of this landscape throughout subsequent gestation (now highlighted in **ED Figure 2e**) suggests this form of regulation may persist for longer within the placental lineage than within the embryo proper, which becomes highly dependent on DNA methylation within several days following implantation.

We have taken the effort to highlight these intriguing behaviors and the relevance of further in vivo work in our discussion, and have also more thoroughly contextualized our expression data in **ED Fig. 9c** (pasted below for your convenience) that the inhibitors affect much of the genome but not the genes associated with the hypermethylation signature.

One limitation also is that we finally do not learn how the interplay between DNA methylation and H3K27me3 operates in TSCs, while this was the original and announced goal of the study.

It is still unclear as to why the co-occurrence of the two marks is tolerated genome-wide in TSCs, even at hypermethylated CGIs. Maybe different flavors of the PRC2 complex exist in this cell type, for example, but this is not studied here.

As outlined, we have added additional experiments to strengthen the conclusion regarding the co-occurrence and interplay, including EED ChIP-seq, ChIP-BS-seq, and quantitative MS of 5-methyl and 5-hydroxymethylcytosine. We also added the TET3 knockout to further clarify the minimal role of this pathway in maintaining the intermediate methylation landscape of TSCs. As the reviewer points out, different avenues may be pursued from here on to identify the upstream signals that trigger the global reconfiguration, how the redirection of the repressive pathways is orchestrated, and its developmental function. We have expanded our discussion on page 10 and 11 to contextualize the relevance of our findings, including these unanswered questions. Similarly, we have added additional mESC interactome data for PRC2 in **ED Figure 8g-i**. We find that, although PRC2 does share certain interaction partners between ESCs and TSCs, unique interaction partners in either context may be required to redirect these regulators from mutually-exclusive to shared genomic functions.

My major concerns are highlighted below, and these are experimentally addressable:

- One major claim of the manuscript is the unique genome co-occurrence of DNA methylation and PRC2-related H3K27me3 in TSCs compared to ESCs, confirming what the authors had previously observed in dissected extraembryonic (EXE) tissues at E6.5. However, to demonstrate that the two marks really exist at the same locus in the same cell and not in separate cells, the authors need to perform ChIP-seq followed by bisulfite sequencing. This was challenging in *in vivo* EXE tissues, regarding the low amount of material, but this should not be a limitation anymore in cultured TSCs. Results from this experiment would be equally interesting, whether they confirm direct co-occupancy or reveal differential occupancy in distinct cells. And this would provide clearer insights into the relationship between the two marks.

We fully agree with the reviewer that ChIP-BS-seq would provide additional value and confidence for our central observations. We also agree that this is experimentally feasible.

To ensure high quality data, we first set out to characterize additional antibodies (Thermo Fisher MA5-11198, Merck Millipore 07-449, CST #9733). After updating and improving our existing protocols, we performed ChIP-seq followed by bisulfite sequencing for TSCs as well as ESCs (2 replicates each). We generated on average around 72 million paired-end reads per sample and confirm the inverse relationship between H3K27me3 and DNA methylation in ESCs but global co-occupancy of these modifications in TSCs (**Figures 2h,i, ED Figure 4c**). In line with our previous observations, we find the methylation rates obtained from this experiment are virtually identical to the methylation levels and patterns observed in WGBS. Combined with our additional inhibitor-withdrawal experiments (which highlight the stability of the TSC epigenome to return to this state after perturbation) this surprising finding confirms the joint occupancy of TSC chromatin by both DNA and H3K27 methylation.

To further codify these relationships, we performed EED ChIP-seq and generated genome-wide occupancy maps for PRC2 in WT TSCs and ESCs, which establish a high concordance between the enrichment for the PRC2 complex with H3K27me3 (see **Figures 2h,i, ED Figure 4c**).

We also explored the possibility of DNMT3B ChIP-seq, but did not pursue these efforts given the extensive work of others that demonstrate the difficulty in acquiring reliable genomic distributions for this regulator class: other labs have published enrichment data (mostly using tagged versions of DNMT3s), including Baubec et al., Nature 2015 – biotin tagged DNMT3B1 and DNMT3A2 (stably integrated RMCE), Nowialis et al. Nature Com. 2019 – FLAG tagged DNMT3B (lentiviral transduction), Rinaldi et al. Cell stem cell 2016 – endogenous DNMT3A/B ChIP in human epidermal stem cells/diff. keratinocytes, Manzo et al., Methods Mol. Biol., 2018. We also considered the lack of ChIP-grade anti-DNMT antibodies, which would lead to poor signal-to-noise ratios when generating genome-wide binding maps. Based on these previous works, we can conclude that quantifying stable occupancy/enrichment is particularly challenging for factors with global functions and broad binding profiles (see Baubec et al. 2015). We also believe that the global nature and high catalytic activity of *de novo* methyltransferases is sufficiently confirmed using our functional KO studies and DNMT1 inhibitor/withdrawal experiments, both of which establish a global function for the *de novo* methyltransferases to preserve intermediate methylation levels.

In summary, our 4 samples of Eed ChIP and 4 samples of H3K27me3 ChIP-BS have added several details that support our proposed model. We have also done additional IPs followed by Western blot that support the interaction between DNMT3B with PRC2 and also highlight the continued stability of the PRC2 complex upon inhibition with the EZH2 inhibitor EPZ6438 (**ED Figure 8c**). Further, the inclusion and validation of our TET3 KO suggests only limited involvement of the catalytic demethylation pathways to maintain this landscape.

- To document whether the co-occurrence of H3K27me3 and DNA methylation in TSCs results from antagonistic relationships between EED/PRC2 and DNMT3B that are driven by direct physical interactions between these two entities, the authors used EED-centered co-IP and

mass spec. However, one necessary and essential control that is missing here would be to carry out such direct interaction assay in ESCs, where the two marks are usually exclusive.

We agree with the reviewer that the biochemical interactions between EED/DNMT3B is an important feature of our manuscript, but would like to highlight that enrichment using these strategies does not necessarily confirm a direct interaction so much as biochemical association. We have made the interpretational details of this clearer in the manuscript.

Additionally, we now include additional IP Western Blot data for EED in mESCs. Notably, we still see this interaction with DNMT3B in mESCs, which is itself consistent with prior descriptions in pluripotent cells^{3,4}. We have included this data in the manuscript (**ED Figure 8g**) as well as language that contextualizes this finding in the corresponding results section and the discussion of the text. Namely, we do not consider this finding problematic, in large part because of the substantial epigenomic and genetic work within the rest of the manuscript confirming that these regulators operate on the same loci in TSCs and their modifications co-occupy the same nucleosomes (see above).

The shared presence of these regulators in TSCs and mESCs, as well as their biochemical interactions being similar according to IP-Western, highlights the need for further investigation into how their regulatory behaviors could change according to developmental context. To this end, we also include IP-MS on EED in mESCs using the same antibody but against an IgG control. We confirm the pulldown of the full PRC2 complex by significant enrichment of SUZ12 and EZH2, which we also see in TSCs. Notably, in these data, DNMT3B does not meet statistical criteria for being called as enriched in mESCs, suggesting it may be weaker in pluripotent cells than TSCs. Using this data, we compare and contrast significant interaction partners to highlight shared and unique features of the PRC2 interactome in TSCs (**ED Figure 8g-i**). Notably, we find that while the shared interactome between mESCs and TSCs converge on core regulators associated with H3K27 methylation (SUZ12, EZH2, JARID2, AEPB2, PHF19, RBBP4, and others), the mESC and TSC interactomes appear to diverge according to local vs global genomic regulatory functions (see **ED Figure 8h,i**). For example, the mESC interactome appears to be specifically enriched for biological and molecular functions associated with pre-mRNA binding and mRNA processing, a well-established interaction for this regulator at “poised” promoters during the early stages of transcriptional induction¹⁷⁻²⁰. In contrast, we see substantially more interactions with multiple protein families with nucleolar, nuclear matrix, RNA binding/processing functions and broad genomic binding within TSCs.

Although largely descriptive, we believe these proteomic data provides notable additional context for interpreting our functionally-validated results and thank the reviewer for this suggestion.

- In their former in vivo paper (Smith et al. 2017), the authors reported that Eed-KO EXE tissues did not show changes in genome-wide DNA methylation levels and loss of DNA methylation at hypermethylated CGIs and concluded that de novo DNA methylation of CGIs required PRC2 in this tissue. Here, in cultured TSCs, Eed-KO rather results in genome-wide gain of DNA

methylation, including at PMDs and hyper CGIs, therefore highlighting antagonistic relationships. How can the authors reconcile these opposite results? It is possible that I missed differences in the genomic sites that were assessed in the two studies.

We thank the reviewer for requesting clarity in this point and have added additional context to the manuscript as well as the additional Extended Data Panels **7a and b** to address this point. Specifically, our 2017 paper (and 2020 follow up, see Ref 15) perturb epigenetic regulators zygotically, meaning they are absent at the time of genome remethylation at implantation (the moment where embryonic and extraembryonic landscapes emerge). In this context, we see that the epigenetic landscapes of the epiblast and ExE merge towards similar aberrant patterns consistent with loss of PRC2 activity (**see figure below, now ED Figure 7b**). Our efforts in TSCs therefor suggest that PRC2 has different functions to first establish the disparate landscapes between the embryo and placenta and then to maintain the placental landscape alongside DNMT3B. We have added this context and comparative analysis to the corresponding sections in the text as well as highlight the need for further investigation using placenta-specific knockouts in the discussion.

- One key question is whether the H3K27me3 mark or EED/the PRC2 complex itself is required for generating intermediate DNA methylation levels. Because DNA methylation gain is equally observed in Eed-KO, where both H3K27marks are abolished and the PRC2 complex is destabilized, and upon treatment with inhibitor of EZH2 activity, the authors conclude-in their discussion-that the mark itself may be responsible for a competition with DNMT3B. As a support, they mention that "the EZH2i is expected to preserve the complex and only block its catalytic activity". How solid is this statement? It is not backed up by publication references. Do we know indeed that the PRC2 complex is not altered in its composition upon EZH2i? Because the authors reported direct interactions between EED/PRC2 and DNMT3B, the question remains open.

We appreciate this request from the reviewer and would highlight the extensive use of this and other small molecule inhibitors in the field without proper due diligence in this area. We now include several additional experiments to characterize the effects of this inhibitor as it differs from genetic knockout. These include IP-Westerns that establish slight downregulation of all three complex components (EED, SUZ12, EZH2) but continued formation of the complex (**ED Figure 8c**). Moreover, we include ChIP-seq for EED in EZH2i treated cells that confirm continued occupancy of EED within the genome, though with slightly lower signal intensity compared to the untreated TSC control (**ED Figure 8d,e**). Notably, EED signal does change more at certain positions more than others, which could indicate the longer-term effects on TSCs after continued culture, but this does not impact the overall claim that EZH2i treatment does not impact the ability for PRC2 to form our bind the genome.

We have added these panels as well as updated our conclusions to include these considerations, which we highlight are all spaces for further investigation. They do not affect our major conclusions or claims, that both H3K27me3 and DNA methylation co-occupy chromatin on a global scale and that they can be reversibly altered and brought back to the same unusual form of genome regulation. They do highlight an important consideration, that PRC2 KO may have a more dramatic effect on chromatin status than inhibitor treatment, which we highlight as a point for further consideration in the text.

Minor comments:

- Compared to male ESCs, female ESCs tend to lose DNA methylation genome-wide. Although it does not seem to be the case from the presented figures, the authors should mention in their manuscript whether female and male TSCs show the same DNA methylation levels.

We thank the reviewer for this comment, we have now included a panel that confirms the sex of our four TSC lines (**ED Figure 1d**) and indicated the sex of the respective lines below the violin plots comparing methylation levels across our two male and two female lines (**ED Figure 1e**).

Reviewer #2:

Remarks to the Author:

In this study, the authors undertake a detailed analysis of the status of DNA methylation and Polycomb regulation in mouse trophoblast stem cells. They confirm that the intermediate state of DNA methylation observed in trophoblast *in vivo* is maintained in the *in vitro* cell lines. They show that there is a balanced stable antagonistic relationship between the two repressive pathways in maintaining this unusual intermediate state. They use appropriate sequencing tools and algorithms to assess the epigenetic landscape in the cells and they also use a combination of genetic mutations and chemical inhibitors to verify the roles of the different DNMTs and Polycomb genes in this process.

Overall, the quality of the data presented seems high and the datasets will be a strong resource for the field. My only major query relates to the lack of much discussion or experimentation relevant to the biological significance of this intermediate methylation state. The authors state that there is little work on the epigenetic regulation of TS cells and, while this may be generally true, there are a number of groups who have contributed to this area in both stem cells and in the embryo. Miriam Hemberger's group in particular- note Senner *et al*, *Stem Cells* 2012- Branco *et al* *Dev Cell* 2016. Also note Legault *et al* 2020 *Epigenetics: Developmental genome-wide DNA methylation asymmetry between mouse placenta and embryo*. This study identifies specific DMRs between embryo and placenta in early development and shows that hypomethylated DMRs in the placenta are enriched in genes involved in reproductive and germline functions. They also look at the roles of the DNMTs in the process. It would be good to integrate the findings in the current study more carefully with their potential biological significance as derived from previous studies.

We thank the reviewer for their overall enthusiasm and for their curiosity about the possible developmental function or meaning of this form of epigenetic regulation. Although our manuscript is largely focused on confirming the nature of this epigenetic interaction, we agree that additional biological and developmental context would substantially improve how these findings motivate the field.

To these ends, we have incorporated additional *in vivo* analysis to confirm that this landscape persists through gestation into the later stages of placental development, including within both the labyrinth and junctional zones (**ED Figure 2e**). We also specifically examine the transcriptional effects of our perturbations on ExE hypo DMRs as identified by Legault *et al* (**ED Figure 10**).

We have also better incorporated the work of prior groups in this space, particularly in regards to the transcriptional effects of the DNMT1 inhibitor, which does show some increased transcription at previously methylated germline associated genes. We specifically highlight that this may reflect the overall overlap between gametogenesis and placental gene regulatory networks, which may be further amplified once DNA methylation is removed. Please also see the added context to the text (note the refs 63-65 in this short section are from the manuscript and not from this document):

Notably, TSCs (and the placenta in general) share aspects of their gene regulatory network with the male germline, including multiple genes whose promoters are otherwise methylated within the embryo proper⁶³⁻⁶⁵. Although we saw no effect on the expression of shared gametogenesis-placental genes (**ED Fig. 10a**), the de-repression of other members of this network by DNMT inhibition may reflect a role for DNA methylation as a buffering mechanism during placental development.

Because this largely describes a negative result, we have elected to show the gene expression for known TSC/progenitor and TGC markers as **ED Figure 10** as both boxplots and a per gene resolution heatmap and provide the full resolution heatmap for each placental state.

It would also be useful to have more data from the current study on the changes in morphology and gene expression when TS cells were treated with inhibitors. Fig 5a shows that TS cells change morphology after treatment with inhibitors- it is stated that they flatten and senesce. But looking at the images suggests they might be ceasing mitosis and becoming trophoblast giant cells- the terminal differentiation stage for TS cells. They state that genes involved in core cellular functions are affected- but did they look to see if cells differentiated? And what happens to the DMRs when the TS cells differentiate?

We also thank the reviewer for requesting this additional clarity, as our initial hypothesis was also that simultaneous depletion of DNA and H3K27 methylation may be triggering differentiation into TGCs. We now include an extensive panel series related to classic and newly described markers of differentiated placental cell types derived from scRNA-seq studies, as well as for the complete prolactin locus (Refs 7-13). In general, these feature sets show minimal transcriptional changes across our inhibitor series, although they do show a subtle decrease in canonical TSC/ExE marker gene expression as well as a moderate gain of some (but not all) Trophoblast Giant Cell (TGC) markers (see **ED Figure 10**). To us, it is difficult to discriminate between a direct effect on these genes as a consequence of the inhibitor treatment

and low-level destabilization of their stem cell identity as part of their cell cycle arrest. To us, these data highlight the comparative resilience of TSCs to differentiation in the absence of major epigenetic pathways, properties that are otherwise only described for naïve mESCs. We similarly highlight the genes associated with ExE-specific hypomethylation as described by Legault et al., which again show only minimal sensitivity to these inhibitors.

Similarly, in **ED Figure 2e** we examine the in vivo dynamics of ExE hyper CGI containing genes from the E6.5 ExE into the late stage labyrinth and junctional zones (taken from Ref 14). There we find minimal dynamics of these regions across the entirety of placental development, as well as persistence of global intermediate methylation levels at HMDs, PMDs, and developmental CGI promoters. These results confirm that this landscape is a general feature of the placental epigenome, the developmental role for which remains to be discovered and requires functional evaluation in vivo.

The reviewer does highlight the possibility of examining the differentiation potential of TSCs under various inhibitor treatments. However, the protocols for doing such are largely limited to trophoblast giant cells and the heterogeneity of this induced state is not well described. Given the comparative lack of control we feel we would have with this system, we felt it better to highlight gene expression changes in response to inhibitor treatment and highlight the need to investigate their roles in placental differentiation as part of a dedicated follow up study.

Reviewer #3:

Remarks to the Author:

Extraembryonic epigenome is less studied compared to embryonic cells. In this manuscript, the authors utilized trophoblast stem cells (TSCs) as a model to explore the extraembryonic lineages characteristics. They found TSCs preserved hypermethylated CGIs, which also contained high levels of H3K27me3. These non-canonical intermediate methylated CGIs were targets of DNMT3B and PRC2, which physically interact and antagonize each other, ensuring the special DNA methylation landscape of TSCs. The epigenome of extraembryonic lineages is indeed worth profiled. However, the relationship between methylome and H3K27me3 distribution seems not to be consistent between TSCs and extraembryonic ectoderm (ExE). The main shortage of the study is the mechanism underlying the co-occupancy of H3K27 methylation and DNA methylation. The authors provided some circumstantial evidence arguing that DNMT3B and PRC2 may directly interact. This really needs to be solidly clarified before publication.

Major points:

1. 4 different TSC lines were used in this study. However, genetic disruptions were performed with different lines: TSCDnmt3bKO from TSC2 (female), TSCRnf2KO and TSCKdm2bKO from TSC1 (male), TSCEedKO from TSC3 (male). Violin plots in ED Fig. 1d lacks TSC1 cells. Because TSCRnf2KO and TSCEedKO were derived from different parental cells, it is not so solid to conclude that "PRC2 knockout cells showed the most dramatic increase in DNA methylation" without illustrating the parental epigenome status respectively.

Thank you for the suggestion, we have now included the specific TSC lines utilized to generate these KO's within the Extended data (most prominently in **ED Figures 1e,f and 5b**) to highlight that the general variation across WT TSC lines is far less substantial than those seen for any of our regulator knockouts.

2. To my understanding, TSC cells were explored to mimic the in vivo system of extraembryonic lineages. In this manuscript, *eed*-null results in global increase of DNA methylation as shown in Fig 3c, 3d, 3h and ED Fig. 4b-f, except canonical CGIs. However, in the Nature paper published by the authors in 2017, global DNAm patterns were preserved in *eed*-null ExE, while CGI methylation was disrupted. What is the explanation for the opposite patterns?

We thank the reviewer for this additional request and have addressed these issues above for Reviewer 1 (copied here for the reviewer's convenience). In short, these data suggest that PRC2 has roles both establishing and maintaining the placental epigenome, as zygotic knockouts show a failure to methylate target CpG islands and acute knockout within TSCs leads to global hypermethylation. We've included our interpretation in the revised manuscript, as well as added contextualizing Extended Data Figures (**ED Figure 7a and b**). We also endeavored to derive TSCs from *Eed* KO blastocysts but with no success, consistent with our in vivo findings that PRC2 is essential for early placental differentiation.

Comments addressed to Reviewer 1:

We thank the reviewer for requesting clarity in this point and have added additional context to the manuscript as well as the additional Extended Data Panels **7a and b** to address this point. Specifically, our 2017 paper (and 2020 follow up, see Ref 15) perturb epigenetic regulators zygotically, meaning they are absent at the time of genome remethylation at implantation (the moment where embryonic and extraembryonic landscapes emerge). In this context, we see that the epigenetic landscapes of the epiblast and ExE merge towards similar aberrant patterns consistent with loss of PRC2 activity (**see figure below, now ED Figure 7b**). Our efforts in TSCs therefor suggest that PRC2 has different functions to first establish the disparate landscapes between the embryo and placenta and then to maintain the placental landscape alongside DNMT3B. We have added this context and comparative analysis to the corresponding sections in the text as well as highlight the need for further investigation using placenta-specific knockouts in the discussion.

3. Fig. 2g, and ED Fig. 3g, the author concluded that "H3K4me3 retained its expected negative correlation with DNA methylation". However, DNA methylation level of HMD, PMD and Hyper CGIs were elevated in TSC*Kdm2b*KO cells in Fig 3c and 3d? In regions of "CGIs hypermethylated in PRC KOs" in ED Fig. 5c, DNA methylation level also got enhanced along with the increased H3K4me3 signals in TSC*Kdm2b*KO cells. Any explanations?

We thank the reviewer for pointing out this lack of clarity on our part and have rephrased these points in the text. First, we meant to highlight that, overall, hypomethylated CpG islands in TSCs retain the canonical relationship between H3K4me3 and DNA methylation (high H3K4me3 and low DNA methylation), and this holds true for CGIs that are not methylated by any PRC regulator KO. We have revised this section of the text accordingly:

As expected, H3K4me3 enriched regions of the genome remained negatively correlated with DNA methylation in TSCs, particularly at the CGI-enriched promoters of housekeeping genes. In keeping with this rule, H3K4me3 was generally depleted from methylated CGIs, despite their frequent localization within developmental gene promoters (**Fig. 2g, ED Fig. 3g**). In contrast, we also found that H3K4me3 was enriched across a variety of intermediately methylated intergenic and repetitive contexts, in particular at IAP-family endogenous retroviruses, that may reflect a lineage-specific activity (**ED Fig. 3d,g**)^{38,39}.

Moreover, the full number of CGIs that become hypermethylated in PRC KO's is larger than the number that were already called as methylated when ExE is compared to Epiblast, many of which are H3K4me3. We have clarified this concept in the text as well updated **ED Fig. 6b** to reflect this point:

To our surprise, core PRC component (EED or RNF2) knockouts exhibited strong genome-wide DNA methylation gains, including across CGIs. In particular, PRC2 knockout cells showed the most dramatic increase in DNA methylation, and include thousands of CGIs that were previously unmethylated in WT TSCs (**Fig. 3b-d, ED Fig. 5,6**).

Finally, KDM2B is a subcomplex component of PRC1 (ncPRC1.1). When knocked out, a subset of CpG islands that become hypermethylated in EED or RNF2 knockouts remain unmethylated. In these contexts, we find that the underlying epigenetic status remains enriched for H3K4me3, suggesting that removal of this modification is required for these islands to become hypermethylated. Please see **Figure 3f and revised text**, below:

Finally, our knockout for the non-canonical PRC1 complex component KDM2B showed the same global trend, but the majority of PRC-sensitive CGIs ($n=3,276$ of 3,967) remained at WT methylation levels (**Fig. 3b-d, ED Fig. 6b**).

4. The chromatin colocalization profile of DNMT3B and PRC2 is absent. DNMT3B and PRC2 ChIP-seq should be included. Also, while the co-occurrence of DNA methylation and H3K27 methylation were analyzed in bulk level, it's better to perform bisulfite sequencing using the

DNA retrieved from H3K27me3 ChIP which would be a direct evidence showing the co-occurrence of the two modifications.

We thank the reviewer for this suggestion and have added both EED ChIP-seq and ChIP-BS-seq data to the manuscript (**Figure 2h-j and ED Figure 4**). Our findings are described above in the revision summary and to Reviewer 1. In short, these results confirm that H3K27me3 and intermediate methylation co-occupy the same loci within TSCs, and that H3K27me3-enriched chromatin has identical DNA methylation levels compared to our WGBS data. Our EED ChIP-seq further confirms the continued presence of this regulator across developmental gene promoters. Combined with our functional observations derived from our genetic knockout study, these results confirm the unique global relationship between these regulator classes, which can persist indefinitely in TSCs and may do so across the entirety of placental differentiation.

Please also note our comments above in regards to the DNMT3B ChIP, which we copy here for the Reviewer's convenience:

We also explored the possibility of Dnmt3b ChIP-seq, but did not pursue these efforts given the extensive work of others that demonstrate the difficulty in acquiring reliable genomic distributions for this regulator class: other labs have published enrichment data (mostly using tagged versions of Dnmt3s), including Baubec et al., Nature 2015 – biotin tagged DNMT3B1 and DNMT3A2 (stably integrated RMCE), Nowialis et al. Nature Com. 2019 – FLAG tagged DNMT3B (lentiviral transduction), Rinaldi et al. Cell stem cell 2016 – endogenous DNMT3A/B ChIP in human epidermal stem cells/diff. keratinocytes, Manzo et al., Methods Mol. Biol., 2018. We also considered the lack of ChIP-grade anti-DNMT antibodies, which would lead to poor signal-to-noise ratios when generating genome-wide binding maps. Based on these previous works, we can conclude that quantifying stable occupancy/enrichment is particularly challenging for factors with global functions and broad binding profiles (see Baubec et al. 2015). We also believe that the global nature and high catalytic activity of *de novo* methyltransferases is sufficiently confirmed using our functional KO studies and for DNMT1 inhibitor/withdrawal experiments, both of which establish a global function for the *de novo* methyltransferases to preserve intermediate methylation levels.

5. Fig. 4d, co-IP, which confirmed the physical interaction between PRC2 and DNMT3B, was performed with 110 mM NaCl, a salt concentration far below the standard line, why? Can the authors repeat this experiment with higher salt concentration?

We would like to thank the reviewer for pointing out this miscommunication in our methods. To clarify, our co-IPs were carried out at 220mM NaCl, and not 110 mM NaCl as originally interpreted, due to the additional NaCl and KCl present in the 0.5x PBS that comprises Nuclear Lysis Buffer (NLB). Briefly, our 2X NLB stock contains 274mM NaCl (and 5.4mM KCl) that comes from the PBS stock solution, which is supplemented with an additional 600mM NaCl, raising the total and final monovalent salt concentration to 880mM. 0.5X NLB therefore contains

$(274+600)/4 = 218.5\text{mM NaCl}$ (and 1.35mM KCl), so a final monovalent salt concentration of 220mM .

We selected these conditions based upon the previous literature for immunoprecipitating and characterizing PRC2 in mammalian cells, including:

- 1) Dynamic Protein Interactions of the Polycomb Repressive Complex 2 during Differentiation of Pluripotent Cells, Oliviero et al. 2016
- 2) ASXL1 Mutations Promote Myeloid Transformation through Loss of PRC2-Mediated Gene Repression, Abdel-Wahab et al. 2012

We have now updated our methods to reflect these experimental details more clearly:

“EED (anti-EED, Abcam, #ab4469) and control IgG (anti-rabbit IgG, Cell Signaling, #2729) immunoprecipitations were carried out using whole-cell lysates prepared from wildtype TS cells. Briefly, ~5-10 mio cells were resuspended with $600\mu\text{L}$ of 0.5x Nuclear Lysis Buffer, (NLB, comprised of 218.5mM NaCl , 1.35mM KCl , $4\text{ mM Na}_2\text{HPO}_4$, $1\text{mM KH}_2\text{PO}_4$, 0.5% Triton X-100, 0.05% Tween-20, $\text{pH}=7.4$) + 1x cOmplete™ Protease Inhibitor Cocktail. We selected these conditions from the published literature as sufficient for stringent characterization of PRC2 subcomplex characterization in mammalian cells. Resuspended samples were sonicated with Bioruptor Sonicator (30s ON/OFF, 5 cycles on LO) and centrifuged for 10 min at $\sim 20.000g$ at 4°C to remove cellular debris. Antibodies were then added to clarified lysates and immune-complexes are allowed to form overnight ($\sim 16\text{hrs}$) in the cold-room with end-to-end rotation“.

6. DNA methylation and H3K27me3 co-existed in hyper CGIs. Does this reflect cell/allele heterogeneity?

We thank the reviewer for this comment and have clarified our definition of methylation entropy in the text (**Figure 1c, ED Fig. 1f**), which specifically distinguishes between models of cellular versus allelic heterogeneity^{21,22}. Namely, the high entropy of these intermediately methylated islands is strongly indicative of allelic heterogeneity and dynamic methylation, which we confirm by examining the DNA methylation status of H3K27me3-enriched chromatin (see above).

Revised text in manuscript:

Interestingly, we found that TSCs maintain their global methylation in a state of high entropy, an intrinsic form of disordered DNA methylation characterized by a large number of unique epialleles distributed across individually measured reads (**Fig. 1c, ED Fig. 1f**). As such, intermediate methylation in TSCs appears to reflect an allelic, population-wide form of epigenetic heterogeneity rather than the persistence of fully methylated reads across subpopulations of cells.

7. Do the Tet enzymes play any roles in the heterogeneity and dynamics of the methylome in TSCs? This needs to be discussed at least.

We agree that the roles of the TET enzymes require additional clarification and now include an additional TET3 KO TSC line (derived from TSC line 1), as well as functional MS-based validation that TET3 is the major enzymatic contributor of 5-hmC in the TSC genome (see **Figures 3b,c** and **ED Figures 5b-d**). Our results establish that, despite globally intermediate methylation levels, TSCs do not show notable enrichment for 5-hmC, nor does TET3 KO lead to globally increased methylation levels. We do not want to rule out the possibility of interactions between the TETs and PRCs may play a role in maintaining this epigenome, particularly given the growing literature of non-catalytic regulator roles for these enzymes, and have made sure our new results are appropriately contextualized in the discussion.

Minor points:

1. Fig. 4b, why was mean methylation level in the right EZH2i-5weeks higher than the left EZH2i-5 weeks? DNA methylation levels increased to ~85-90% after 4 or 5-weeks EZH2i treatment in Fig. 4b and ED Fig. 7a, not ~100% as described in the text.

We have clarified the median methylation values of 5 week EZH2i inhibitor treated TSCs in text to better reflect the levels reached. The differences in methylation levels reflect that the left and right panels reflect independent experiments: the left panel describes the gain and loss of DNA methylation upon inhibitor treatment and withdrawal, while the right panel is designed to test the continued presence of de novo DNMT activity by pulsed treatment with DNMT1i. We have clarified these details within the figure legend to highlight why these methylation values are different.

2. Typo "Mass Spectometry" page 5, line 13.

We have corrected the mistake.

3. Fig. 4e, two dots were labeled as EZH2.

Two unique isoforms of EZH2 are recognized and called by the Mass Spec experiment, we have clarified these details in the corresponding Figure 4e legend: "EZH2 is plotted twice because of the recovery of two distinguishable isoforms, Q61188;D3Z774 and Q6AXH7, respectively."

References:

- 1 Brinkman, A. B. *et al.* Sequential ChIP-bisulfite sequencing enables direct genome-scale investigation of chromatin and DNA methylation cross-talk. *Genome Res* **22**, 1128-1138, doi:10.1101/gr.133728.111 (2012).
- 2 Statham, A. L. *et al.* Bisulfite sequencing of chromatin immunoprecipitated DNA (BisChIP-seq) directly informs methylation status of histone-modified DNA. *Genome Res* **22**, 1120-1127, doi:10.1101/gr.132076.111 (2012).
- 3 Neri, F. *et al.* Dnmt3L antagonizes DNA methylation at bivalent promoters and favors DNA methylation at gene bodies in ESCs. *Cell* **155**, 121-134, doi:10.1016/j.cell.2013.08.056 (2013).
- 4 van Mierlo, G. *et al.* Integrative Proteomic Profiling Reveals PRC2-Dependent Epigenetic Crosstalk Maintains Ground-State Pluripotency. *Cell stem cell* **24**, 123-137 e128, doi:10.1016/j.stem.2018.10.017 (2019).
- 5 Senner, C. E. *et al.* TET1 and 5-Hydroxymethylation Preserve the Stem Cell State of Mouse Trophoblast. *Stem Cell Reports* **15**, 1301-1316, doi:10.1016/j.stemcr.2020.04.009 (2020).
- 6 Senner, C. E., Krueger, F., Oxley, D., Andrews, S. & Hemberger, M. DNA methylation profiles define stem cell identity and reveal a tight embryonic-extraembryonic lineage boundary. *Stem Cells* **30**, 2732-2745, doi:10.1002/stem.1249 (2012).
- 7 Simmons, D. G. & Cross, J. C. Determinants of trophoblast lineage and cell subtype specification in the mouse placenta. *Dev Biol* **284**, 12-24, doi:10.1016/j.ydbio.2005.05.010 (2005).
- 8 Branco, M. R. *et al.* Maternal DNA Methylation Regulates Early Trophoblast Development. *Dev Cell* **36**, 152-163, doi:10.1016/j.devcel.2015.12.027 (2016).
- 9 Marsh, B. & Blelloch, R. Single nuclei RNA-seq of mouse placental labyrinth development. *Elife* **9**, doi:10.7554/eLife.60266 (2020).
- 10 Nelson, A. C., Mould, A. W., Bikoff, E. K. & Robertson, E. J. Single-cell RNA-seq reveals cell type-specific transcriptional signatures at the maternal-foetal interface during pregnancy. *Nat Commun* **7**, 11414, doi:10.1038/ncomms11414 (2016).
- 11 Rossant, J. & Cross, J. C. Placental development: lessons from mouse mutants. *Nat Rev Genet* **2**, 538-548, doi:10.1038/35080570 (2001).
- 12 Simmons, D. G., Rawn, S., Davies, A., Hughes, M. & Cross, J. C. Spatial and temporal expression of the 23 murine Prolactin/Placental Lactogen-related genes is not associated with their position in the locus. *BMC Genomics* **9**, 352, doi:10.1186/1471-2164-9-352 (2008).
- 13 Zhou, X. *et al.* Single-cell RNA-seq revealed diverse cell types in the mouse placenta at mid-gestation. *Exp Cell Res* **405**, 112715, doi:10.1016/j.yexcr.2021.112715 (2021).
- 14 Decato, B. E., Lopez-Tello, J., Sferruzzi-Perri, A. N., Smith, A. D. & Dean, M. D. DNA Methylation Divergence and Tissue Specialization in the Developing Mouse Placenta. *Mol Biol Evol* **34**, 1702-1712, doi:10.1093/molbev/msx112 (2017).
- 15 Grosswendt, S. *et al.* Epigenetic regulator function through mouse gastrulation. *Nature* **584**, 102-108, doi:10.1038/s41586-020-2552-x (2020).
- 16 Walter, M., Teissandier, A., Perez-Palacios, R. & Bourc'his, D. An epigenetic switch ensures transposon repression upon dynamic loss of DNA methylation in embryonic stem cells. *Elife* **5**, doi:10.7554/eLife.11418 (2016).

- 17 Davidovich, C., Zheng, L., Goodrich, K. J. & Cech, T. R. Promiscuous RNA binding by Polycomb repressive complex 2. *Nature structural & molecular biology* **20**, 1250-1257, doi:10.1038/nsmb.2679 (2013).
- 18 Jia, J. *et al.* Regulation of pluripotency and self-renewal of ESCs through epigenetic-threshold modulation and mRNA pruning. *Cell* **151**, 576-589, doi:10.1016/j.cell.2012.09.023 (2012).
- 19 Kaneko, S., Son, J., Bonasio, R., Shen, S. S. & Reinberg, D. Nascent RNA interaction keeps PRC2 activity poised and in check. *Genes Dev* **28**, 1983-1988, doi:10.1101/gad.247940.114 (2014).
- 20 Kaneko, S., Son, J., Shen, S. S., Reinberg, D. & Bonasio, R. PRC2 binds active promoters and contacts nascent RNAs in embryonic stem cells. *Nature structural & molecular biology* **20**, 1258-1264, doi:10.1038/nsmb.2700 (2013).
- 21 Scherer, M. *et al.* Quantitative comparison of within-sample heterogeneity scores for DNA methylation data. *Nucleic Acids Res* **48**, e46, doi:10.1093/nar/gkaa120 (2020).
- 22 Xie, H. *et al.* Genome-wide quantitative assessment of variation in DNA methylation patterns. *Nucleic Acids Res* **39**, 4099-4108, doi:10.1093/nar/gkr017 (2011).

Decision Letter, first revision:

Message: Our ref: NCB-A48515A

29th November 2022

Dear Alex,

Thank you for submitting your revised manuscript "Dynamic antagonism between key repressive pathways in the placental epigenome" (NCB-A48515A). It has now been seen by the original referees and their comments are below. The reviewers find that the paper has improved in revision, and therefore we'll be happy in principle to publish it in Nature Cell Biology, pending minor revisions to comply with our editorial and formatting guidelines.

If the current version of your manuscript is in a PDF format, please email us a copy of the file in an editable format (Microsoft Word or LaTeX)-- we cannot proceed with PDFs at this stage.

Thank you again for your interest in Nature Cell Biology. Please do not hesitate to contact me if you have any questions.

Best wishes,
Stelios

Stylianos Lefkopoulos, PhD
He/him/his
Associate Editor
Nature Cell Biology
Springer Nature
Heidelberger Platz 3, 14197 Berlin, Germany

E-mail: stylianos.lefkopoulos@springernature.com
Twitter: @s_lefkopoulos

Reviewer #1 (Remarks to the Author):

The authors have nicely answered my remarks and questions, and provided new experimental data that greatly strengthen their findings, both mechanistically and conceptually.
I am am fully supportive of the publication of this work.

Reviewer #2 (Remarks to the Author):

This is an extensively revised version of a previous submission. The major takeaways have not changed: They confirm that the intermediate state of DNA methylation observed in trophoblast in vivo is maintained in the in vitro trophoblast stem cell lines. They use appropriate sequencing tools and algorithms to assess the epigenetic landscape in the cells and they also use a combination of genetic mutations and chemical inhibitors to verify the roles of the different DNMTs and Polycomb genes in maintaining a balanced stable antagonistic relationship between the two repressive pathways to ensure maintenance of this unusual intermediate state.

The authors have performed considerable additional experimentation and carefully addressed the different reviewer comments. I have no further comments.

Reviewer #3 (Remarks to the Author):

The authors have adequately addressed my main concerns and I support to publish this study.

Decision letter, Author Guidance:

Message: Our ref: NCB-A48515A

13th December 2022

Dear Dr. Meissner,

Thank you for your patience as we've prepared the guidelines for final submission of your Nature Cell Biology manuscript, "Dynamic antagonism between key repressive pathways in the placental epigenome" (NCB-A48515A). Please carefully follow the step-by-step instructions provided in the attached file, and add a response in each row of the table to indicate the changes that you have made. Please also check and comment on any additional marked-up edits we have proposed within the text. Ensuring that each point is addressed will help to ensure that your revised manuscript can be swiftly handed over to our production team.

In recognition of the time and expertise our reviewers provide to Nature Cell Biology's editorial process, we would like to formally acknowledge their contribution to the external peer review of your manuscript entitled "Dynamic antagonism between key repressive pathways in the placental epigenome". For those reviewers who give their assent, we will be publishing their names alongside the published article.

Nature Cell Biology offers a Transparent Peer Review option for new original research manuscripts submitted after December 1st, 2019. As part of this initiative, we encourage our authors to support increased transparency into the peer review process by agreeing to have the reviewer comments, author rebuttal letters, and editorial decision letters published as a Supplementary item. When you submit your final files please clearly state in your cover letter whether or not you would like to participate in this initiative. Please note that failure to state your preference will result in delays in accepting your manuscript for publication.

Cover suggestions

As you prepare your final files we encourage you to consider whether you have any images or illustrations that may be appropriate for use on the cover of Nature Cell Biology.

Nature Cell Biology has now transitioned to a unified Rights Collection system which will allow our Author Services team to quickly and easily collect the rights and permissions required to publish your work. Approximately 10 days after your paper is formally accepted, you will receive an email in providing you with a link to complete the grant of rights. If your paper is eligible for Open Access, our Author Services team will also be in touch regarding any additional information that may be required to arrange payment for your article.

Please note that *Nature Cell Biology* is a Transformative Journal (TJ). Authors may publish their research with us through the traditional subscription access route or make their paper immediately open access through payment of an article-processing charge (APC). Authors will not be required to make a final decision about access to their article until it has been accepted. Find out more about Transformative Journals

Authors may need to take specific actions to achieve compliance with funder and institutional open access mandates. If your research is supported by a funder that requires immediate open access (e.g. according to Plan S principles) then you should select the gold OA route, and we will direct you to the compliant route where possible. For authors selecting the subscription publication route, the journal's standard licensing terms will need to be accepted, including self-archiving policies. Those licensing terms will supersede any other terms that the author or any third party may assert apply to any version of the manuscript.

<https://mts-ncb.nature.com/cgi-bin/main.plex?el=A3C4BHD7A6BTru7J2A9ftdXOim5vJ5fwCxEYrEuEYRPgZ>

Best regards,

Kendra Donahue
Staff
Nature Cell Biology

On behalf of

Stylianos Lefkopoulos, PhD
He/him/his
Associate Editor
Nature Cell Biology
Springer Nature
Heidelberger Platz 3, 14197 Berlin, Germany

E-mail: stylianos.lefkopoulos@springernature.com

Twitter: @s_lefkopoulos

Reviewer #1:

Remarks to the Author:

The authors have nicely answered my remarks and questions, and provided new experimental data that greatly strengthen their findings, both mechanistically and conceptually.

I am am fully supportive of the publication of this work.

Reviewer #2:

Remarks to the Author:

This is an extensively revised version of a previous submission. The major takeaways have not changed: They confirm that the intermediate state of DNA methylation observed in trophoblast in vivo is maintained in the in vitro trophoblast stem cell lines. They use appropriate sequencing tools and algorithms to assess the epigenetic landscape in the cells and they also use a combination of genetic mutations and chemical inhibitors to verify the roles of the different DNMTs and Polycomb genes in maintaining a balanced stable antagonistic relationship between the two repressive pathways to ensure maintenance of this unusual intermediate state.

The authors have performed considerable additional experimentation and carefully addressed the different reviewer comments. I have no further comments.

Reviewer #3:

Remarks to the Author:

The authors have adequately addressed my main concerns and I support to publish this study.

Attachment: NCB-A48515A Meissner_AuthorGuidance.docx - 13th December 22 14:36:34

Attachment: SNTPS Reporting Summary - Meissner.pdf - 13th December 22 14:36:34

Decision Letter, second revision:

Message: Dear Alex,

I am pleased to inform you that your manuscript, "Dynamic antagonism between key repressive pathways maintains the placental epigenome", has now been accepted for publication in Nature Cell Biology. Congratulations to you and your team!

Once your paper has been scheduled for online publication, the Nature press office will be in touch to confirm the details. An online order form for reprints of your paper is available at <https://www.nature.com/reprints/author-reprints.html>. All co-authors, authors'

institutions and authors' funding agencies can order reprints using the form appropriate to their geographical region.

Please note that *Nature Cell Biology* is a Transformative Journal (TJ). Authors may publish their research with us through the traditional subscription access route or make their paper immediately open access through payment of an article-processing charge (APC). Authors will not be required to make a final decision about access to their article until it has been accepted. Find out more about Transformative Journals

If you have not already done so, we strongly recommend that you upload the step-by-step protocols used in this manuscript to the Protocol Exchange (www.nature.com/protocolexchange), an open online resource established by Nature Protocols that allows researchers to share their detailed experimental know-how. All uploaded protocols are made freely available, assigned DOIs for ease of citation and are fully searchable through nature.com. Protocols and Nature Portfolio journal papers in which they are used can be linked to one another, and this link is clearly and prominently visible in the online versions of both papers. Authors who performed the specific experiments can act as primary authors for the Protocol as they will be best placed to share the methodology details, but the Corresponding Author of the present research paper should be included as one of the authors. By uploading your Protocols to Protocol Exchange, you are enabling researchers to more readily reproduce or adapt the methodology you use, as well as increasing the visibility of your protocols and papers. You can also establish a dedicated page to collect your lab Protocols. Further information can be found at www.nature.com/protocolexchange/about

With kind regards,
Stelios

Stylios Lefkopoulos, PhD
He/him/his
Associate Editor
Nature Cell Biology
Springer Nature
Heidelberger Platz 3, 14197 Berlin, Germany

E-mail: stylios.lefkopoulos@springernature.com
Twitter: @s_lefkopoulos

** Visit the Springer Nature Editorial and Publishing website
at www.springernature.com/editorial-and-publishing-jobs for more information about our
career opportunities. If you have any questions please click here.**

Final Decision Letter: